# Model-Free Active Exploration
# in Reinforcement Learning

**Alessio Russo**
Division of Decision and Control Systems
KTH Royal Institute of Technology
Stockholm, SE

**Alexandre Proutiere**
Division of Decision and Control Systems
KTH Royal Institute of Technology
Stockholm, SE

## Abstract

We study the problem of exploration in Reinforcement Learning and present a novel model-free solution. We adopt an information-theoretical viewpoint and start from the instance-specific lower bound of the number of samples that have to be collected to identify a nearly-optimal policy. Deriving this lower bound along with the optimal exploration strategy entails solving an intricate optimization problem and requires a model of the system. In turn, most existing sample optimal exploration algorithms rely on estimating the model. We derive an approximation of the instance-specific lower bound that only involves quantities that can be inferred using model-free approaches. Leveraging this approximation, we devise an ensemble-based model-free exploration strategy applicable to both tabular and continuous Markov decision processes. Numerical results demonstrate that our strategy is able to identify efficient policies faster than state-of-the-art exploration approaches.

## 1   Introduction

Efficient exploration remains a major challenge for reinforcement learning (RL) algorithms. Over the last two decades, several exploration strategies have been proposed in the literature, often designed with the aim of minimizing regret. These include model-based approaches such as Posterior Sampling for RL [36](PSRL) and Upper Confidence Bounds for RL [4, 25, 2](UCRL), along with model-free UCB-like methods [19, 56]. Regret minimization is a relevant objective when one cares about the rewards accumulated during the learning phase. Nevertheless, an often more important objective is to devise strategies that explore the environment so as to learn efficient policies using the fewest number of samples [16]. Such an objective, referred to as Best Policy Identification (BPI), has been investigated in simplistic Multi-Armed Bandit problems [16, 21] and more recently in tabular MDPs [28, 29]. For these problems, tight instance-specific sample complexity lower bounds are known, as well as model-based algorithms approaching these limits. However, model-based approaches may be computationally expensive or infeasible to obtain. In this paper, we investigate whether we can adapt the design of these algorithms so that they become model-free and hence more practical.

Inspired by [28, 29], we adopt an information-theoretical approach, and design our algorithms starting from an instance-specific lower bound on the sample complexity of learning a nearly-optimal policy in a Markov decision process (MDP). This lower bound is the value of an optimization problem, referred to as the lower bound problem, whose solution dictates the optimal exploration strategy in an environment. Algorithms designed on this instance-specific lower bound, rather than minimax bounds, result in truly adaptive methods, capable of tailoring their exploration strategy according to the specific MDP's learning difficulty. Our method estimates the solution to the lower bound problem and employs it as our exploration strategy. However, we face two major challenges: (1) the

---

Code repository: https://github.com/rssalessio/ModelFreeActiveExplorationRL

37th Conference on Neural Information Processing Systems (NeurIPS 2023).

lower bound problem is non-convex and often intractable; (2) this lower bound problem depends on the initially unknown MDP. In [29], the authors propose MDP-NAS, a model-based algorithm that explores according to the estimated MDP. They convexify the lower bound problem and explore according to the solution of the resulting simplified problem. However, this latter problem still has a complicated dependency on the MDP. Moreover, extending MDP-NAS to large MDPs is challenging since it requires an estimate of the model, and the capability to perform policy iteration. Additionally, MDP-NAS employs a *forced exploration* technique to ensure that the *parametric* uncertainty (the uncertainty about the true underlying MDP) diminishes over time — a method, as we argue later, that we believe not to be efficient in handling this uncertainty.

We propose an alternative way to approximate the lower bound problem, so that its solution can be learnt via a model-free approach. This solution depends only on the $Q$-function and the variance of the value function. Both quantities can advantageously be inferred using classical stochastic approximation methods. To handle the parametric uncertainty, we propose an ensemble-based method using a bootstrapping technique. This technique is inspired by posterior sampling and allows us to quantify the uncertainty when estimating the $Q$-function and the variance of the value function.

Our contributions are as follows: (1) we shed light on the role of the instance-specific quantities needed to drive exploration in uncertain MDPs; (2) we derive an alternate upper bound of the lower bound problem that in turn can be approximated using quantities that can be learned in a model-free manner. We then evaluate the quality of this approximation on various environments: (*i*) a random MDP, (*ii*) the Riverswim environment [47], and (*iii*) the Forked Riverswim environment (a novel environment with high sample complexity); (3) based on this approximation, we present Model Free Best Policy Identification (MF-BPI), a model-free exploration algorithm for tabular and continuous MDPs. For the tabular MDPs, we test the performance of MF-BPI on the Riverswim and the Forked Riverswim environments, and compare it to that of Q-UCB [19, 56], PSRL[36], and MDP-NAS[29]. For continuous state-spaces, we compare our algorithm to IDS[33] and BSP [39] (Boostrapped DQN with randomized prior value functions) and assess their performance on hard-exploration problems from the DeepMind BSuite [41] (the DeepSea and the Cartpole swingup problems).

## 2   Related Work

The body of work related to exploration methods in RL problems is vast, and we mainly focus on online discounted MDPs (for the generative setting, refer to the analysis presented in [17, 28]). Exploration strategies in RL often draw inspiration from the approaches used in multi-armed bandit problems [26, 49], including $\epsilon$-greedy exploration, Boltzmann exploration [57, 49, 26, 1], or more advanced procedures, such as Upper-Confidence Bounds (UCB) methods [2, 3, 26] or Bayesian procedures [52, 58, 14, 44]. We first discuss tabular MDPs, and then extend the discussion to the case of RL with function approximation.

**Exploration in tabular MDPs.** Numerous algorithms have been proposed with the aim of matching the PAC sample complexity minimax lower bound $\tilde{\Omega}\left(\frac{|S||A|}{\varepsilon^2(1-\gamma)^3}\right)$ [25]. In the design of these algorithms, model-free approaches typically rely on a UCB-like exploration [2, 26], whereas model-based methods leverage estimates of the MDP to drive the exploration. Some well-known model-free algorithms are MEDIAN-PAC [42], DELAYED Q-LEARNING [48] and Q-UCB [56, 19]. Some notable model-based algorithms include: DEL [34], an algorithm that achieves asymptotically optimal instance-dependent regret; UCRL [25], an algorithm that uses extended value-iteration to compute an optimistic MDP; PSRL [36], that uses posterior sampling to sample an MDP. Other algorithms include MBIE [47], E3 [22], R-MAX [9, 20], and MORMAX [50]. Most of existing algorithms are designed towards regret minimization. Recently, however, there has been a growing interest towards exploration strategies with minimal sample complexity, see e.g. [60, 28]. In [28, 29], the authors showed that computing an exploration strategy with minimal sample complexity requires to solve a non-convex problem. To overcome this challenge, they derived a tractable approximation of the lower bound problem, whose solution provides an efficient exploration policy under the generative model [28] and the forward model [29]. This policy necessitates an estimate of the model, and includes a forced exploration phase (an $\epsilon$-soft policy to guarantee that all state-action pairs are visited infinitely often). In [51], the above procedure is extended to linear MDPs, but there again, computing an optimal exploration strategy remains challenging. On a side note, in [55], the authors provide an

alternative bound in the tabular case for episodic MDPs, and later extend it to linear MDPs [54]. The episodic setting is further explored in [53] for deterministic MDPs.

**Exploration in Deep Reinforcement Learning (DRL).** Exploration methods in DRL environments face several challenges, related to the fact that the state-action spaces are often continuous, and other issues related to training deep neural architectures [46]. The main issue in these large MDPs is that good exploration becomes extremely hard when either the reward is sparse/delayed or the observations contain distracting features [10, 59]. Numerous heuristics have been proposed to tackle these challenges, such as (1) adding an entropy term to the optimization problem to encourage the policy to be more randomized [31, 18] or (2) injecting noise in the observations/parameters [15, 43]. More generally, exploration techniques generally fall into two categories: *uncertainty-based* and *intrinsic-motivation-based* [59, 24]. Uncertainty-based methods decouple the uncertainty into *parametric* and *aleatoric* uncertainty. Parametric uncertainty [14, 32, 23, 59] quantifies the uncertainty in the parameters of the state-action value. This uncertainty vanishes as the agent explores and learns. The aleatoric uncertainty accounts for the inherent randomness of the environment and of the policy [32, 23, 59]. Various methods have been proposed to address the parametric uncertainty, including UCB-like mechanisms [11, 59], or TS-like (Thompson Sampling) techniques [38, 36, 5, 35, 37, 40]. However, computing a posterior of the $Q$-values is a difficult task. For instance, Bayesian DQN [5] extends Randomized Least-Squares Value Iteration (RLSVI) [38] by considering the features prior to the output layer of the deep-$Q$ network as a fixed feature vector, in order to recast the problem as a linear MDP. Non-parametric posterior sampling methods include Bootstrapped DQN (and Bootstrapped DQN with prior functions) [37, 39, 40], which maintains several independent $Q$-value functions and randomly samples one of them to explore the environment. Bootstrapped DQN was extended in various ways by integrating other techniques [6, 27]. For the sake of brevity, we refer the reader to the survey in [59] for an exhaustive list of algorithms. Most of these algorithms do not directly account for aleatoric uncertainty in the value function. This uncertainty is usually estimated using methods like Distributional RL [8, 13, 30]. Well-known exploration methods that account for both aleatoric and epistemic uncertainties include Double Uncertain Value Network (DUVN) [32] and Information Directed Sampling (IDS) [23, 33]. The former uses Bayesian dropout to measure the epistemic uncertainty, and the latter uses distributional RL [8] to estimate the variance of the returns. In addition, IDS uses bootstrapped DQN to estimate the parametric uncertainty in the form of a bound on the estimate of the suboptimality gaps. These uncertainties are then combined to compute an exploration strategy. Similarly, in [12], the authors propose UA-DQN, an approach that uses QR-DQN [13] to learn the parametric and aleatoric uncertainties from the quantile networks. Lastly, we refer the reader to [59, 45, 7] for the class of intrinsic-motivation-based methods.

## 3 Preliminaries

**Markov Decision Process.** We consider an infinite-horizon discounted Markov Decision Process (MDP), defined by the tuple $\phi = (S, A, P, q, \gamma, p_0)$. $S$ is the state space, $A$ is the action space, $P : S \times A \to \Delta(S)$ is the distribution over the next state given a state-action pair $(s, a)$, $q : S \times A \to \Delta([0, 1])$ is the distribution of the collected reward (with support in $[0, 1]$), $\gamma \in [0, 1)$ is the discount factor and $p_0$ is the distribution over the initial state.

Let $\pi : \mathcal{S} \to \Delta(A)$ be a stationary Markovian policy that maps a state to a distribution over actions, and denote by $r(s, a) = \mathbb{E}_{r \sim q(\cdot|s,a)}[r]$ the average reward collected when an action $a$ is chosen in state $s$. We denote by $V^\pi(s) = \mathbb{E}_\phi^\pi[\sum_{t \geq 0} \gamma^t r(s_t, a_t)|s_0 = s]$ the discounted value of policy $\pi$. We denote by $\pi^\star$ an optimal stationary policy: for any $s \in \mathcal{S}$, $\pi^\star(s) \in \arg\max_\pi V^\pi(s)$ and define $V^\star(s) = \max_\pi V^\pi(s)$. For the sake of simplicity, we assume that the MDP has a unique optimal policy (we extend our results to more general MDPs in the appendix). We further define $\Pi_\varepsilon^\star(\phi) = \{\pi : \|V^\pi - V^{\pi^\star}\|_\infty \leq \varepsilon\}$, the set of $\varepsilon$-optimal policies in $\phi$ for $\varepsilon \geq 0$. Finally, to avoid technicalities, we assume (as in [29]) that the MDP $\phi$ is communicating (that is, for every pair of states $(s, s')$, there exists a deterministic policy $\pi$ such that state $s'$ is accessible from state $s$ using $\pi$).

We denote by $Q^\pi(s, a) := r(s, a) + \gamma \mathbb{E}_{s' \sim P(\cdot|s,a)}[V^\pi(s')]$ the $Q$-function of $\pi$ in state $(s, a)$. We also define the sub-optimality gap of action $a$ in state $s$ to be $\Delta(s, a) := Q^\star(s, \pi^\star(s)) - Q^\star(s, a)$, where $Q^\star$ is the $Q$-function of $\pi^\star$, and let $\Delta_{\min} := \min_{s, a \neq \pi^\star(s)} \Delta(s, a)$ be the minimum gap in $\phi$. For some policy $\pi$, we define $\mathrm{Var}_{sa}[V^\pi] := \mathrm{Var}_{s' \sim P(\cdot|s,a)}[V^\pi(s')]$ to be the variance of the value function $V^\pi$ in the next state after taking action $a$ in state $s$. More generally, we define $M_{sa}^k[V^\pi] :=$

$\mathbb{E}_{s' \sim P(\cdot|s,a)}\left[\left(V^\pi(s') - \mathbb{E}_{\bar{s} \sim P(\cdot|s,a)}[V^\pi(\bar{s})]\right)^{2^k}\right]$ to be the $2^k$-th moment of the value function in the next state after taking action $a$ in state $s$. We also let $\mathrm{MD}_{sa}[V^\pi] := \|V^\pi - \mathbb{E}_{s' \sim P(\cdot|s,a)}[V^\pi]\|_\infty$ be the span of $\phi$ under $\pi$, *i.e.*, the maximum deviation from the mean of the next state value after taking action $a$ in state $s$.

**Best policy identification and sample complexity lower bounds.** The MDP $\phi$ is initially unknown, and we are interested in the scenario where the agent interacts sequentially with $\phi$. In each round $t \in \mathbb{N}$, the agent selects an action $a_t$ and observes the next state and the reward $(s_{t+1}, r_t)$: $s_{t+1} \sim P(\cdot|s_t, a_t)$ and $r_t \sim q(\cdot|s_t, a_t)$. The objective of the agent is to learn a policy in $\Pi_\varepsilon^\star(\phi)$ (possibly $\pi^\star$) as fast as possible. This objective is often formalized in a PAC framework where the learner has to stop interacting with the MDP when she can output an $\varepsilon$-optimal policy with probability at least $1 - \delta$. In this formalism, the learner strategy consists of (i) a sampling rule or exploration strategy; (ii) a stopping time $\tau$; (iii) an estimated optimal policy $\hat{\pi}$. The strategy is called $(\varepsilon, \delta)$-PAC if it stops almost surely, and $\mathbb{P}_\phi[\hat{\pi} \in \Pi_\varepsilon^\star(\phi)] \geq 1 - \delta$. Interestingly, one may derive instance-specific lower bounds of the sample complexity $\mathbb{E}_\phi[\tau]$ of any $(\varepsilon, \delta)$-PAC algorithm [28, 29], which involves computing an optimal allocation vector $\omega_{\mathrm{opt}} \in \Delta(S \times A)$ (where $\Delta(S \times A)$ is the set of distributions over $S \times A$) that specifies the proportion of times an agent needs to sample each pair $(s, a)$ to confidently identify the optimal policy:

$$\liminf_{\delta \to 0} \frac{\mathbb{E}_\phi[\tau]}{\mathrm{kl}(\delta, 1 - \delta)} \geq T_\varepsilon(\omega_{\mathrm{opt}}) \text{ where } T_\varepsilon(\omega)^{-1} := \inf_{\psi \in \mathrm{Alt}_\varepsilon(\phi)} \mathbb{E}_{(s,a) \sim \omega}[\mathrm{KL}_{\phi|\psi}(s,a)], \qquad (1)$$

and $\omega_{\mathrm{opt}} = \arg\inf_{\omega \in \Omega(\phi)} T_\varepsilon(\omega)^{-1}$. Here, $\mathrm{Alt}_\varepsilon(\phi)$ is the set of confusing MDPs $\psi$ such that the $\varepsilon$-optimal policies of $\phi$ are not $\varepsilon$-optimal in $\psi$, i.e., $\mathrm{Alt}_\varepsilon(\phi) := \{\psi : \phi \ll \psi, \Pi_\varepsilon^\star(\phi) \cap \Pi_\varepsilon^\star(\psi) = \emptyset\}$. In this definition, if the next state and reward distributions under $\psi$ are $P'(s, a)$ and $q'(s, a)$, we write $\phi \ll \psi$ if for all $(s, a)$ the distributions of the next state and of the rewards satisfy $P(s, a) \ll P'(s, a)$ and $q(s, a) \ll q'(s, a)$. We further let $\mathrm{KL}_{\phi|\psi}(s,a) := \mathrm{KL}(P(s,a), P'(s,a)) + \mathrm{KL}(q(s,a), q'(s,a))$. $\Omega(\phi)$ is the set of possible allocations; in the generative case it is $\Delta(S \times A)$, while with navigation constraints we have $\Omega(\phi) := \{\omega \in \Delta(S \times A) : \omega(s) = \sum_{s',a'} P(s|s', a')\omega(s', a')\}, \forall s \in S\}$, with $\omega(s) := \sum_a \omega(s, a)$. Finally, $\mathrm{kl}(a, b)$ is the KL-divergence between two Bernoulli distributions of means $a$ and $b$.

## 4 Towards Efficient Exploration Allocations

We aim to extend previous studies on best policy identification to online model-free exploration. In this section, we derive an approximation to the bound proposed in [28], involving quantities learnable via stochastic approximation, thereby enabling the use of model-free approaches.

The optimization problem (1) leading to instance-specific sample complexity lower bounds has an important interpretation [28, 29]. An allocation $\omega_{\mathrm{opt}}$ corresponds to an exploration strategy with minimal sample complexity. To devise an efficient exploration strategy, one could then think of estimating the MDP $\phi$, and solving (1) for this estimated MDP to get an approximation of $\omega_{\mathrm{opt}}$. There are two important challenges towards applying this approach:

(i) Estimating the model can be difficult, especially for MDPs with large state and action spaces, and arguably, a model-free method would be preferable.

(ii) The lower bound problem (1) is, in general, non-convex [28, 29].

A simple way to circumvent issue (ii) involves deriving an upper bound of the value of the sample complexity lower bound problem (1). Specifically, one may derive an upper bound $U(\omega)$ of $T_\varepsilon(\omega)$ by convexifying the corresponding optimization problem. The exploration strategy can then be the $\omega^\star$ that achieves the infimum of $U(\omega)$. This approach ensures that we identify an approximately optimal policy, at the cost of *over-exploring* at a rate corresponding to the gap $U(\omega^\star) - T_\varepsilon(\omega_{\mathrm{opt}})$. Note that using a lower bound of $T_\varepsilon(\omega)$ would not guarantee the identification of an optimal policy, since we would explore "less" than required. The aforementioned approach was already used in [28] where the authors derive an explicit upper bound $U_0(\omega)$ of $T_0(\omega)$. We also apply it, but derive an upper bound such that implementing the corresponding allocation $\omega^\star$ can be done in a model-free manner (hence solving the first issue (i)).

### 4.1 Upper bounds on $T_\varepsilon(\omega)$

The next theorem presents the upper bound derived in [28].

**Theorem 4.1** ([28]). *Consider a communicating MDP $\phi$ with a unique optimal policy $\pi^\star$. For all vectors $\omega \in \Delta(S \times A)$,*

$$T_0(\omega) \leq U_0(\omega) := \max_{(s,a):a \neq \pi^\star(s)} \frac{H_0(s,a)}{\omega(s,a)} + \max_s \frac{H_0^\star}{\omega(s,\pi^\star(s))}, \qquad (2)$$

*with*

$$\begin{cases} H_0(s,a) = \frac{2}{\Delta(s,a)^2} + \max\left( \frac{16\,\mathrm{Var}_{sa}[V^\star]}{\Delta(s,a)^2}, \frac{6\,\mathrm{MD}_{sa}[V^\star]^{4/3}}{\Delta(s,a)^{\frac{4}{3}}} \right), \\ H_0^\star = \frac{2}{\Delta_{\min}^2(1-\gamma)^2} + \min\left( \frac{27}{\Delta_{\min}^2(1-\gamma)^3}, \max\left( \frac{16\max_s \mathrm{Var}_{s\pi^\star(s)}[V^\star]}{\Delta_{\min}^2(1-\gamma)^2}, \frac{6\max_s \mathrm{MD}_{s\pi^\star(s)}[V^\star]^{4/3}}{\Delta_{\min}^{4/3}(1-\gamma)^{4/3}} \right) \right). \end{cases}$$

In the upper bound presented in this theorem, the following quantities characterize the *hardness* of learning the optimal policy: $\Delta(s,a)$ represents the difficulty of learning that in state $s$ action $a$ is sub-optimal; the variance $\mathrm{Var}_{sa}[V^\star]$ measures the aleatoric uncertainty in future state values; and the span $\mathrm{MD}_{sa}[V^\star]$ of the optimal value function can be seen as another measure of aleatoric uncertainty, large whenever there is a significant variability in the value for the possible next states.

Estimating the span $\mathrm{MD}_{sa}[V^\star]$, in an online setting, is a challenging task for large MDPs. Our objective is to derive an alternative upper bound that, in turn, can be approximated using quantities that can be learned in a model-free manner. We observe that the variance of the value function, and more generally its moments $M_{sa}^k[V^\star]^{2^{-k}}$ for $k \geq 1$ (see Appendix C), are smaller than the span. By refining the proof techniques used in [28], we derive the following alternative upper bound.

**Theorem 4.2.** *Let $\varepsilon \geq 0$ and let $k(s,a) := \arg\sup_{k \in \mathbb{N}} M_{sa}^k[V^\star]^{2^{-k}}$ (for brevity, we write $k$ instead of $k(s,a)$). Then, $\forall \omega \in \Delta(S \times A)$, we have $T_\varepsilon(\omega) \leq U(\omega)$, with*

$$U(\omega) := \max_{s,a \neq \pi^\star(s)} \left( \frac{2 + 8\varphi^2 M_{sa}^k[V^\star]^{2^{1-k}}}{\omega(s,a)\Delta(s,a)^2} + \max_{s'} \frac{C(s')(1+\gamma)^2}{\omega(s',\pi^\star(s'))\Delta(s,a)^2(1-\gamma)^2} \right), \qquad (3)$$

*where $C(s') = \max\left(4, 16\gamma^2\varphi^2 M_{s',\pi^\star(s')}^k[V^\star]^{2^{1-k}}\right)$ and $\varphi$ is the golden ratio.*

We can observe that in the worst case, the upper bound $U(\omega^\star)$ of the sample complexity lower bound, with $\omega^\star = \arg\inf_\omega U(\omega)$, scales as $O\left(\frac{|S||A|\max_s \mathrm{MD}_{s,\pi^\star(s)}[V^\star]^2}{\Delta_{\min}^2(1-\gamma)^2}\right)$. Since $\mathrm{MD}_{sa}[V^\star] \leq (1-\gamma)^{-1}$, then $U(\omega^\star)$ scales at most as $O\left(\frac{|S||A|}{\Delta_{\min}^2(1-\gamma)^4}\right)$. However, the following questions arise: (1) Can we select a single value of $k$ that provides a good approximation across all states and actions? (2) How much does this bound improve on that of Theorem 4.1? As we illustrate in the example presented in the next subsection, we believe that actually selecting $k = 1$ for all states and actions leads to sufficiently good results. With this choice, we obtain the following approximation:

$$U_1(\omega) := \max_{s,a \neq \pi^\star(s)} \left( \frac{2 + 8\varphi^2 \mathrm{Var}_{sa}[V^\star]}{\omega(s,a)\Delta(s,a)^2} + \max_{s'} \frac{C'(s')(1+\gamma)^2}{\omega(s',\pi^\star(s'))\Delta(s,a)^2(1-\gamma)^2} \right), \qquad (4)$$

where $C'(s') = \max\left(4, 16\gamma^2\varphi^2 \mathrm{Var}_{s',\pi^\star(s')}[V^\star]\right)$. $U_1(\omega)$ resembles the term in Theorem 4.1 (note that we do not know whether $U_1$ is a valid upper bound for $T_\varepsilon$). For the second question, our numerical experiments (presented below) suggest that $U(\omega)$ is a tighter upper bound than $U_0(\omega)$.

### 4.2 Example on Tabular MDPs

In Figure 1, we compare the characteristic time upper bounds obtained in the previous subsection. These upper bounds correspond to the allocations $\omega^\star$, $\omega_0^\star$, and $\omega_1^\star$ obtained by minimizing, over $\Delta(S \times A)^1$, $U(\omega)$, $U_0(\omega)$, and $U_1(\omega)$, respectively. We evaluated these characteristic times on various MDPs: (1) a random MDP (see Sec. A in the appendix); (2) the `RiverSwim` environment

---

[1]Results are similar when we account for the navigation constraints. We omit these results for simplicity.

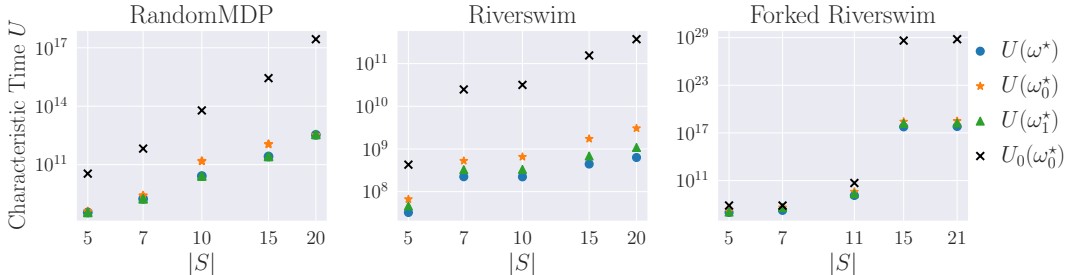

Figure 1: Comparison of the upper bounds (2) and (3) for different sizes of $S$ and $\gamma = 0.95$. We evaluated different allocations using $U_0(\omega)$ and $U(\omega)$. The allocations are: $\omega_0^\star$ (the optimal allocation in (2), $\omega^\star$ (the optimal allocation in (3) and $\omega_1^\star$ (the optimal allocation in (4) by setting $k = 1$ uniformly across states and actions). For the random MDP we show the median value across 30 runs.

[47]; (3) the `Forked RiverSwim`, a novel environment where the agent needs to constantly explore two different states to learn the optimal policy (compared to the `RiverSwim` environment, the sample complexity is higher; refer to Appendix A for a complete description).

We note that across all plots, the optimal allocation $\omega_0^\star$ has a quite large characteristic time (black cross). Instead, the optimal allocation $\omega^\star$ (blue circle) computed using our new upper bound (3) achieves a lower characteristic time. When we evaluate $\omega_0^\star$ on the new bound (3) (orange star), we observe similar characteristic times.

Finally, to verify that we can indeed choose $k = 1$ uniformly across states and actions, we evaluated the characteristic time $\omega_1^\star$ computed using (4) (green triangle). Our results indicate that the performance is not different from those obtained with $\omega^\star$, suggesting that the quantities of interest (gaps and variances) are enough to learn an efficient exploration allocation. We investigate the choice of $k$ in more detail in Appendix A.

## 5 Model-Free Active Exploration Algorithms

In this section we present MF-BPI, a model-free exploration algorithm that leverages the optimal allocations obtained through the previously derived upper bound of the sample complexity lower bound. We first present an upper bound $\tilde{U}(\omega)$ of $U(\omega)$, so that it is possible to derive a closed form solution of the optimal allocation (an idea previously proposed in [28]).

**Proposition 5.1.** *Assume that $\phi$ has a unique optimal policy $\pi^\star$. For all $\omega \in \Delta(S \times A)$, we have:*

$$U(\omega) \leq \tilde{U}(\omega) := \max_{s, a \neq \pi^\star(s)} \frac{H(s, a)}{\omega(s, a)} + \frac{H}{\min_{s'} \omega(s', \pi^\star(s'))},$$

*with $H(s, a) := \frac{2 + 8\varphi^2 M_{sa}^k [V^\star]^{2^{1-k}}}{\Delta(s, a)^2}$ and $H := \frac{\max_{s'} C(s')(1+\gamma)^2}{\Delta_{\min}^2 (1-\gamma)^2}$. The minimizer $\tilde{\omega}^\star := \arg\inf_\omega \tilde{U}(\omega)$ satisfies $\tilde{\omega}^\star(s, a) \propto H(s, a)$ for $a \neq \pi^\star(s)$ and $\tilde{\omega}^\star(s, \pi^\star(s)) \propto \sqrt{H \sum_{s, a \neq \pi^\star(s)} H(s, a) / |S|}$ otherwise.*

In the MF-BPI algorithm, we estimate the gaps $\Delta(s, a)$ and $M_{sa}^k [V^\star]$ for a fixed small value of $k$ (we later explain how to do this in a model-free manner.) and compute the corresponding allocation $\tilde{\omega}^\star$. This allocation drives the exploration under MF-BPI. Using this design approach, we face two issues:

**(1) Uniform $k$ and regularization.** It is impractical to estimate $M_{sa}^k [V^\star]$ for multiple values of $k$. Instead, we fix a small value of $k$ (*e.g.*, $k = 1$ or $k = 2$) for all state-action pairs (refer to the previous section for a discussion on this choice). Then, to avoid excessively small values of the gaps in the denominator, we regularize the allocation $\tilde{\omega}^\star$ by replacing, in the expression of $H(s, a)$ (resp. $H_{\min}$), $\Delta(s, a)$ (resp. $\Delta_{\min}$) by $(\Delta(s, a) + \lambda)$ (resp. $(\Delta_{\min} + \lambda)$) for some $\lambda > 0$.

**(2) Handling parametric uncertainty via bootstrapping.** The quantities $\Delta(s, a)$ and $M_{sa}^k [V^\star]$ required to compute $\tilde{\omega}^\star$ remain unknown during training, and we adopt the Certainty Equivalence principle, substituting the current estimates of these quantities to compute the exploration strategy.

---

**Algorithm 1** Boostrapped MF-BPI (Boostrapped Model Free Best Policy Identification)

---

**Require:** Parameters $(\lambda, k, p)$; ensemble size $B$; learning rates $\{(\alpha_t, \beta_t)\}_t$.

1: Initialize $Q_{1,b}(s,a) \sim \mathcal{U}([0, 1/(1-\gamma)])$ and $M_{1,b}(s,a) \sim \mathcal{U}([0, 1/(1-\gamma)^{2^k}])$ for all $(s,a) \in S \times A$ and $b \in [B]$.
2: **for** $t = 0, 1, 2, \ldots,$ **do**
3:     Bootstrap a sample $(\hat{Q}_t, \hat{M}_t)$ from the ensemble, and compute the allocation $\omega^{(t)}$ using Proposition 5.1. Sample $a_t \sim \omega^{(t)}(s_t, \cdot)$; observe $(r_t, s_{t+1}) \sim q(\cdot|s_t, a_t) \otimes P(\cdot|s_t, a_t)$.
4:     **for** $b = 1, \ldots, B$ **do**
5:         With probability $p$, using the experience $(s_t, a_t, r_t, s_{t+1})$, update $Q_{t,b}$ and $M_{t,b}$ using Equations (5) and (6).
6:     **end for**
7: **end for**

---

By doing so, we are inherently introducing parametric uncertainty into these terms that is not taken into account by the allocation $\tilde{\omega}^\star$. To deal with this uncertainty, the traditional method, as used e.g. in [28, 29]), involves using $\epsilon$-soft exploration policies to guarantee that all state-action pairs are visited infinitely often. This ensures that the estimation errors vanish as time grows large. In practice, we find this type of forced exploration inefficient. In MF-BPI, we opt for a bootstrapping approach to manage parametric uncertainties, which can augment the traditional forced exploration step, leading to more principled exploration.

### 5.1 Exploration in tabular MDPs.

The pseudo-code of MF-BPI for tabular MDPs is presented in Algorithm 1. In round $t$, MF-BPI explores the MDP using the allocation $\omega^{(t)}$ estimating $\tilde{\omega}^\star$. To compute this allocation, we use Proposition 5.1 and need (i) the sub-optimality gaps $\Delta(s,a)$, which can be easily derived from the $Q$-function; (ii) the $2^k$-th moment $M_{sa}^k[V^\star]$, which can always be learnt by means of stochastic approximation. In fact, for any Markovian policy $\pi$ and pair $(s,a)$ we have $M_{sa}^k[V_\phi^\pi] = \frac{1}{\gamma^{2^k}} \mathbb{E}_{s' \sim P(\cdot|s,a)}[\delta^\pi(s,a,s')^{2^k}]$, where $\delta^\pi(s,a,s') = r(s,a) + \gamma \mathbb{E}_{a' \sim \pi(\cdot|s')}[Q^\pi(s',a')] - Q^\pi(s,a)$ is a variant of the TD-error. MF-BPI then uses an asynchronous two-timescale stochastic approximation algorithm to learn $Q^\star$ and $M_{sa}^k[V^\star]$,

$$Q_{t+1}(s_t, a_t) = Q_t(s_t, a_t) + \alpha_t(s_t, a_t) \left( r_t + \gamma \max_a Q_t(s_{t+1}, a) - Q_t(s_t, a_t) \right), \qquad (5)$$

$$M_{t+1}(s_t, a_t) = M_t(s_t, a_t) + \beta_t(s_t, a_t) \left( (\delta_t'/\gamma)^{2^k} - M_t(s_t, a_t) \right), \qquad (6)$$

where $\delta_t' = r_t + \gamma \max_a Q_{t+1}(s_{t+1}, a) - Q_{t+1}(s_t, a_t)$, and $\{(\alpha_t, \beta_t)\}_{t \geq 0}$ are learning rates satisfying $\sum_{t \geq 0} \alpha_t(s,a) = \sum_{t \geq 0} \beta_t(s,a) = \infty$, $\sum_{t \geq 0} (\alpha_t(s,a)^2 + \beta_t(s,a)^2) \leq \infty$, and $\frac{\alpha_t(s,a)}{\beta_t(s,a)} \to 0$.

MF-BPI uses bootstrapping to handle parametric uncertainty. We maintain an ensemble of $(Q, M)$-values, with $B$ members, from which we sample $(\hat{Q}_t, \hat{M}_t)$ at time $t$. This sample is generated by sampling a uniform random variable $\xi \sim \mathcal{U}([0,1])$ and, for each $(s,a)$ set $\hat{Q}_t(s,a) = \text{Quantile}_\xi(Q_{t,1}(s,a), \ldots, Q_{t,B}(s,a))$ (assuming a linear interpolation). This method is akin to sampling from the parametric uncertainty distribution (we perform the same operation also to compute $\hat{M}_t$). This sample is used to compute the allocation $\omega^{(t)}$ using Proposition 5.1 by setting $\Delta_t(s,a) = \max_{a'} \hat{Q}_t(s,a') - \hat{Q}_t(s,a)$, $\pi_t^\star(s) = \arg\max_a \hat{Q}_t(s,a)$ and $\Delta_{\min,t} = \min_{s, a \neq \pi_t^\star(s)} \Delta_t(s,a)$. Note that, the allocation $\omega^{(t)}$ can be mixed with a uniform policy, to guarantee asymptotic convergence of the estimates. Upon observing an experience, with probability $p$, MF-BPI updates a member of the ensemble using this new experience. $p$ tunes the rate at which the models are updated, similar to sampling with replacement, speeding up the learning process. Selecting a high value for $p$ compromises the estimation of the parametric uncertainty, whereas choosing a low value may slow down the learning process.

**Exploration without bootstrapping?** To illustrate the need for our bootstrapping approach, we tried to use the allocation $\omega^{(t)}$ mixed with a uniform allocation. In Figure 2, we show the results on Riverswim-like environments with 5 states. While forced exploration ensures infinite visits to all

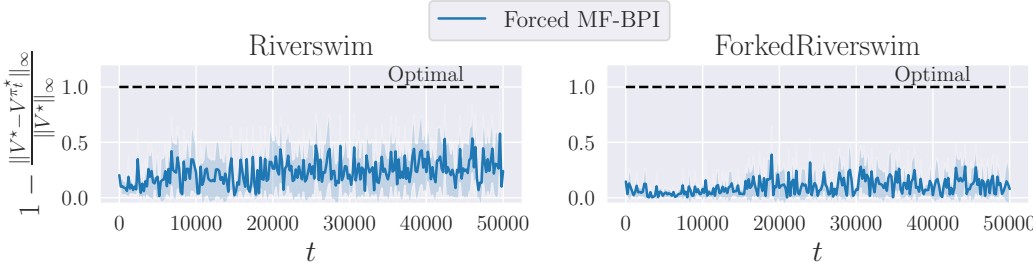

Figure 2: Forced exploration example with 5 states. We explore according to $\omega^{(t)}(s_t, a) = (1 - \epsilon_t) \frac{\tilde{\omega}_t^\star(s_t, a)}{\sum_{a'} \tilde{\omega}_t^\star(s_t, a')} + \epsilon_t \frac{1}{|A|}$, mixing the estimate of the allocation $\tilde{\omega}^\star$ from Proposition 5.1 with a uniform policy, with $\epsilon_t = \max(10^{-3}, 1/N_t(s_t))$ where $N_t(s)$ indicates the number of times the agent visited state $s$ up to time $t$. Shade indicates $95\%$ confidence interval.

state-action pairs, this guarantee only holds asymptotically. As a result, the allocation mainly focuses on the current MDP estimate, neglecting other plausible MDPs that could produce the same data. This makes the forced exploration approach too sluggish for effective convergence, suggesting its inadequacy for rapid policy learning. These results highlight the need to account for the uncertainty in $Q, M$ when computing the allocation.

## 5.2 Extension to Deep Reinforcement Learning

To extend bootstrapped MF-BPI to continuous MDPs, we propose DBMF-BPI (see Algorithm 2, or Appendix B). DBMF-BPI uses the mechanism of prior networks from BSP [39](bootstrapping with additive prior) to account for uncertainty that does not originate from the observed data. As before, we keep an ensemble $\{Q_{\theta_1}, \ldots, Q_{\theta_B}\}$ of $Q$-values (with their target networks) and an ensemble $\{M_{\tau_1}, \ldots, M_{\tau_B}\}$ of $M$-values, as well as their prior networks. We use the same procedure as in the tabular case to compute $(\hat{Q}_t, \hat{M}_t)$ at time $t$, except that we sample $\xi \sim \mathcal{U}([0, 1])$ every $T_s \propto (1-\gamma)^{-1}$ training steps (or at the end of an episode) to make the training procedure more stable. The quantity $\hat{Q}_t$ is used to compute $\pi_t^\star(s_t)$ and $\Delta_t(s_t, a)$. We estimate $\Delta_{\min,t}$ via stochastic approximation, with the minimum gap from the last batch of transitions sampled from the replay buffer serving as a target. To derive the exploration strategy, we compute $H_t(s_t, a) = \frac{2 + 8\varphi^2 \hat{M}_t(s_t, a)^{2^{1-k}}}{(\Delta_t(s_t, a) + \lambda)^2}$ and $H_t = \frac{4(1+\gamma)^2 \max(1, 4\gamma^2 \varphi^2 \hat{M}_t(s_t, \pi_t^\star(s_t))^{2^{1-k}})}{(\Delta_{\min,t} + \lambda)^2 (1-\gamma)^2}$. Next, we set the allocation $\omega_o^{(t)}$ as follows: $\omega_o^{(t)}(s_t, a) = H_t(s_t, a)$ if $a \neq \pi_t^\star(s_t)$ and $\omega_o^{(t)}(s_t, a) = \sqrt{H_t \sum_{a \neq \pi_t^\star(s_t)} H_t(s_t, a)}$ otherwise.

Finally, we obtain an $\epsilon_t$-soft exploration policy $\omega^{(t)}(s_t, \cdot)$ by mixing $\omega_o^{(t)}(s_t, \cdot)/\sum_a \omega_o^{(t)}(s_t, a)$ with a uniform distribution (using an exploration parameter $\epsilon_t$).

---

**Algorithm 2** DBMF-BPI (Deep Bootstrapped Model Free BPI)

---

**Require:** Parameters $(\lambda, k)$; ensemble size $B$; exploration rate $\{\epsilon_t\}_t$; estimate $\Delta_{\min,0}$; mask probability $p$.
1: Initialize replay buffer $\mathcal{D}$, networks $Q_{\theta_b}, M_{\tau_b}$ and targets $Q_{\theta_b'}$ for all $b \in [B]$.
2: **for** $t = 0, 1, 2, \ldots,$ **do**
3:     **Sampling step.**
4:         Compute allocation $\omega^{(t)} \leftarrow \texttt{ComputeAllocation}(s_t, \{Q_{\theta_b}, M_{\tau_b}\}_{b \in [B]}, \Delta_{\min,t}, \gamma, \lambda, k, \epsilon_t)$.
5:         Sample $a_t \sim \omega^{(t)}(s_t, \cdot)$ and observe $(r_t, s_{t+1}) \sim q(\cdot|s_t, a_t) \otimes P(\cdot|s_t, a_t)$.
6:         Add transition $z_t = (s_t, a_t, r_t, s_{t+1})$ to the replay buffer $\mathcal{D}$.
7:     **Training step.**
8:         Sample a batch $\mathcal{B}$ from $\mathcal{D}$, and with probability $p$ add the $i^{th}$ experience in $\mathcal{B}$ to a sub-batch $\mathcal{B}_b, \forall b \in [B]$. Update the $(Q, M)$-values of the $b^{th}$ member in the ensemble using $\mathcal{B}_b$: $\{Q_{\theta_b}, Q_{\theta_b'}, M_{\tau_b}\}_{b \in [B]} \leftarrow \texttt{Training}(\{\mathcal{B}_b, Q_{\theta_b}, Q_{\theta_b'}, M_{\tau_b}\}_{b \in [B]})$.
9:         Update estimate $\Delta_{\min,t+1} \leftarrow \texttt{EstimateMinimumGap}(\Delta_{\min,t}, \mathcal{B}, \{Q_{\theta_b}\}_{b \in [B]})$.
10: **end for**

---

# 6 Numerical Results

We evaluate the performance of MF-BPI on benchmark problems and compare it against state-of-the-art methods (details can be found in Appendix A).

**Tabular MDPs.** In the tabular case, we compared various algorithms on the `Riverswim` and `Forked Riverswim` environments. We evaluate MF-BPI with (1) bootstrapping and with (2) the forced exploration step using an $\epsilon$-soft exploration policy, MDP-NAS [29], PSRL [36] and Q-UCB [19, 56]. For MDP-NAS, the model of the MDP was initialized in an optimistic way (with additive smoothing).

In both environments, we varied the size of the state space. In Figure 3, we show $1 - \frac{\|V^\star - V^{\pi_T^\star}\|_\infty}{\|V^\star\|_\infty}$, a performance measure for the estimated policy $\pi_T^\star$ after $T = |S| \times 10^4$ steps with $\gamma = 0.99$. Results (the higher the better) indicate that bootstrapped MF-BPI can compete with model-based and model-free algorithms on hard-exploration problems, without resorting to expensive model-based procedures. Details of the experiments, including the initialization of the algorithms, are provided in Appendix A.

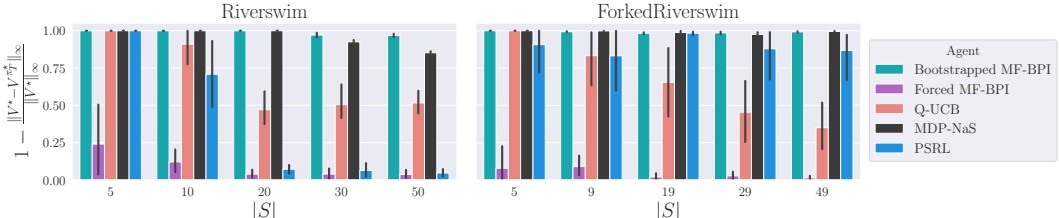

Figure 3: Evaluation of the estimated optimal policy $\pi_T^\star$ after $T$ steps for MF-BPI, Q-UCB, MDP-NAS and PSRL. Results are averaged across 10 seeds and lines indicate 95% confidence intervals.

**Deep RL.** In environments with continuous state space, we compared DBMF-BPI with BSP [40, 39] (Bootstrapped DQN with randomized priors) and IDS [33] (Information-Directed Sampling). We also evaluated DBMF-BPI against BSP2, a variant of BSP that uses the same masking mechanism as DBMF-BPI for updating the ensemble. These methods were tested on challenging exploration problems from the DeepMind behavior suite [41] with varying levels of difficulty: (1) a stochastic version of DeepSea and (2) the Cartpole swingup problem. The DeepSea problem includes a 5% probability of the agent slipping, i.e., that an incorrect action is executed, which increases the aleatoric variance.

The results for the Cartpole swingup problem are depicted in Figure 4 for various difficulty levels $k$ (see also Appendix A.5 for more details), demonstrating the ability of DBMF-BPI to quickly learn an efficient policy. While BSP generally performs well, there is a notable difference in performance when compared to DBMF-BPI. For a fair comparison, we used the same network initialization across all methods, except for IDS. Untuned, IDS performed poorly; proper initialization improved its performance, but results remained unsatisfactory. In Figure 5, we present two exploration metrics

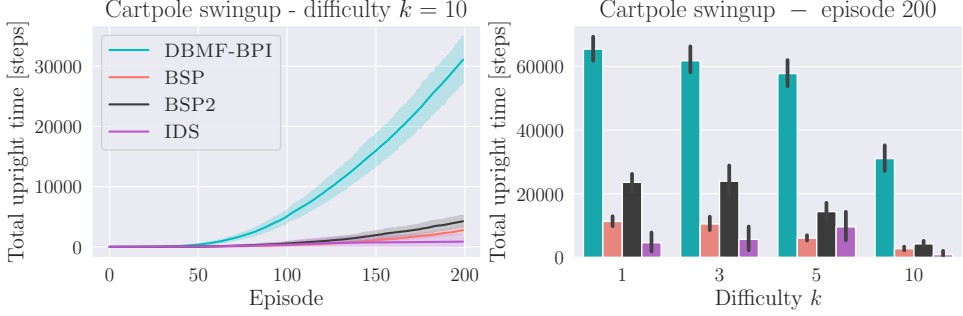

Figure 4: Cartpole swingup problem. On the left: total upright time at a difficulty level of $k = 10$. On the right: total upright time after 200 episodes for different difficulties $k$. To observe a positive reward, the pole's angle must satisfy $\cos(\theta) > k/20$, and the cart's position should satisfy $|x| \leq 1 - k/20$. Bars and shaded areas indicate 95% confidence intervals.

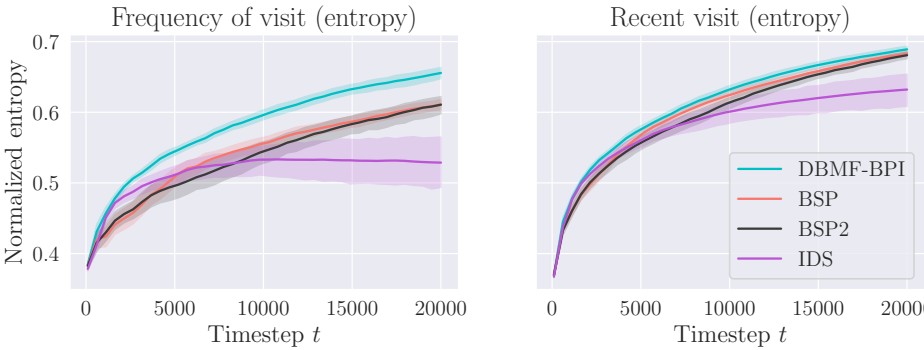

Figure 5: Exploration in Cartpole swingup for $k = 5$. On the left, we show the entropy of visitation frequency for the state space $(x, \dot{x}, \theta, \dot{\theta})$ during training. On the right, we show a measure of the dispersion of the most recent visits; smaller values indicate that the agent is less explorative as $t$ increases.

for difficulty $k = 5$. The frequency of visits measures the uniformity and dispersion of visits across the state space, while the second metric evaluates the recency of visits to different regions, capturing how frequently the methods keep visiting previously visited states (a smaller value indicates that the agent tends to concentrate on a specific region of the state space). For detailed analysis, please refer to appendix A.

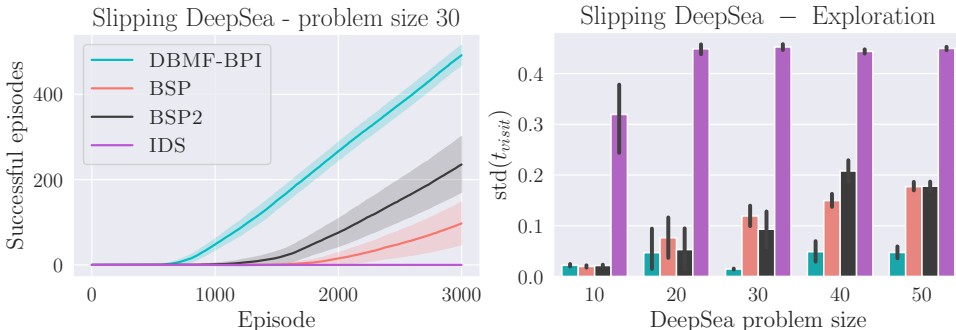

Figure 6: Slipping DeepSea problem. On the left: total number of successful episodes (*i.e.*, that the agent managed to reach the final reward) for a grid with $30^2$ input features. On the right: standard deviation of $t_{\text{visit}}$ at the last episode, depicting how much each agent explored (the lower the better).

For the slipping DeepSea problem, results are depicted in Fig. 6 (see also Appendix A.4 for more details). Besides the number of successful episodes, we also display the standard deviation of $(t_{\text{visit}})_{ij}$ across all cells $(i, j)$, where $(t_{\text{visit}})_{ij}$ indicates the last timestep $t$ that a cell $(i, j)$ was visited (normalized by $NT$, the product of the grid size, and the number of episodes). The right plot shows $\text{std}(t_{\text{visit}})$ for different problem sizes, highlighting the good exploration properties of DBMF-BPI. Additional details and exploration metrics can be found in Appendix A.

## 7 Conclusions

In this work, we studied the problem of exploration in Reinforcement Learning and presented MF-BPI, a model-free solution for both tabular and continuous state-space MDPs. To derive this method, we established a novel approximation of the instance-specific lower bound necessary for identifying nearly-optimal policies. Importantly, this approximation depends only on quantities learnable via stochastic approximation, paving the way towards model-free methods. Numerical results on hard-exploration problems highlighted the effectiveness of our approach for learning efficient policies over state-of-the-art methods.

## Acknowledgments

This research was supported by the Swedish Foundation for Strategic Research through the CLAS project (grant RIT17-0046) and partially supported by the Wallenberg AI, Autonomous Systems and Software Program (WASP) funded by the Knut and Alice Wallenberg Foundation. The authors would also like to thank the anonymous reviewers for their valuable and insightful feedback. On a personal note, Alessio Russo wishes to personally thank Damianos Tranos, Yassir Jedra, Daniele Foffano, and Letizia Orsini for their invaluable assistance in reviewing the manuscript.

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
