# Appendix

## Contents

**Appendix introduction**

We start by examining the wider impact of our work and acknowledging its limitations. This provides a balanced view of our contribution and points out areas for future research.

Next, we turn to the numerical results. Here, we give a more detailed account of our findings and include additional results for further clarity. We also introduce and describe the new Forked RiverSwim environment, an advanced version of the existing RiverSwim model, which has a larger sample complexity.

In the subsequent section, we break down the algorithms used in our study. This gives a deeper understanding of the methods underpinning our research.

We wrap up the appendix by providing all the proofs that support our conclusions.

**Broader impact**

This paper primarily focuses on foundational research in reinforcement learning, specifically the exploration problem, and proposes a novel model-free exploration strategy. While our work does not directly engage with societal impact considerations, we acknowledge the importance of considering the broader implications of AI technologies. As our proposed method improves the efficiency of reinforcement learning algorithms, it could potentially be applied in a wide range of contexts, some of which could have societal impacts. For instance, reinforcement learning is used in decision-making systems, which could include areas like healthcare, finance, and autonomous vehicles, where biases or errors could have significant consequences. Hence, while the direct societal impact of our work may not be immediately apparent, we strongly encourage future researchers and practitioners who apply these techniques to carefully consider the ethical implications and potential negative impacts in their specific use-cases. The responsible use of AI, including the mitigation of bias and the respect for privacy, should always be a priority.

**Limitations**

While our work presents significant advancements in the area of reinforcement learning, it also has its limitations that need to be acknowledged:

- **Assumptions**: Our approach relies on the assumption that the MDP is communicating. The instance-specific lower bound we propose may not be as effective if this assumption does not hold.

- **Scalability**: Our method, despite being model-free, still relies on stochastic approximations, which may not scale well with the complexity and size of certain MDPs.

- **Comparison with Model-Based Approaches**: While we have shown that our approach performs competitively with existing model-based exploration algorithms in hard-exploration environments, a comprehensive comparison across a wider range of environments is needed. It is possible that our method may not perform as well in some MDPs as the model-based approaches.

- **Bootstrapping**: Although bootstrapping has proven to be an effective technique, its usage is yet to be fully understood in RL applications. To achieve a more profound theoretical comprehension, a comprehensive analysis is necessary.

These limitations present opportunities for future research and the continued evolution of efficient exploration in reinforcement learning.

# A    Numerical Results

The appendix begins with the numerical results. We first introduce the Forked RiverSwim environment, a more complex variant of the traditional RiverSwim model.

Our discussion continues with a detailed exposition of Section 4.2, providing further experimental details. We conclude this section with additional findings related to both the tabular case and two specific problems: the CartPole Swing-Up and the Slipping DeepSea.

## A.1    The Forked Riverswim Environment

The `Forked RiverSwim`$(N)$ is a novel environment (see also Figure 7) where the agent needs to constantly explore two different states, $(s_g, s'_g)$, to learn the optimal policy. The number of states is $2N - 1$, and there are 3 actions.

The environment is similar to `RiverSwim`, but the initial state $s_1$ forks into two rivers: the final state in both branches of the rivers ($s_g$ and $s'_g$) have a similar high reward. Furthermore, the agent can deterministically switch between the two branches at any intermediate state. Intermediate states do not give any reward. Moreover, a little subtlety is that the agent can exploit the deterministic transition between $s_1$ and $s'_2$ to deterministically transition to $s_2$ (although this has a small effect as $N$ grows large).

Lastly, the Bernoulli rewards in $s_g$ and $s'_g$, which are the *highly rewarding* states, are quite similar (1 vs 0.95). Therefore, an optimal policy that starts in $s_1$ should achieve a slightly better reward than the optimal policy on the `RiverSwim` environment with $N + 1$ states (due to the fact that the transition to $s_2$ from $s_1$ can be made in a deterministic way).

Due to these reasons, this variant introduces additional complexity into the decision-making process. It is reasonable that a learning algorithm may learn an approximately good greedy policy in a short time-span, but not exactly the optimal one. In fact, we may expect an algorithm to take longer (compared to `Riverswim`) to learn the true optimal policy. Finally, always compared to `RiverSwim`, the sample complexity is of orders of magnitude higher, as also depicted in Figure 1. For a Python implementation, please refer to the GitHub repository of this manuscript.

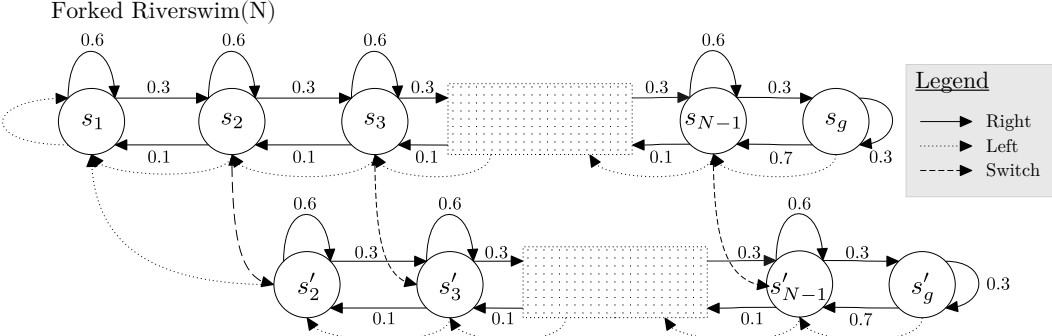

Figure 7: `Forked Riverswim`$(N)$ with $|S| = 2N - 1$ states. When taking action `left` in $s_1$ the agent observes a Bernoulli reward $r$ of parameter 0.05. When taking action `right` in $s_g$ (resp. $s'_g$) the agent observes a reward $r$ drawn from a Bernoulli of parameter 1 (resp. 0.95). In all other states the reward is 0. Action `left` and `switch` are deterministic, while the probability of action `Right` is indicated in the figure. The square boxes indicate that the pattern of states is being repeated from $s_3$ (or $s'_3$) until $s_{N-1}$ (or $s'_{N-1}$). This variant introduces additional complexity into the decision-making process, as the Bernoulli rewards in $s_g$ and $s'_g$ are quite similar (1 vs 0.95).

## A.2    Details of Example 4.2

In the following we report the details of Section 4.2. In Section 4.2 we evaluated the characteristic time of three different environments with same discount factor $\gamma = 0.95$:

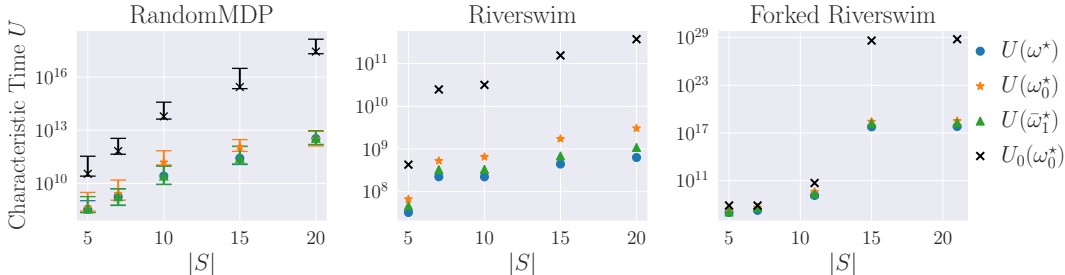

Figure 8: Comparison of (2) and (3) for discount $\gamma = 0.99$ and different sizes of the state space $S$. We evaluated different allocations using $U_0(\omega)$ and $U(\omega)$. The allocations are: $\omega^\star$ (the optimal allocation in Equation (3)), $\omega_0^\star$ (the optimal allocation in Equation (2)) and $\omega_1^\star$ (the optimal allocation that we get from (3) by setting $k = 1$ uniformly across states and actions). Results for the RandomMDP indicate the median and the bars 95% confidence intervals across 30 runs.

1. `RandomMDP`: an MDP with $|S|$ states and 3 actions. The transition probability for each $(s, a)$ is drawn from a Dirichlet distribution $\text{Dir}(\alpha_1, \ldots, \alpha_{|S|})$, with $\alpha_i = \alpha_{i-1} + (i - 1)/10$ and $\alpha_1 = 1$. The rewards also follow the same Dirichlet distribution, that is, for each $(s, a)$ we sample a $|S|$-dimensional vector $q$ of rewards from $\text{Dir}(\alpha_1, \ldots, \alpha_{|S|})$. This vector defines the rewards in the next state $r(s, a, s') = q_{s'}$, with $q \sim \text{Dir}(\alpha_1, \ldots, \alpha_{|S|})$. For this type of environment see also the details of the instance-specific quantities in Table 1.

2. `RiverSwim`: this environment is specified in [60], but we refer to the version used in [38] for a direct comparison. The reward is always 0 except in the initial state $s_1$, and the final state $s_{|S|}$. In the initial state we have $q(1|s_1, \text{left}) = 0.05$ (probability 0.05 of observing a reward of 1, and 0.95 probability of observing a reward of 0), while in the final state $q(1|s_{|S|}, \text{right}) = 1$. All other rewards are set to 0. Transition probabilities are the same as in [60]. For this type of environment see also the details of the instance-specific quantities in Table 2.

3. `Forked RiverSwim`: we refer the reader to Appendix A.1 for a description of this environment. For this type of environment see also the details of the instance-specific quantities in Table 3.

Interestingly, these environments have different properties that make them suitable for analysis: (1) the `RandomMDP` environment has very small gaps and variances; (2) the `Riverswim` environment has a relatively larger maximum span; (3) the `Forked Riverswim` environment, in contrast to the `Riverswim` environment, has a very small minimum gap $\Delta_{\min}$ and similar values for the span.

| $|S|$ | $\Delta_{\min}$ | $\max_{sa} \Delta_{sa}$ | $\min_{sa} \text{MD}_{sa}[V^\star]$ | $\max_{sa} \text{MD}_{sa}[V^\star]$ | $\min_{sa} \text{Var}_{sa}[V^\star]$ | $\max_{sa} \text{Var}_{sa}[V^\star]$ | $\max_{s,a,k} M_{sa}^k[V^\star]^{2^{-k}}$ |
|---|---|---|---|---|---|---|---|
| 5 | $1.1 \cdot 10^{-2}$ | $1.6 \cdot 10^{-1}$ | $6.4 \cdot 10^{-2}$ | $1.0 \cdot 10^{-1}$ | $8.3 \cdot 10^{-4}$ | $3.4 \cdot 10^{-3}$ | $1.0 \cdot 10^{-1}$ |
| 10 | $2.3 \cdot 10^{-3}$ | $6.3 \cdot 10^{-2}$ | $2.7 \cdot 10^{-2}$ | $3.6 \cdot 10^{-2}$ | $1.4 \cdot 10^{-4}$ | $3.7 \cdot 10^{-4}$ | $3.6 \cdot 10^{-2}$ |
| 25 | $1.2 \cdot 10^{-4}$ | $1.0 \cdot 10^{-2}$ | $4.6 \cdot 10^{-3}$ | $5.1 \cdot 10^{-3}$ | $3.3 \cdot 10^{-6}$ | $4.9 \cdot 10^{-6}$ | $5.1 \cdot 10^{-3}$ |
| 50 | $9.5 \cdot 10^{-6}$ | $1.9 \cdot 10^{-3}$ | $9.1 \cdot 10^{-4}$ | $9.5 \cdot 10^{-4}$ | $1.2 \cdot 10^{-7}$ | $1.4 \cdot 10^{-7}$ | $9.5 \cdot 10^{-4}$ |
| 100 | $1.1 \cdot 10^{-6}$ | $3.7 \cdot 10^{-4}$ | $1.8 \cdot 10^{-4}$ | $1.8 \cdot 10^{-4}$ | 0 | 0 | $1.8 \cdot 10^{-4}$ |

Table 1: Details of the instance-specific quantities for the `RandomMDP` environment (we evaluated up to $k = 19$). Results indicate an average over 300 different realizations. Confidence intervals are omitted for brevity, and values are rounded up to the 1st decimal.

| $\|S\|$ | $\Delta_{\min}$ | $\max_{sa} \Delta_{sa}$ | $\min_{sa} \mathrm{MD}_{sa}[V^\star]$ | $\max_{sa} \mathrm{MD}_{sa}[V^\star]$ | $\min_{sa} \mathrm{Var}_{sa}[V^\star]$ | $\max_{sa} \mathrm{Var}_{sa}[V^\star]$ | $\max_{s,a,k} M^k_{sa}[V^\star]^{2^{-k}}$ |
|---|---|---|---|---|---|---|---|
| 5 | $7.6 \cdot 10^{-2}$ | $1.3 \cdot 10^0$ | $1.7 \cdot 10^0$ | $3.0 \cdot 10^0$ | 0 | $3.6 \cdot 10^{-1}$ | $1.1 \cdot 10^0$ |
| 10 | $3.4 \cdot 10^{-2}$ | $1.3 \cdot 10^0$ | $2.5 \cdot 10^0$ | $4.5 \cdot 10^0$ | 0 | $3.7 \cdot 10^{-1}$ | $1.1 \cdot 10^0$ |
| 25 | $1.9 \cdot 10^{-2}$ | $1.3 \cdot 10^0$ | $2.5 \cdot 10^0$ | $5.0 \cdot 10^0$ | 0 | $3.7 \cdot 10^{-1}$ | $1.1 \cdot 10^0$ |
| 50 | $8.4 \cdot 10^{-3}$ | $1.3 \cdot 10^0$ | $2.7 \cdot 10^0$ | $5.4 \cdot 10^0$ | 0 | $3.7 \cdot 10^{-1}$ | $1.1 \cdot 10^0$ |
| 100 | $2.1 \cdot 10^{-4}$ | $1.3 \cdot 10^0$ | $2.9 \cdot 10^0$ | $5.5 \cdot 10^0$ | 0 | $3.7 \cdot 10^{-1}$ | $1.1 \cdot 10^0$ |

Table 2: Details of the instance-specific quantities for the `Riverswim` environment (we evaluated up to $k = 19$). Values are rounded up to the 1st decimal.

| $\|S\|$ | $\Delta_{\min}$ | $\max_{sa} \Delta_{sa}$ | $\min_{sa} \mathrm{MD}_{sa}[V^\star]$ | $\max_{sa} \mathrm{MD}_{sa}[V^\star]$ | $\min_{sa} \mathrm{Var}_{sa}[V^\star]$ | $\max_{sa} \mathrm{Var}_{sa}[V^\star]$ | $\max_{s,a,k} M^k_{sa}[V^\star]^{2^{-k}}$ |
|---|---|---|---|---|---|---|---|
| 5 | $1.0 \cdot 10^{-1}$ | $1.4 \cdot 10^0$ | $1.0 \cdot 10^0$ | $2.0 \cdot 10^0$ | 0 | $3.2 \cdot 10^{-1}$ | $1.0 \cdot 10^0$ |
| 11 | $2.8 \cdot 10^{-2}$ | $1.3 \cdot 10^0$ | $1.6 \cdot 10^0$ | $2.9 \cdot 10^0$ | 0 | $4.9 \cdot 10^{-1}$ | $2.0 \cdot 10^0$ |
| 25 | $1.0 \cdot 10^{-6}$ | $1.3 \cdot 10^0$ | $1.7 \cdot 10^0$ | $3.2 \cdot 10^0$ | 0 | $4.8 \cdot 10^{-1}$ | $2.0 \cdot 10^0$ |
| 51 | $1.0 \cdot 10^{-6}$ | $1.3 \cdot 10^0$ | $2.1 \cdot 10^0$ | $4.2 \cdot 10^0$ | 0 | $4.8 \cdot 10^{-1}$ | $2.0 \cdot 10^0$ |
| 101 | $1.0 \cdot 10^{-6}$ | $1.3 \cdot 10^0$ | $2.5 \cdot 10^0$ | $5.1 \cdot 10^0$ | 0 | $4.8 \cdot 10^{-1}$ | $2.0 \cdot 10^0$ |

Table 3: Details of the instance-specific quantities for the `Forked Riverswim` environment (we evaluated up to $k = 19$). Values are rounded up to the 1st decimal.

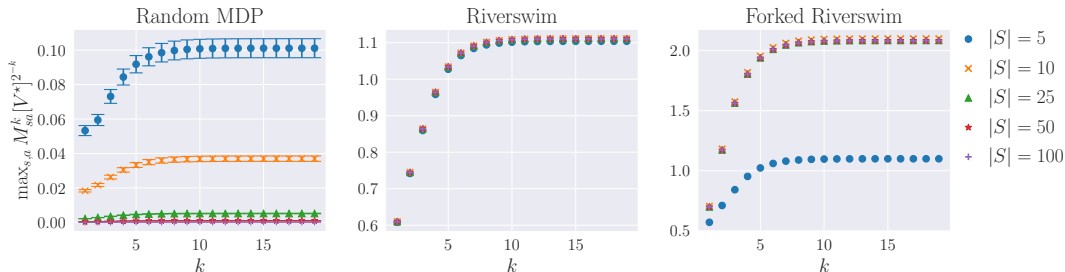

Figure 9: Plot of $\max_{s,a} M^k_{sa}[V^\star]^{2^{-k}}$ for various values of $k$. For the random MDP we depict the median value, as well as the 95% confidence interval.

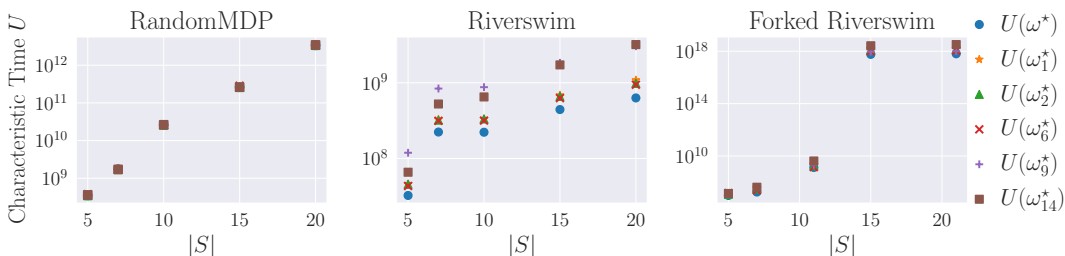

Figure 10: Evaluation of $\omega^\star_k$ for different values of $k$. For the `RandomMDP` we only show the median value over 300 runs.

Finally, in Figure 9, we depict $\max_{s,a} M^k_{sa}[V^\star]^{2^{-k}}$ for different values of $k$, up to $k = 19$. For the `RandomMDP` environment we observe that $\max_{s,a} M^k_{sa}[V^\star]^{2^{-k}}$ tends to the maximum of the span $\max_{sa} \mathrm{MD}_{sa}[V^\star]$, which depends on the size of the state space (as $\|S\|$ grows larger the span diminishes). For the other two environments, `Riverswim` and `Forked Riverswim`, $\max_{s,a} M^k_{sa}[V^\star]^{2^{-k}}$ does not seem to depend on the size of the state space. Furthermore, we also observe a sudden convergence of this quantity for relatively small values of $k$, followed by a relatively very slow increase.

In Figure 10 are shown the results when we evaluate the allocations $\omega^\star_k$ for different values of $k$. In general, we do not observe a striking difference between those allocations.

### A.3 Riverswim and Forked Riverswim - Description and Additional Results

In Figure 11 we present results from the Riverswim and ForkedRiverswim environments. These results include data from two new algorithms: O-BPI (Online Best Policy Identification) and PS-MDP-NAS (Posterior Sampling for MDP-NaS).

O-BPI is a novel algorithm that draws inspiration from MDP-NAS. However, a distinguishing characteristic is its use of stochastic approximation to determine the $Q$-values and $M$-values. These values, as for MF-BPI, are used to compute the allocation $\omega$ by solving the sample-complexity bound $\inf_{\omega \in \Omega(\phi)} U(\omega)$ with navigation constraints. On the other hand, PS-MDP-NAS is an adaptation of MDP-NAS that uses posterior sampling over the MDP's model to address the parametric uncertainty. It's worth noting that both these algorithms, O-BPI and PS-MDP-NAS, are model-based, and a detailed description of these algorithms is available in the following section, see **??** and Algorithm 3.

The results in Figure 11 clearly show the superiority of these allocation computing methods compared to other algorithms such as PSRL and Q-UCB.

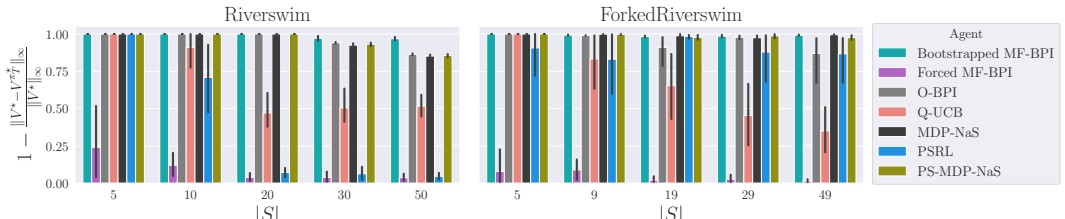

Figure 11: Evaluation of the estimated optimal policy $\pi_T^\star$ after $T$ steps for MF-BPI, O-BPI, Q-UCB, MDP-NAS, PS-MDP-NAS, and PSRL. Results are averaged across 10 seeds and lines indicate 95% confidence intervals. Note that for `Forked Riverswim` we have $N = 2|S| - 1$.

Figure 12 on the next page provides a visualization of the performance of each algorithm over the entire horizon $t = 0, \ldots, T - 1$. We exhibit the performance of the estimated greedy policy $\pi_t^\star$ at each timestep $t$ for each respective method. The results offer a clear demonstration of the efficiency of those methods based on the instance-specific sample complexity lower bound.

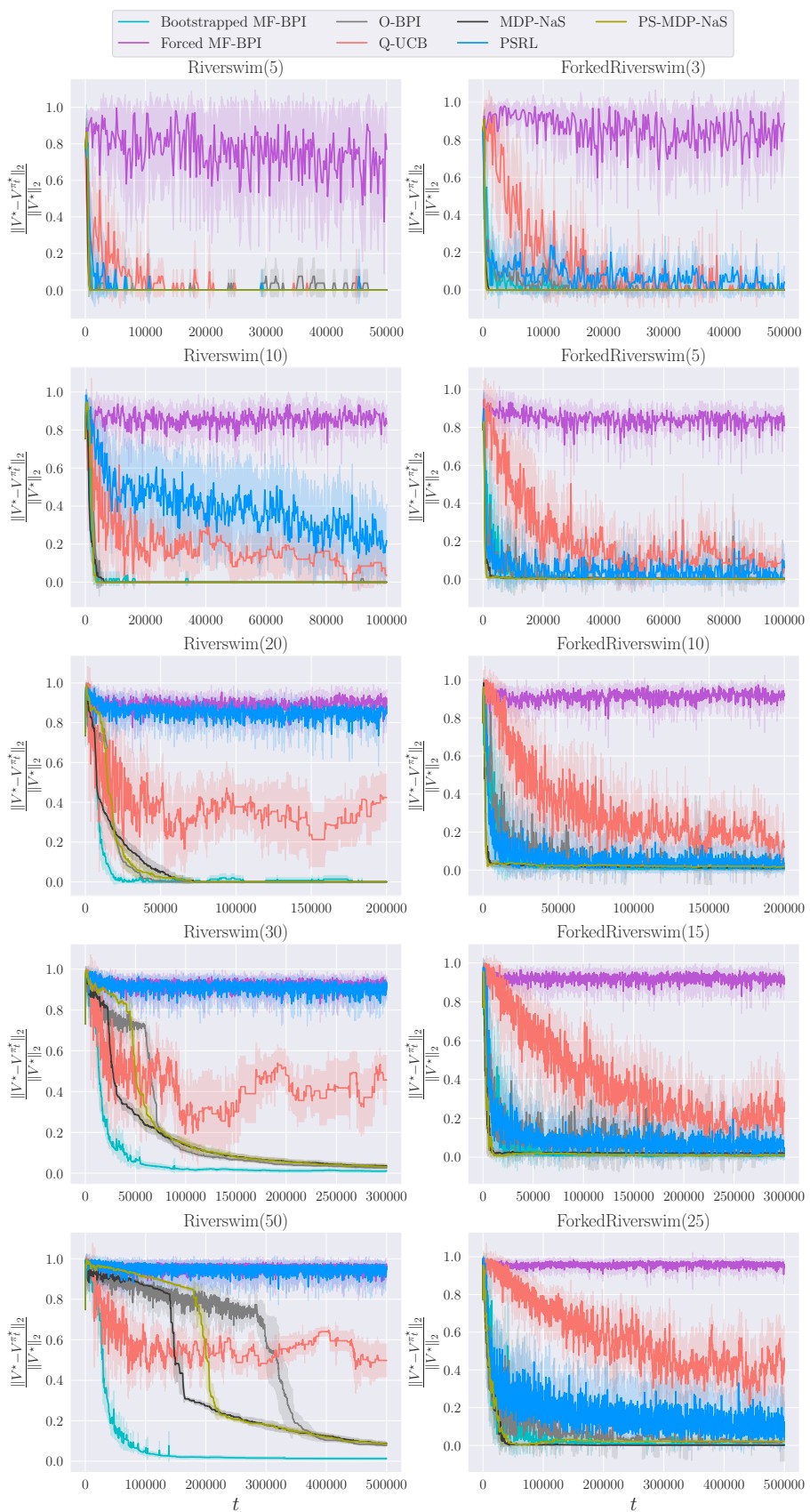

Figure 12: Evaluation of the estimated optimal policy $\pi_t^\star$ for MF-BPI, O-BPI, Q-UCB, MDP-NAS, PS-MDP-NAS, and PSRL. Results are averaged across 10 seeds and lines indicate 95% confidence intervals.

### A.4 Slipping DeepSea - Description and Additional Results

**Description.** The Slipping DeepSea problem is an hard-exploration reinforcement learning problem. In the standard version, there's an $N \times N$ grid, and the agent starts in the top left corner (state 0, 0) and needs to reach the bottom right corner (state $N - 1$, $N - 1$) for a large reward (the state vector is an $N^2$-dimensional vector, that one-hot encodes the agent's position in the grid). The agent can move diagonally, left or right (or down when close to the wall). The agent incurs in a cost when moving of $0.01/N$, while obtaining a positive reward of 1 when reaching the bottom right corner. Furthermore, we introduce the modification that there is a small probability of 0.05 that the incorrect action will be executed. This is a challenging problem because the optimal policy requires the agent to move (incurring a negative reward) many times before eventually reaching the high reward in the bottom right corner. However, due to the stochastic nature of the problem (the chance of slipping), the agent might be forced to take suboptimal actions, making it harder to learn the optimal policy.

**Additional results.** Figure 13 presents additional metrics encapsulating the exploration conducted by each algorithm, offering a comprehensive summary of the exploration process after $T$ episodes for each size $N$ (note that for a given size $N$ the number of input features in the state is $N^2$).

We focus on two key metrics: (a) $(t_{\text{visit}})_{ij}$ and (b) $(t_{\text{avg}})_{ij}$. Here, (a) $(t_{\text{visit}})_{ij}$ represents the last timestep $t$ at which a cell $(i, j)$ was visited (this value is normalized by $NT$, the multiplication of the grid size and the number of episodes), while (b) $(t\text{avg})_{ij}$ signifies the average frequency with which a cell $(i, j)$ was visited. In terms of arrangement, from the top downwards: (1) we present

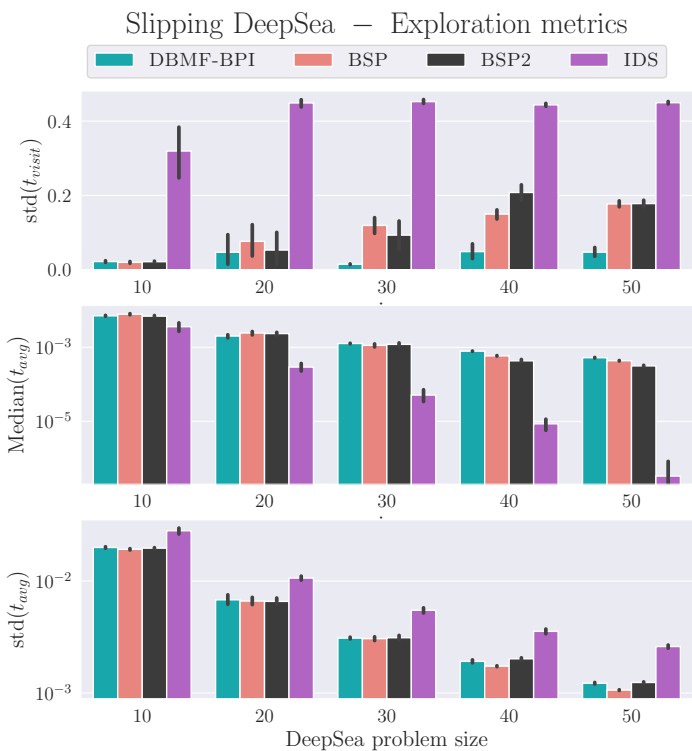

Figure 13: Slipping DeepSea problem - exploration metrics. From top to bottom: (1)standard deviation of $t_{\text{visit}}$ at the last episode, depicting how much each agent explored (the lower the better); (2) median value of $(t_{\text{avg}})_{ij}$, *i.e.*, the median value of a cell's visit frequency; (3) standard deviation of $(t_{\text{avg}})_{ij}$ across all cells. Results are averaged over 24 runs and bars indicate 95% confidence intervals.

the standard deviation of $(t_{\text{visit}})_{ij}$ across all cells; (2) we show the median value of a cell's visit frequency; (3) we depict the standard deviation of $(t_{\text{avg}})_{ij}$ across all cells. From the central plot, we notice that DBMF-BPI tends to visit all cells slightly more frequently. The first plot also highlights that DBMF-BPI maintains a consistent visit rate to all cells. This pattern is a strong indication of DBMF-BPI's explorative behavior. Conversely, neither BSP nor BSP2 match this performance in

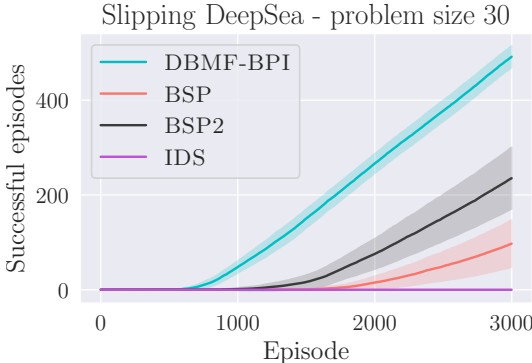

Figure 14: Slipping DeepSea problem. Total number of successful episodes (*i.e.*, that the agent managed to reach the final reward) for a grid with $30^2$ input features.

terms of successful episodes (shown in Figure 14), despite the median value of $t_{\mathrm{avg}}$ being very similar to that of DBMF-BPI. In order to provide a more comprehensive view, Figure 15 and Figure 16 present additional exploration metrics. Specifically, we display $(t_{\mathrm{avg}})_{ij}$ and $(t_{\mathrm{visit}})_{ij}$, respectively, after $T = 3000$ episodes, given a DeepSea problem size of 30. The initial plot illustrates how DBMF-BPI tends to concentrate on the grid's diagonal. However, the bottom plot shows that, in spite of this diagonal focus, DBMF-BPI also maintains a consistent exploration of other cells within the grid. We also observe how BSP seems to uniformly explore all cells, while IDS does not manage to explore the entire grid within the number of episodes. Last, but not least, on the right column in

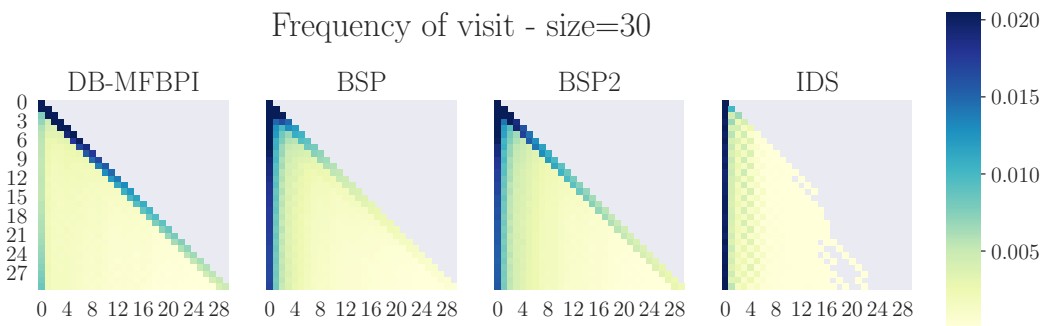

Figure 15: Slipping DeepSea problem. In this figure we depict the average frequency of visits, after 3000 episodes, when the size of the problem is $k = 30$.

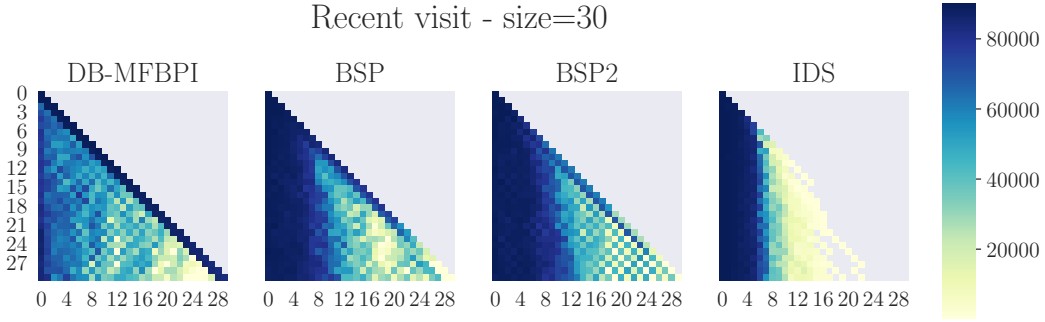

Figure 16: Slipping DeepSea problem. In this figure we depict the last timestep a cell was visited, after 3000 episodes, when the size of the problem is $k = 30$.

Figure 17, are shown the results for the learnt greedy policy $\pi_t^\star$ at time $t$. Clearly, DBMF-BPI is able to learn an efficient policy more quickly than the other methods for different problem sizes.

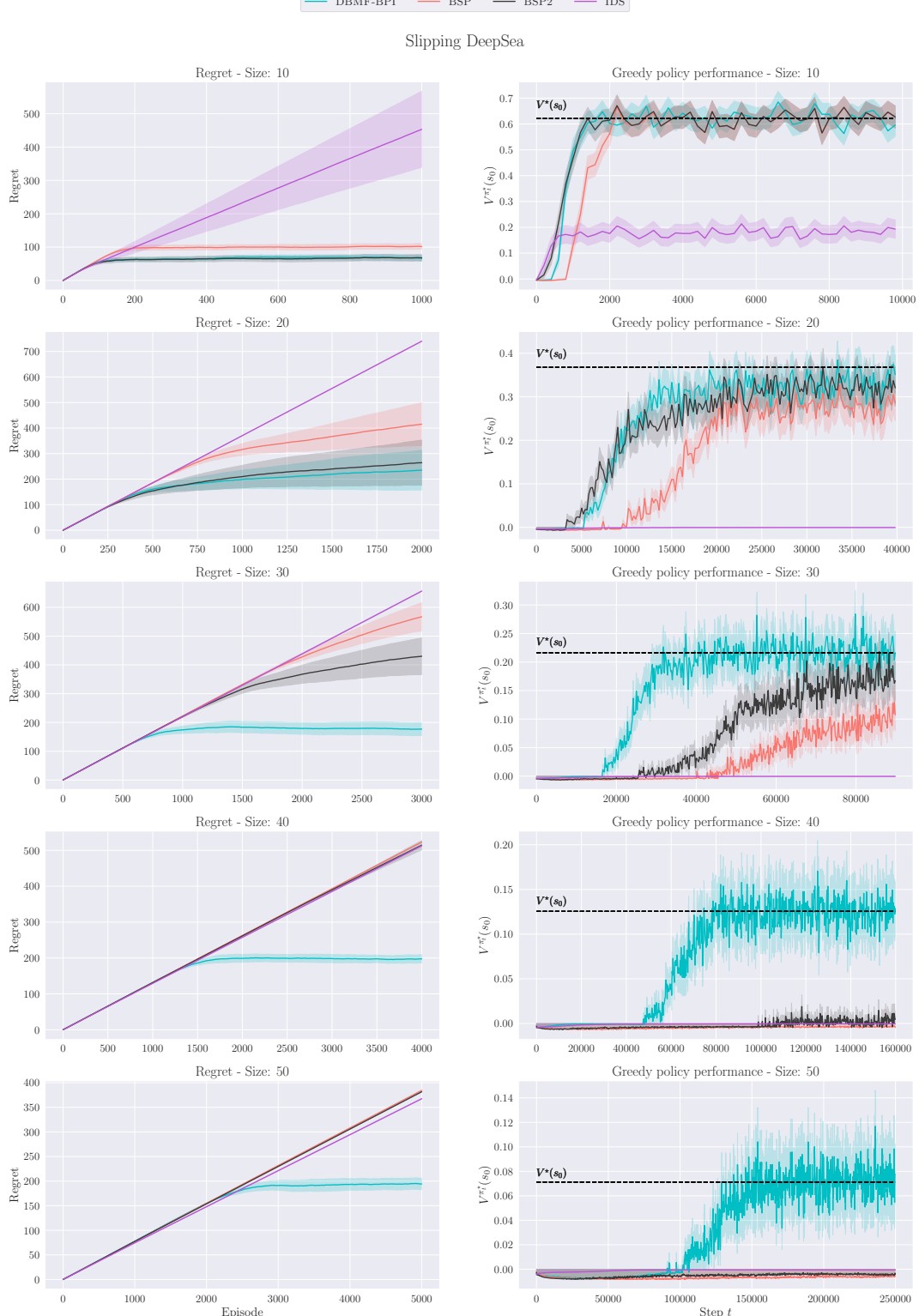

Figure 17: Slipping DeepSea problem - evaluation of the greedy policy. On the left we depict the regret of the learning agent over the number of episodes $T$ for each problem size $k$. On the right, we display the average value of the learnt greedy policy $\pi_t^\star$ at time $t$ (black dashed-line indicates the average optimal value). Results are averaged over 24 runs, and the shaded area depicts 95% confidence intervals.

## A.5 Cartpole Swingup - Description and Additional Results

**Description.** In this subsection we present additional results for the Cartpole swingup problem. The cartpole swingup problem is a classic problem in control theory and reinforcement learning [9]. The task is to balance a pole that is attached by an un-actuated joint to a cart, which moves along a frictionless track. The system is controlled by applying a force to the cart. Initially, the pole is hanging down and the goal is to swing it up so it stays upright. In contrast to the classic cartpole balance problem, the pole needs not only to be balanced when it's upright but also to be swung up to the upright position.

The state of the system at any point in time is described by four variables: the position of the cart $x$, the velocity of the cart $\dot{x}$, the angle of the pole $\theta$, and the angular velocity of the pole $\dot{\theta}$. There are 4 additional variables in the state, and for simplicity we refer the user to [52].

To make the problem more difficult, as in [52] we introduce a parameter $k \in \{1, \ldots, 19\}$ (to not be confused with the parameter of $M_{sa}^{k}[V^{\star}]$) that parameterizes the reward function. Specifically, the agent observes a positive reward of 1 only if the pole's angle satisfies $\cos(\theta) > k/20$, and the cart's position satisfies $|x| \leq 1 - k/20$. There is also a negative reward of $-0.1$ that the agent incurs for moving, which aggravates the explore-exploit tradeoff (algorithms like DQN [41] simply remain still).

**Additional results.** In Figures 18 to 20, we provide supplementary results for this problem. Figure 18 illustrates the total upright time achieved by each learner after 200 episodes, across various difficulty levels, $k$. Here, the total upright time refers to the total count of steps where the pole maintained an angle satisfying $\cos(\theta) > k/20$, concurrently with the cart maintaining a position that satisfied $|x| \leq 1 - k/20$.

Subsequently, Figure 19 showcases the evolution of this metric throughout all 200 episodes.

Figure 20 demonstrates the performance of the learnt greedy policy $\pi_t^{\star}$ over the course of the training. Every 10 episodes, we evaluated the greedy policy over 20 episodes and computed the cumulative reward.

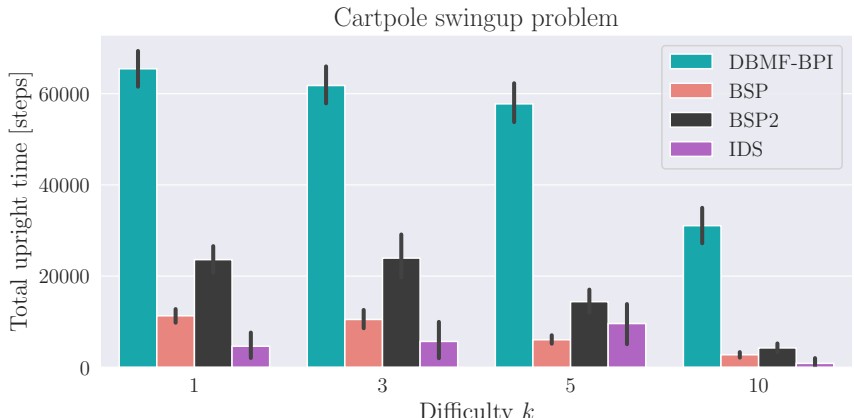

Figure 18: Cartpole swingup problem. Total upright time after 200 episodes for different difficulties $k$. To observe a positive reward, the pole's angle must satisfy $\cos(\theta) > k/20$, and the cart's position should satisfy $|x| \leq 1 - k/20$. Bars indicate $95\%$ confidence intervals.

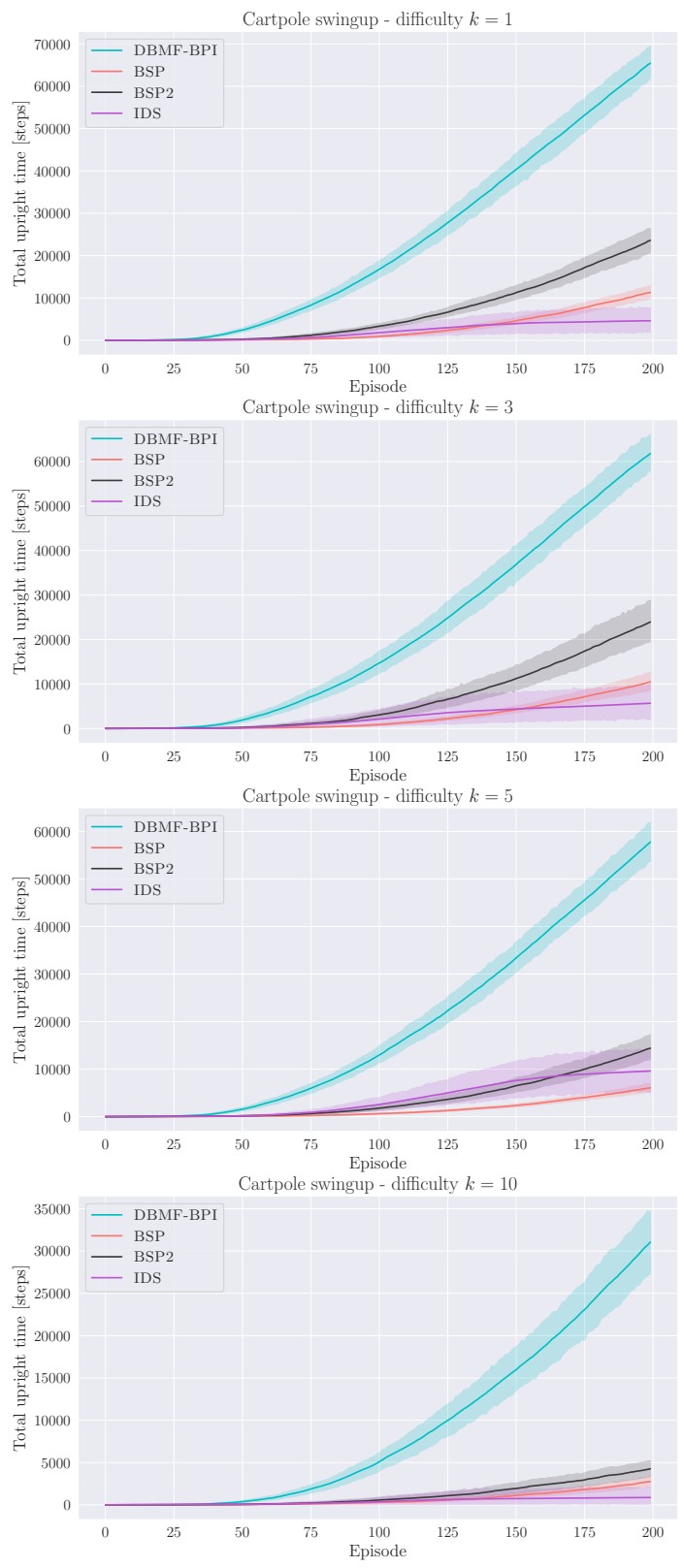

Figure 19: Cartpole swingup problem. Total upright time over 200 episodes for different difficulties $k$. To observe a positive reward, the pole's angle must satisfy $\cos(\theta) > k/20$, and the cart's position should satisfy $|x| \leq 1 - k/20$. Bars indicate $95\%$ confidence intervals.

Figure 20: Cartpole swingup problem. Performance of the learnt greedy policy $\pi_t^\star$ over the training episodes (average cumulative reward collected by the greedy policy).

**Exploration results.** In Figures 21 to 23, we show additional results that illustrate the exploration of the various algorithms for difficulties $k = 3, 5$.

In Figure 21, we display two metrics at each training step $t$: the entropy of visit frequency and the entropy of the most recent visit. The first metric quantifies how thoroughly the method has explored the state space $(x, \dot{x}, \theta, \dot{\theta})$ up to time $t$. To do this, we discretize the state space into bins and tally the occurrences in each bin. We then normalize these counts by their sum and calculate the resulting entropy, which is normalized to the range $[0, 1]$.

While this measure of visit frequency provides some insight, it is insufficient for understanding whether the algorithm continues to explore new states or revisits old ones. To address this, the second metric measures the dispersion of the timing of the last visits to various regions of the state space. A larger dispersion indicates that the algorithm is concentrating on a specific region, resulting in a smaller entropy (and vice-versa). To calculate this, we again use normalized entropy.

Finally, in Figure 22 and Figure 23, we illustrate the visitation frequency after $20K$ training steps for $(x, \dot{x})$ and $(\dot{x}, \dot{\theta})$ at difficulty levels $k = 3$ and $k = 5$. Darker regions signify higher visitation frequencies. The pattern in $(\dot{x}, \dot{\theta})$ is characteristic of algorithms that have learned to stabilize the policy. Notably, DBMF-BPI is also actively exploring various velocities. A similar trend is observed for $(x, \dot{x})$: while most methods focus on an $s$-shaped trajectory, DBMF-BPI also explores other regions of the state space.

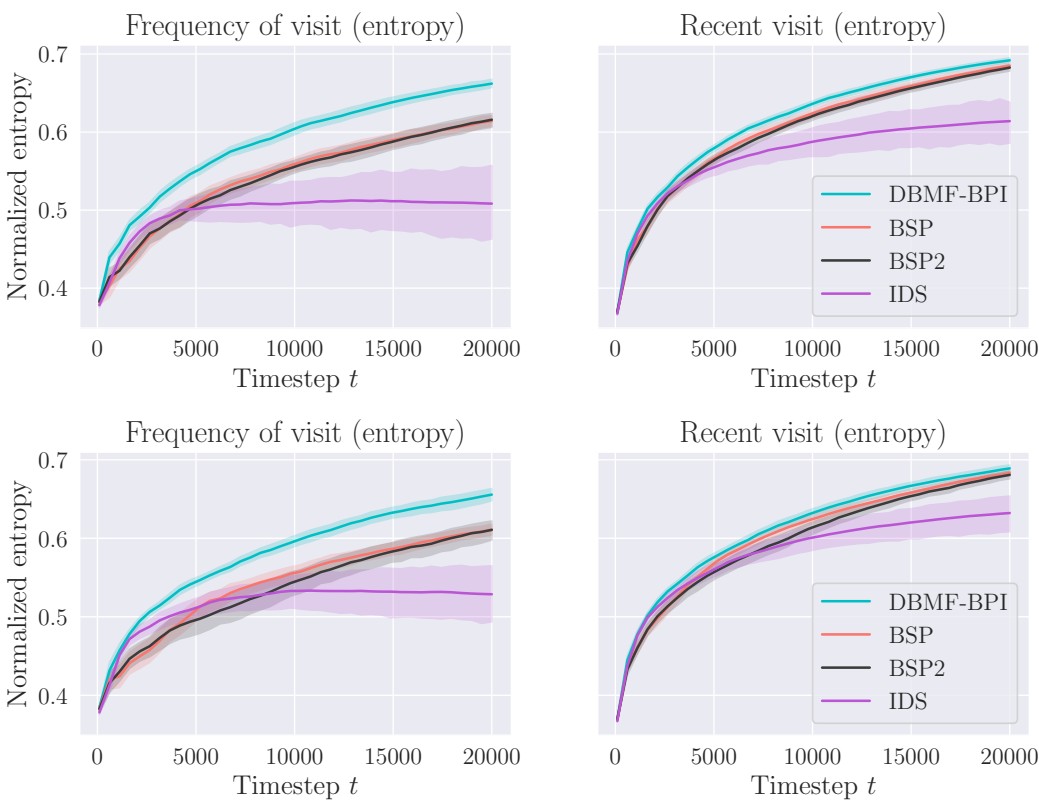

Figure 21: Exploration in Cartpole swingup: At the top, we present results for difficulty $k = 3$, and at the bottom, for $k = 5$. In the left column, we depict the entropy of visitation frequency for the state space $(x, \dot{x}, \theta, \dot{\theta})$ during training. In the right column, we display a measure of the dispersion of the most recent visits; smaller values indicate that the agent is less explorative as $t$ increases.

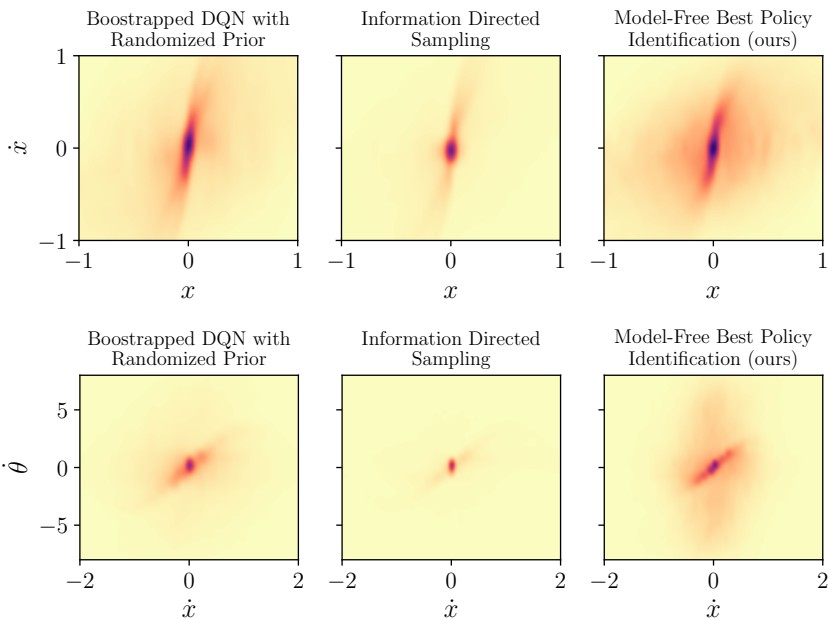

Figure 22: Cartpole Swingup [52] after 20K training steps for difficulty $k = 3$, comparing BSP (Bootstrapped DQN with randomized priors) [50], IDS (Information-Directed Sampling) [44], and MF-BPI (Model-Free Best Policy Identification). Darker areas indicate higher visitation frequency. At the top we show this frequency for $(x, \dot{x})$, the cart's position and linear's velocity, and at the bottom of $(\dot{x}, \dot{\theta})$, the cart's linear and pole's angular velocities.

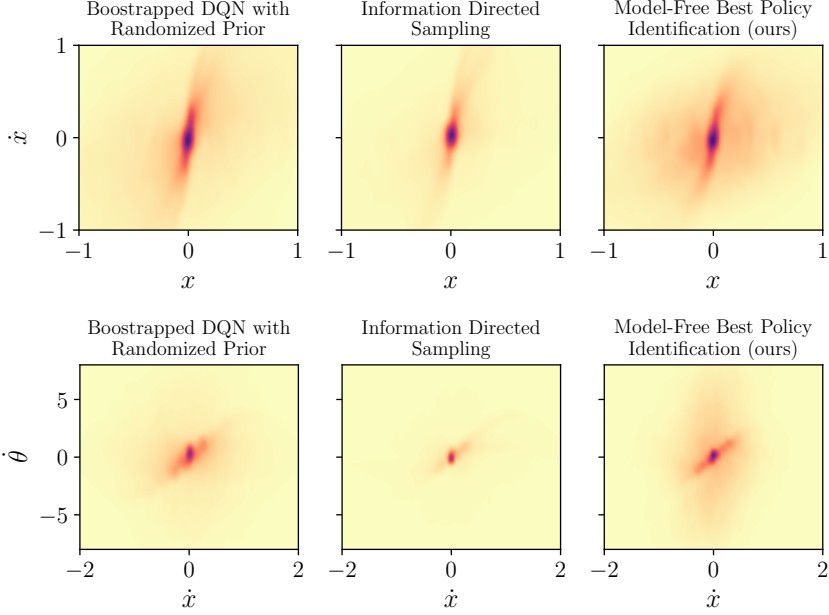

Figure 23: Cartpole Swingup [52] after 20K training steps for difficulty $k = 5$, comparing BSP (Bootstrapped DQN with randomized priors) [50], IDS (Information-Directed Sampling) [44], and MF-BPI (Model-Free Best Policy Identification). Darker areas indicate higher visitation frequency. At the top we show this frequency for $(x, \dot{x})$, the cart's position and linear's velocity, and at the bottom of $(\dot{x}, \dot{\theta})$, the cart's linear and pole's angular velocities.

## A.6 Parameters, Hardware, Code and Libraries

In this section, we outline the parameters used for the simulations, describe the hardware employed to run the simulations, and list the libraries that we used.

### A.6.1 Simulation parameters - Riverswim and Forked Riverswim

In both the Riverswim and Forked Riverswim environments we used a discount factor of $\gamma = 0.99$. Depending on the size of the state space, the horizon length was different. We used $T = 10000 \times |S|$ for the Riverswim environment, and $T = 20000 \times N$ for the Forked Riverswim environment (where $N$ is the length of the main river; see also the description of the environment in Appendix A.1).

We run simulations for 10 different seeds, and evaluated the estimated greedy policy $\pi_t^\star$ every 200 steps. All agents were optimistically initialized (*i.e.*, the $Q$-values were initialized to $1/(1-\gamma)$, etc...), and model-based approaches used additive smoothing (with factor 1).

For the MDP-NAS and PS-MDP-NAS (see next section for a description) we computed the allocation every $T_0 = \min(T_{max}, \max(200, \frac{T_{max}t}{T/2}))$ steps, where $T_{max} = \frac{2000T}{50000}$. For PSRL we computed a new greedy policy every $\lceil 1/(1-\gamma) \rceil$ steps.

We used a learning rate of $\alpha_t = \frac{H+1}{H+k_t}$ to learn the $Q$-values, where $H = (1-\gamma)^{-1}$ and $k_t = N_t(s_t, a_t)$ is the number of visits to $(s_t, a_t)$ at time $t$. Similarly, to learn the $M$-values we used a learning rate of $\beta_t = \alpha_t^{1.1}$ (which was not optimized).

For bootstrapped MFBPI we used a parameter $k = 1$, and an ensemble size $B = 50$ with training probability $p = 0.7$. Similarly, these values were not optimized.

All methods that employed an $\epsilon$-soft policy, obtained a final policy $\omega$ by mixing mixed the original policy $\pi$ with a uniform policy as follows $\omega(a|s) = (1 - \epsilon_t)\pi(a|s) + \epsilon_t/|A|$. The value of $\epsilon_t$ is $\epsilon_t = \min(1, 1/N_t(s_t))$ where $N_t(s_t)$ is the total number of visits to state $s_t$ at time $t$.

### A.6.2 Simulation parameters - Slipping DeepSea

For the DeepSea problem we used a discount factor of $\gamma = 0.99$, and different problem sizes $N \in \{10, 20, 30, 40, 50\}$. The number of training episodes was $T = 100N$. Every 200 steps we evaluated the performance of the estimated greedy policy $\pi_t^\star$ over 20 episodes. For all simulations we used a slipping probability of $0.05$. The number of features in the state is $N^2$, and the number of actions is 2.

Refer to Table 4 for the parameters of the agents.

Table 4: Parameters of the agents for the slipping DeepSea problem.

| Property | DBMF-BPI | BSP | BSP 2 | IDS |
|---|---|---|---|---|
| Ensemble size $Q$ | 20 | 20 | 20 | $20 + \frac{(N-10)}{2}$ |
| Ensemble size $M$ | 20 | N.A. | N.A. | N.A. |
| Hidden layers sizes | [32] | [32] | [32] | [50] |
| Num. of quantiles | N.A. | N.A. | N.A. | 50 |
| Prior scale $Q$-values (depends on $N$) | $\{3, 5, 10, 15, 20\}$ | $\{3, 5, 10, 15, 20\}$ | $\{3, 5, 10, 15, 20\}$ | N.A. |
| Prior scale $M$-values (depends on $N$) | $\{3, 5, 10, 15, 20\}$ | $\{3, 5, 10, 15, 20\}$ | $\{3, 5, 10, 15, 20\}$ | N.A. |
| Replay buffer size | $10^5$ | $10^5$ | $10^5$ | $10^5$ |
| Training period | 1 | 1 | 1 | 1 |
| Target network update period | 4 | 4 | 4 | 4 |
| Batch size | 128 | 128 | 128 | 128 |
| Mask probability $p$ | 0.7 | 0.5 | 0.7 | N.A. |
| Learning rate $Q$-values | $5 \times 10^{-4}$ | $10^{-3}$ | $10^{-3}$ | $5 \times 10^{-4}$ |
| Learning rate $M$-values | $5 \times 10^{-4}$ | N.A. | N.A. | N.A. |
| Learning rate quantile network | N.A. | N.A. | N.A. | $10^{-6}$ |
| $k$ | 2 | N.A. | N.A. | N.A. |

### A.6.3 Simulation parameters - Cartpole Swingup

For the Cartpole swingup problem we used a discount factor of $\gamma = 0.99$, and different difficulties $k \in \{1, 3, 5, 10\}$. The number of training episodes was $T = 200$, and we run simulations for 20 different seeds. Every 10 steps in the training we evaluated the performance of the estimated greedy policy $\pi_t^\star$ over 20 episodes. The state is a vector in $\mathbb{R}^8$ and the number of actions is 3.

For every method, with the exception of IDS, we set up the parameters in the $i^{th}$ layer of each network by sampling from a truncated Gaussian distribution with a 0 mean and a standard deviation of $1/\sqrt{f_{in}}$, where $f_{in}$ represents the number of inputs to the $i^{th}$ layer. Values were cut off at twice the standard deviation. For IDS, enhancing the standard deviation improved results. Specifically, we employed a standard deviation of $1.5/\sqrt{f_{in}}$ for the $Q$-networks ensemble, and a standard deviation of $2/\sqrt{f_{in}}$ for the quantile network. Generally, this initialization mirrors an optimistic initialization. However, the results can vary significantly between runs, and our observation was that the IDS method often exhibited greater variance compared to the other methods incorporated in our study. To conclude, the bias for all layers was set to 0.

Refer to Table 5 for the parameters of the agents.

Table 5: Parameters of the agents for the Cartpole swingup problem.

| Property | DBMF-BPI | BSP | BSP 2 | IDS |
|---|---|---|---|---|
| Ensemble size $Q$ | 20 | 20 | 20 | 20 |
| Ensemble size $M$ | 20 | N.A. | N.A. | N.A. |
| Hidden layers sizes | [50] | [50] | [50] | [50] |
| Num. of quantiles | N.A. | N.A. | N.A. | 50 |
| Prior scale $Q$-values (depends on $N$) | 3 | 3 | 3 | N.A. |
| Prior scale $M$-values (depends on $N$) | 3 | 3 | 3 | N.A. |
| Replay buffer size | $10^5$ | $10^5$ | $10^5$ | $10^5$ |
| Training period | 1 | 1 | 1 | 1 |
| Target network update period | 4 | 4 | 4 | 4 |
| Batch size | 128 | 128 | 128 | 128 |
| Mask probability $p$ | 0.7 | 0.5 | 0.7 | N.A. |
| Learning rate $Q$-values | $5 \times 10^{-4}$ | $5 \times 10^{-4}$ | $5 \times 10^{-4}$ | $5 \times 10^{-4}$ |
| Learning rate $M$-values | $5 \times 10^{-4}$ | N.A. | N.A. | N.A. |
| Learning rate quantile network | N.A. | N.A. | N.A. | $10^{-6}$ |
| $k$ | 2 | N.A. | N.A. | N.A. |

### A.6.4 Hardware and simulation time

To run the simulations, we used a local stationary computer with Ubuntu 20.10, an Intel® Xeon® Silver 4110 Processor (8 cores) and 48GB of ram. On average, it takes approximately 14 days to complete all the simulations contained in this manuscript. Ubuntu is an open-source Operating System using the Linux kernel and based on Debian. For more information, please check `https://ubuntu.com/`.

### A.6.5 Code and libraries

We set up our experiments using Python 3.10 [67] (For more information, please refer to the following link `http://www.python.org`), and made use of the following libraries: Cython [10], NumPy [25], SciPy [68], PyTorch [53], CVXPY [20], MOSEK [1], Seaborn [72], Pandas [40], Matplotlib [27]. In the code, we make use of some code from the Behavior suite [52], which is licensed with the APACHE 2.0 license. Changes, and new code, are published under the MIT license. To run the code, please, read the attached README file for instructions.

# B   Algorithms

In the following section we describe the algorithms that we discuss in this manuscript. For simplicity, we provide a brief summary of them in form of table I.

Table 6: Description of the various algorithms

| Name | Description | Key points |
|---|---|---|
| PS-MDP-NAS | An adaptation of MDP-NAS that uses posterior sampling to sample an MDP $\phi_t$, which is then used to compute the optimal allocation (in Equation (3)). | Requires the user to:
• Keep an estimate of the MDP.
• Perform value/policy iteration.
• Compute the allocation (a convex problem).
• Uses posterior sampling at each time step to sample an MDP and compute the allocation. |
| O-BPI | An adaptation of MDP-NAS that learns the $Q$-values and $M$-values. These values are used to compute the optimal allocation in Equation (3). | • Does not perform value iteration.
• Requires to keep an estimate of the transition function.
• Compute the allocation (a convex problem).
• Uses forced exploration to sample all state-action pairs i.o. |
| Bootstrapped MF-BPI | This algorithm is an extension of O-BPI that computes the allocation using the closed form solution in Proposition 5.1. The $Q, M$-values used to compute the allocation are bootstrap samples. | • Does not perform value iteration and does not require to keep an estimate of the model.
• Closed form solution for the allocation.
• Uses bootstrapping (forced exploration not necessary). |
| DBMF-BPI | An extension of BO-MFPI to the Deep-RL setting. The baseline architecture is inspired from BootstrappedDQN with prior networks. This architecture is then adapted to compute a generative allocation. | • Like boostrapped MF-BPI.
• Requires to keep an ensemble of $Q, M$-networks.
• Can be applied to continuous state spaces. |

## B.1   PS-MDP-NAS - Posterior Sampling for navigating MDPs

In this sub-section we present PS-MDP-NAS, an adaptation of MDP-NAS that uses posterior sampling. An outline of the algorithm is given in Algorithm 3. For simplicity, we omit the use of any stopping rule, since we focus more on the practical implementation of the algorithm.

At each timestep we sample an MDP $\phi_t$ from a posterior distribution, and use it to solve the optimal allocation in Theorem 4.2 with navigation constraints. When computing the optimal allocation $\arg\inf_{\omega \in \Omega(\phi)} U(\omega)$, we limit the maximum number of possible values of $k$ for simplicity.

The algorithm considers a Dirichlet prior for the transition function, and a Gamma prior for the reward distribution. Specifically, for each $(s, a)$ we have a prior hyper-parameter $\rho_{sa} \in \mathbb{R}^{|S|}$ that characterizes the transition function, and two other hyper-parameters $\alpha_{sa}, \beta_{sa} \in \mathbb{R}$ that characterize the reward distribution for each $(s, a)$.

---

**Algorithm 3** PS-MDP-NAS - Posterior Sampling for navigating MDPs

---

**Require:** Parameters $(\rho, \alpha, \beta)$.
1: Initialize counter $N_0(s, a, s') \leftarrow 0$ for all $(s, a, s') \in S \times A \times S$.
2: Observe $s_0 \sim p_0$.
3: **for** $t = 0, 1, 2, \ldots,$ **do**
4:     **Computing the allocation.**
5:         Sample a transition function $P_t(\cdot|s, a) \sim \text{Dir}(\rho_{sa}(t))$ and reward distribution $q_t(\cdot|s, a) \sim$
        $\text{Ber}(\alpha_{sa}(t)/(\alpha_{sa}(t) + \beta_{sa}(t)))$.
6:         Perform policy iteration using $\phi_t = (P_t, q_t)$ and compute $\pi_t^\star$, the greedy policy at time $t$.
        Use $\pi_t^\star$ to derive the various quantities needed to compute the allocation in Thm. 4.2
7:         Compute allocation $\omega^{(t)}$ by solving the optimization problem in Thm. 4.2 using $(P_t, q_t)$.
8:     **Sampling step.**
9:         Sample $a_t \sim \omega^{(t)}(s_t, \cdot)$ and observe $(r_t, s_{t+1}) \sim q(\cdot|s_t, a_t) \otimes P(\cdot|s_t, a_t)$.
10:    **Posterior update.**
11:        Update number of visits $N_{t+1}(s_t, a_t, s_{t+1}) \leftarrow N_t(s_t, a_t, s_{t+1}) + 1$ and total cumulative
        reward $R_{t+1}(s_t, a_t) \leftarrow R_t(s_t, a_t) + r_t$.
12:        Update posterior parameters

$$\rho_{sa}(t+1) \leftarrow \rho_{sa} + N_{t+1}(s, a, s'),$$
$$\alpha_{sa}(t+1) \leftarrow \alpha_{sa} + R_{t+1}(s, a),$$
$$\beta_{sa}(t+1) \leftarrow \beta_{sa} + N_{t+1}(s, a) - R_{t+1}(s, a).$$

13: **end for**

---

After observing an experience at time $t$, the posterior parameters $\rho_{sa}(t), \alpha_{sa}(t), \beta_{sa}(t)$ at time $t$ are computed as follows

$$\rho_{sa}(t) \leftarrow \rho_{sa} + N_t(s, a, s'),$$
$$\alpha_{sa}(t) \leftarrow \alpha_{sa} + R_t(s, a),$$
$$\beta_{sa}(t) \leftarrow \beta_{sa} + N_t(s, a) - R_t(s, a).$$

where $N_t(s, a, s')$ is the number of times the agent experienced state $s'$ after choosing action $a$ in state $s$ up to time $t$, $R_t(s, a) = \sum_{n=0}^{t} r_n \mathbf{1}\{s_n = s \wedge a_n = a\}$ is the total cumulative reward observed up to time $t$ after choosing action $a$ in state $s$, and, lastly, $N_t(s, a) = \sum_{s'} N_t(s, a, s')$ is the total number of times the agent chose action $a$ in state $s$.

## B.2 O-BPI - Online Best Policy Identification

In this part, we introduce O-BPI, or Online Best Policy Identification. This procedure bears resemblance to MDP-NAS, but sidesteps the need for policy iteration at every timestep. Instead, we employ stochastic approximation to learn the $(Q, M)$-values and use these calculated values to compute the allocation. We describe a variation where the user exclusively learns the $M$-function for a given $k$. It's important to note, however, that the agent has the capability to learn multiple $M$-functions, for varying $k$ values, to better approximate the true solution.

We present a version of the algorithm that uses forced exploration, where we mix the allocation that we obtain from Theorem 4.2 with a uniform distribution. It's straightforward to derive an extension using bootstrapping, as we show in subsequent subsections.

**O-BPI.** To compute the allocation $\omega$ we require to estimate the transition function (*e.g.*, using maximum likelihood), and we denote its estimate at time $t$ by $\hat{P}_t$. To derive $\omega$, as for MF-BPI, we compute $\pi_t^\star, \Delta_t$ and $\Delta_{\min, t}$ using the estimate $Q$-function $Q_t$. Then, we solve the following convex problem

$$\arg\inf_{\omega \in \mathcal{C}_t} \max_{s, a \neq \pi_t^\star(s)} \frac{2 + 8\varphi^2 M_t(s, a)^{2^{1-k}}}{\omega(s, a)(\Delta_t(s, a) + \lambda)^2} + \max_{s'} \frac{C_t(s')(1 + \gamma)^2}{\omega(s', \pi^\star(s'))(\Delta_t(s, a) + \lambda)^2(1 - \gamma)^2}, \quad (7)$$

where $C_t(s') = \max\left(4, 16\gamma^2\varphi^2 M_t(s', \pi_t^\star(s'))^{2^{1-k}}\right)$. The constraint set $\mathcal{C}_t$ is simply $\Delta(S \times A)$ in the generative case, and $\mathcal{C}_t = \{\omega : \omega_{s'} = \sum_{s,a} \omega(s, a)\hat{P}_t(s'|s, a), \forall s'\}$ in the case with navigation

constraints. In particular, for finite state-action MDPs we use $N_t(s, a, s')$ (the number of visits up to time $t$ of $(s, a, s')$) to estimate $P$. As in MDP-NAS [38], to ensure that the various estimates asymptotically converge to the true quantities, we force exploration using a D-tracking like procedure, that is, with probability $\epsilon_t \propto 1/N_t(s_t)^\lambda, \lambda \in (0, 1]$, we choose an action uniformly at random in state $s_t$ at time $t$ (since this type of forced exploration is slightly different from the one proposed in [38]. We have the following guarantee.

**Lemma B.1** (Forced exploration). *Let* $\epsilon_t(s) := 1/N_t(s)^\alpha$ *with* $\alpha \in (0, 1]$. *Then,* O-BPI *satisfies* $\mathbb{P}_\phi(\forall (s, a) \in S \times A, \lim_{t \to \infty} N_t(s, a) = \infty) = 1$.

*Proof.* The lemma follows from Observation 1 in [59]. We use the fact that in communicating MDPs every state gets visited infinitely often as long as each action is chosen infinitely often in each state. Denote by $\mathbb{P}(a_t = a | s_t = s, N_t(s) = i)$ the probability that action $a$ is executed at the $i^{th}$ visit to state $s$. The forced exploration step in O-BPI ensures that $\mathbb{P}(a_t = a | s_t = s, N_t(s) = i) \geq \epsilon_t(s)/|A| = 1/(i^\alpha|A|)$. Consequently, for all $(s, a)$ and $0 < \alpha \leq 1$ we have that

$$\sum_{i=1}^\infty \mathbb{P}(a_t = a | s_t = s, N_t(s) = i) \geq \frac{1}{|A|} \sum_{i=1}^\infty \frac{1}{i^\alpha} = \infty.$$

By the Borel-Cantelli lemma it follows that asymptotically each action is chosen infinitely often in each state, which yields the desired result. □

### B.3 Boostrapped MF-BPI - Model Free Best Policy Identification

In this section, we describe in more detail bootstrapped MF-BPI. MF-BPI is a model-free algorithm that adapts exploration based on a sample-complexity bound, while using bootstrapping to characterize the epistemic uncertainty. The algorithm also relies on a closed-form solution for computing the allocation $\omega$, which eliminates the need for solving an optimization problem.

Recall the closed form solution for the allocation $\omega$ from Corollary 5.1:

$$\omega(s, a) \propto \begin{cases} H(s, a) & \text{if } a \neq \pi^\star(s) \\ \sqrt{H \sum_{s, a \neq \pi^\star(s)} H(s, a)/|S|} & \text{otherwise,} \end{cases} \tag{8}$$

where $H(s, a)$ and $H$ are defined as follows:

$$H(s, a) = \frac{2 + 8\varphi^2 M_{sa}^k [V^\star]^{2^{1-k}}}{(\Delta(s, a) + \lambda)^2}, \tag{9}$$

$$H = \frac{\max_{s'} C(s')(1 + \gamma)^2}{(\Delta_{\min} + \lambda)^2 (1 - \gamma)^2}, \tag{10}$$

for some fixed value $k$ that should be treated as a hyper-parameter, parameter $\lambda \geq 0$ and $C(s') = \max\left(4, 16\gamma^2\varphi^2 M_{s', \pi^\star(s')}^k [V^\star]^{2^{1-k}}\right)$. Rather than resorting to policy iteration, our approach involves learning the $Q$-values and $M$-values through stochastic approximation. The algorithm keeps track of estimates $Q_t(s, a)$ and $M_t(s, a)$ for all states and actions up to a given time $t$. The updates for the stochastic approximation are carried out at every time step $t$ and can be represented as:

$$Q_{t+1}(s_t, a_t) = Q_t(s_t, a_t) + \alpha_t(s_t, a_t)\left(r_t + \gamma \max_a Q_t(s_{t+1}, a) - Q_t(s_t, a_t)\right), \tag{11}$$

$$M_{t+1}(s_t, a_t) = M_t(s_t, a_t) + \beta_t(s_t, a_t)\left((\delta_t'/\gamma)^{2^k} - M_t(s_t, a_t)\right). \tag{12}$$

In this equation, $\delta_t' = r_t + \gamma \max_a Q_{t+1}(s_{t+1}, a) - Q_{t+1}(s_t, a_t)$, and $(\alpha_t, \beta_t)_{t \geq 0}$ are the learning rates that meet the Robbins-Monroe conditions [13].

**Bootstrap sample.** Our method employs a bootstrap sampling strategy to estimate uncertainties in a non-parametric way. This approach can augment a forced exploration step, ensuring the convergence of the estimates asymptotically. We maintain a collection of $(Q, M)$-values and produce a new bootstrap sample $(\hat{Q}_t, \hat{M}_t)$ at every time step $t$. In particular, we start with an ensemble of $Q$-functions $Q_1, \ldots, Q_B$ (similarly for $M_1, \ldots, M_B$) initialized uniformly at random in $[0, 1/(1 - \gamma)]$

---

**Algorithm 4** Bootstrapped MF-BPI

---

**Require:** Parameters $(\lambda, k, p)$; ensemble size $B$; learning rates $\{(\alpha_{t,b}, \beta_{t,b})\}_{t,b}$.

1: Initialize $Q_{1,b}(s,a) \sim \mathcal{U}([0, 1/(1-\gamma)])$ and $M_{1,b}(s,a) \sim \mathcal{U}([0, 1/(1-\gamma)^{2^k}])$ for all $(s,a) \in S \times A$ and $b \in [B]$.

2: Observe $s_0 \sim p_0$.

3: **for** $t = 0, 1, 2, \ldots,$ **do**

4:     **Compute allocation.**

5:     Sample $\xi \sim \mathcal{U}([0,1])$ and set, $\hat{Q}_t(s,a) = \text{Quantile}_\xi(\{Q_{t,1}(s,a), \ldots, Q_{t,B}(s,a)\})$ (sim. $\hat{M}_t$) for all $(s,a)$.

6:     Compute generative solution $\omega_o^{(t)}$ using Proposition 5.1. Let $\pi_t^\star(s) = \arg\max_a \hat{Q}_t(s,a)$, $\Delta_t(s,a) = \hat{Q}_t(s, \pi_t^\star(s)) - \hat{Q}_t(s,a)$, $\Delta_{\min,t} = \min_{s, a \neq \pi_t^\star(s)} \Delta_t(s,a)$ and

$$H_t(s,a) := \frac{2 + 8\varphi^2 \hat{M}_t(s,a)^{2^{1-k}}}{(\Delta_t(s,a) + \lambda)^2},$$

$$H_t := \frac{\max_{s'} 4(1+\gamma)^2 \max(1, 4\gamma^2 \varphi^2 \hat{M}_t(s', \pi_t^\star(s'))^{2^{1-k}})}{(\Delta_{\min,t} + \lambda)^2 (1-\gamma)^2}$$

7:     Set

$$\omega_o^{(t)}(s_t, a) = \begin{cases} H_t(s_t, a) & \text{if } a \neq \pi_t^\star(s_t), \\ \sqrt{H_t \sum_{s, a \neq \pi_t^\star(s)} H_t(s,a)/|S|} & \text{otherwise}. \end{cases}$$

8:     Let $\omega^{(t)}(s_t, a) = \frac{\omega_o^{(t)}(s_t, a)}{\sum_{a'} \omega_o^{(t)}(s_t, a')}$ be the policy at time $t$ in state $s_t$.

9:     Sample $a_t \sim \omega^{(t)}(s_t, \cdot)$; observe $(r_t, s_{t+1}) \sim q(\cdot|s_t, a_t) \otimes P(\cdot|s_t, a_t)$.

10:     **Training step.**

11:     **for** $b = 1, \ldots, B$ **do**

12:         With probability $p$, using the experience $(s_t, a_t, r_t, s_{t+1})$, update the values $Q_{t,b}, M_{t,b}$ using Equations (5) and (6)

$$Q_{t+1,b}(s_t, a_t) = Q_{t,b}(s_t, a_t) + \alpha_{t,b}(s_t, a_t)\left(r_t + \gamma \max_a Q_{t,b}(s_{t+1}, a) - Q_{t,b}(s_t, a_t)\right),$$

$$M_{t+1,b}(s_t, a_t) = M_{t,b}(s_t, a_t) + \beta_{t,b}(s_t, a_t)\left((\delta'_{t,b}/\gamma)^{2k} - M_{t,b}(s_t, a_t)\right),$$

        where $\delta'_{t,b} = r_t + \gamma \max_a Q_{t+1,b}(s_{t+1}, a) - Q_{t+1,b}(s_t, a_t)$.

13:     **end for**

14:     Compute greedy policy as

$$\bar{\pi}_t^\star(s) \leftarrow \text{Median}(\{\arg\max_a Q_{t+1,1}(s,a), \ldots, \arg\max_a Q_{t+1,B}(s,a)\}).$$

15: **end for**

---

(similarly $[0, 1/(1-\gamma)^{2k}]$). It's important to highlight that initializing the ensemble members across the full range of potential values is essential to account for uncertainties not arising from the collected data.

At each timestep $t$, a bootstrap sample $\hat{Q}_t(s,a)$ (and similarly $\hat{M}_t$) is generated by sampling a uniform random variable $\xi \sim \mathcal{U}([0,1])$, and, for each $(s,a)$, $\hat{Q}_t(s,a) = \text{Quantile}_\xi(Q_{t,1}(s,a), \ldots, Q_{t,B}(s,a))$; in other words, $\hat{Q}_t(s,a)$ is the $\xi$-quantile of $Q_{t,1}(s,a), \ldots, Q_{t,B}(s,a)$ assuming a linear interpolation between the $Q$-values. This method is akin to sampling from an empirical cumulative distribution function (CDF), where the CDF in this case embodies the uncertainty over the $Q$-values. We also apply the same bootstrap sampling procedure to $\hat{M}_t(s,a)$. Empirically, we found the method to work for small values of the ensemble size $B$ for most problems, but we did not conduct extensive research on this topic. A promising venue of research is to study how the parameter $p$ and the ensemble size $B$ can be tuned for the problem at hand.

**Computing the allocation.** This bootstrap sample $(\hat{Q}_t, \hat{M}_t)$ is subsequently used to calculate the allocation $\omega^{(t)}$, where $\Delta_t(s,a) = \max_{a'} \hat{Q}_t(s,a') - \hat{Q}_t(s,a)$, $\pi_t^\star(s) = \arg\max_a \hat{Q}_t(s,a)$, and $\Delta_{\min,t} = \min_{s,a \neq \pi_t^\star(s)} \Delta_t(s,a)$. Then, we set

$$H_t(s,a) := \frac{2 + 8\varphi^2 \hat{M}_t(s,a)^{2^{1-k}}}{(\Delta_t(s,a) + \lambda)^2},$$

$$H_t := \frac{\max_{s'} 4(1+\gamma)^2 \max(1, 4\gamma^2\varphi^2 \hat{M}_t(s', \pi_t^\star(s'))^{2^{1-k}})}{(\Delta_{\min,t} + \lambda)^2(1-\gamma)^2}$$

as well as

$$\omega_o^{(t)}(s_t, a) = \begin{cases} H_t(s_t, a) & \text{if } a \neq \pi_t^\star(s_t), \\ \sqrt{H_t \sum_{s,a\neq\pi_t^\star(s)} H_t(s,a)/|S|} & \text{otherwise}. \end{cases}$$

The final policy is then obtain by normalizing $\omega_o^{(t)}$: $\omega^{(t)}(s_t, a) = \frac{\omega_o^{(t)}(s_t,a)}{\sum_{a'} \omega_o^{(t)}(s_t,a')}$.

**Greedy policy.** Lastly, an overall greedy policy $\bar{\pi}_t^\star$ can be estimated by using the ensemble of $Q$-functions. For example, by majority voting as

$$\bar{\pi}_t^\star(s) \leftarrow \text{Mode}(\{\arg\max_a Q_{t+1,1}(s,a), \ldots, \arg\max_a Q_{t+1,B}(s,a)\}).$$

### B.4 DBMF-BPI - Deep Boostrapped Model Free Best Policy Identification

To generalize bootstrapped MF-BPI to continuous Markov Decision Processes (MDPs), we propose DBMF-BPI. DBMF-BPI leverages the concept of prior networks from BSP (Bootstrapping with Additive Prior) [50], to account for uncertainty not arising from the observed data.

**Ensemble.** As in the previous method, we maintain an ensemble of $Q$-values $Q_{\theta_1}, \ldots, Q_{\theta_B}$ (along with their target networks) and an ensemble of $M$-values $M_{\tau_1}, \ldots, M_{\tau_B}$. Specifically, $Q$-values are computed as follows for a generic $b$-th member of the ensemble

$$Q_{\theta_b}(s,a) = Q_{\theta_{b,0}}(s,a) + \beta_Q Q_{\theta_{b,p}}(s,a),$$

where $\beta_Q \geq 0$ is a hyper-parameter defining the scale of the prior, $\theta_{b,0}$ is a learnable parameter, and $Q_{\theta_{b,p}}$ is a fixed, randomly-initialize, $Q$-network that serves as a randomized prior value function. Similarly, we compute the $M$-values as

$$M_{\tau_b}(s,a) = M_{\tau_{b,0}}(s,a) + \beta_M M_{\tau_{b,p}}(s,a),$$

where $\beta_M \geq 0$ is a hyper-parameter, $\tau_{b,0}$ is a learnable parameter and $M_{\tau_{b,p}}$ is a fixed random prior network for the $M$- function.

Note that the function of the prior network is to guarantee that the $(Q, M)$-values are capable of covering the full spectrum of potential values. This is similar to the random initialization procedure in MF-BPI. An alternate strategy might involve initializing the network in an optimistic way (*i.e.*, by sampling parameters from a Gaussian distribution with larger variance), but our observations indicate that this may lead to worse performance.

**Bootstrap sample.** As before, at each timestep $t$ a bootstrap sample $\hat{Q}_t(s,a)$ (and similarly $\hat{M}_t$) is generated by sampling a uniform random variable $\xi \sim \mathcal{U}([0,1])$, and, for each $(s,a)$, set $\hat{Q}_t(s,a) = \text{Quantile}_\xi(Q_{t,\theta_1}(s,a), \ldots, Q_{t,\theta_B}(s,a))$. However, for numerical stability, we found it was most effective to sample $\xi$ at the end of an episode, or every $n \propto (1-\gamma)^{-1}$ steps.

**Computing the allocation.** Using the bootstrap sample $(\hat{Q}_t, \hat{M}_t)$ we compute the allocation as follows. We set

$$H_t(s_t, a) = \frac{2 + 8\varphi^2 \hat{M}_t(s_t,a)^{2^{1-k}}}{(\Delta_t(s_t,a) + \lambda)^2}, \tag{13}$$

$$H_t = \frac{4(1+\gamma)^2 \max(1, 4\gamma^2\varphi^2 \hat{M}_t(s_t, \pi_t^\star(s_t))^{2^{1-k}})}{(\Delta_{\min,t} + \lambda)^2(1-\gamma)^2}, \tag{14}$$

where $\pi_t^\star(s_t) = \arg\max_a \hat{Q}_t(s_t, a)$. Note that $H_t$ is an approximation of the true value (we are not taking the maximum over all possible states). Subsequently, we establish the allocation $\omega_o^{(t)}$: $\omega_o^{(t)}(s_t, a) = H_t(s_t, a)$ if $a \neq \pi_t^\star(s_t)$, and $\omega_o^{(t)}(s_t, a) = \sqrt{H_t \sum_{a \neq \pi_t^\star(s_t)} H_t(s_t, a)}$ otherwise. In the final step, we construct an $\epsilon_t$-soft exploration policy $\omega^{(t)}(s_t, \cdot)$ by blending $\omega_o^{(t)}(s_t, \cdot) / \sum_a \omega_o^{(t)}(s_t, a)$ with a uniform distribution, utilizing an exploration parameter $\epsilon_t$.

**Training and minimum gap estimation.** The training procedure follows that of the classical DQN algorithm [41]. Each $Q$-network is trained by minimizing an MSE loss criterion. We use also the MSE loss to train the $M$-networks over a batch sampled from the replay buffer (note that the $M$-networks do not require a target network).

Next, $\Delta_{\min,t}$ is estimated through stochastic approximation, using the smallest gap from the most recent batch of transitions retrieved from the replay buffer as a reference. In particular, the target is given by the following expression

$$\delta_t = \min_{b \in [B]} \min_{j \in \mathcal{B}} \max_{a \neq \pi_{\theta_b}^\star(s_j)} Q_{\theta_b}(s_j, \pi_{\theta_b}^\star(s_j)) - Q_{\theta_b}(s_j, a)$$

with $\pi_{\theta_b}(s) = \arg\max_a Q_{\theta_b}(s, a)$. The estimate is then updated as $\Delta_{\min,t+1} \leftarrow (1 - \alpha_t)\Delta_{\min,t} + \alpha_t \delta_t$ for some learning rate $\alpha_t = O(1/t)$.

**Greedy policy** Lastly, an overall greedy policy $\bar{\pi}_t^\star$ can be estimated by using the ensemble of $Q$-functions. For example, by majority voting as

$$\bar{\pi}_t^\star(s) \leftarrow \text{Mode}(\{\arg\max_a Q_{t+1,\theta_1}(s, a), \dots, \arg\max_a Q_{t+1,\theta_B}(s, a)\}).$$

The full pseudo-code of the algorithm can be found in the next page.

**Algorithm 5** DBMF-BPI (Deep Bootstrapped Model Free BPI) - Full Algorithm

---

**Require:** Parameters $(\lambda, k)$; ensemble size $B$; exploration rate $\{\epsilon_t\}_t$; estimate $\Delta_{\min,0}$; mask probability $p$.
1: **function** MainLoop
2:     Initialize replay buffer $\mathcal{D}$, networks $Q_{\theta_b}, M_{\tau_b}$ and targets $Q_{\theta_b'}$ for all $b \in [B]$.
3:     **for** $t = 0, 1, 2, \ldots,$ **do**
4:         **Sampling step.**
5:         Compute allocation $\omega^{(t)} \leftarrow \texttt{ComputeAllocation}(s_t, \{Q_{\theta_b}, M_{\tau_b}\}_{b \in [B]}, \Delta_{\min,t}, \gamma, \lambda, k, \epsilon_t)$.
6:         Sample $a_t \sim \omega^{(t)}(s_t, \cdot)$ and observe $(r_t, s_{t+1}) \sim q(\cdot | s_t, a_t) \otimes P(\cdot | s_t, a_t)$.
7:         Add transition $z_t = (s_t, a_t, r_t, s_{t+1})$ to the replay buffer $\mathcal{D}$.
8:         **Training step.**
9:         Sample a batch $\mathcal{B}$ from $\mathcal{D}$, and with probability $p$ add the $i^{th}$ experience in $\mathcal{B}$ to a sub-batch $\mathcal{B}_b, \ \forall b \in [B]$. Update the $(Q, M)$-values of the $b^{th}$ member in the ensemble using $\mathcal{B}_b$: $\{Q_{\theta_b}, Q_{\theta_b'}, M_{\tau_b}\}_{b \in [B]} \leftarrow \texttt{Training}(\{\mathcal{B}_b, Q_{\theta_b}, Q_{\theta_b'}, M_{\tau_b}\}_{b \in [B]})$.
10:         Update estimate $\Delta_{\min,t+1} \leftarrow \texttt{EstimateMinimumGap}(\Delta_{\min,t}, \mathcal{B}, \{Q_{\theta_b}\}_{b \in [B]})$.
11:     Compute greedy policy as

$$\bar{\pi}_t^{\star}(s) \leftarrow \text{Median}(\{\arg\max_a Q_{t+1,\theta_1}(s, a), \ldots, \arg\max_a Q_{t+1,\theta_B}(s, a)\}).$$

12:     **end for**
13: **end function**

1: **function** EstimateAllocation$(s_t, \{Q_{\theta_b}, M_{\tau_b}\}_{b \in [B]}, \Delta_{\min,t} \gamma, \lambda, k, \epsilon_t)$
2:     Sample $\xi \sim \mathcal{U}([0, 1])$ and set, $\hat{Q}_t(s_t, a) = \text{Quantile}_\xi(\{Q_{t,\theta_1}(s_t, a), \ldots, Q_{t,\theta_B}(s_t, a)\})$ (sim. $\hat{M}_t$).
3:     Let $\pi_t^{\star}(s_t) = \arg\max_a \hat{Q}_t(s_t, a)$, and set $\Delta_t(s_t, a) = \hat{Q}_t(s_t, \pi_t^{\star}(s_t)) - \hat{Q}_t(s_t, a)$.
4:     Compute MDP-related quantities

$$H_t(s_t, a) := \frac{2 + 8\varphi^2 \hat{M}_t(s_t, a)^{2^{1-k}}}{(\Delta_t(s_t, a) + \lambda)^2},$$

$$H_t := \frac{4(1 + \gamma)^2 \max(1, 4\gamma^2 \varphi^2 \hat{M}_t(s_t, \pi_t^{\star}(s_t))^{2^{1-k}})}{(\Delta_{\min,t} + \lambda)^2 (1 - \gamma)^2}$$

5:     Set

$$\omega_o^{(t)}(s_t, a) = \begin{cases} H_t(s_t, a) & \text{if } a \neq \pi_t^{\star}(s_t), \\ \sqrt{H_t \sum_{a \neq \pi_t^{\star}(s_t)} H_t(s_t, a)} & \text{otherwise}. \end{cases}$$

6:     **Return** $\omega^{(t)}(s_t, a) = \frac{\epsilon_t}{|A|} + (1 - \epsilon_t) \frac{\omega_o^{(t)}(s_t, a)}{\sum_{a'} \omega_o^{(t)}(s_t, a')}$, the policy at time $t$ in state $s_t$.
7: **end function**

1: **function** Training$(\{\mathcal{B}_b, Q_{\theta_b}, Q_{\theta_b'}, M_{\tau_b}\}_{b \in [B]})$
2:     **for** each model in the ensemble $b = 1, \ldots, B$ **do**
3:         Compute targets $y_j = r_j + \gamma \max_a Q_{\theta_b'}(s_{j+1}, a)$ and perform a gradient descent step on $Q_{\theta_b}$ using $\nabla_{\theta_b}(y_j - Q_{\theta_b}(s_j, a_j))^2$ for all $j \in \mathcal{B}_b$.
4:         Compute targets $\bar{y}_j = (r_j + \max_a Q_{\theta_b}(s_{j+1}, a) - Q_{\theta_b}(s_j, a_j))/\gamma$ and perform a gradient descent step on $M_{\tau_b}$ using $\nabla_{\tau_b}(\bar{y}_j^{2^k} - M_{\tau_b}(s_j, a_j))^2$.
5:     **end for**
6:     Every $K$ steps update target models: $\theta_{b'} \leftarrow \theta_b$ for all $b \in [B]$.
7:     **Return** updated models $\{Q_{\theta_b}, Q_{\theta_b'}, M_{\tau_b}\}_{b \in [B]}$.
8: **end function**

1: **function** EstimateMinimumGap$(\Delta_{\min,t}, \mathcal{B}, \{Q_{\theta_b}\}_{b \in [B]})$
2:     Set learning rate $\alpha_t = O(1/t)$.
3:     Update estimate of $\Delta_{\min,t}$: let $\pi_{\theta_b}^{\star}(s_j) = \arg\max_a Q_{\theta_b}(s_j, a)$ and compute target

$$\delta_t = \min_{b \in [B]} \min_{j \in \mathcal{B}} \max_{a \neq \pi_{\theta_b}^{\star}(s_j)} Q_{\theta_b}(s_j, \pi_{\theta_b}^{\star}(s_j)) - Q_{\theta_b}(s_j, a)$$

    and update estimate $\Delta_{\min,t+1} \leftarrow (1 - \alpha_t)\Delta_{\min,t} + \alpha_t \delta_t$.
4:     **Return** updated estimate $\Delta_{\min,t+1}$.
5: **end function**

---

## C Proofs

In this appendix, we provide the proofs of our main results. We start with some preliminary results. We then introduce new notation to accommodate extensions beyond the assumptions made in the main body of the paper, and prove our main theorem. Specifically, we broaden our sample-complexity bounds to encompass communicating MDPs without a unique optimal policy.

### C.1 Preliminaries

Let $V : \mathcal{S} \to \mathbb{R}$ be a bounded function. We show that $\mathrm{Var}_{sa}[V] \leq \mathrm{MD}_{sa}[V]^2$. This inequality follows directly from the Bhatia-Davis inequality [12]. Applied to the value function of our MDP, this result implies that in the bound derived in Theorem 4.1, the term corresponding to the span of $V^\star$ might be sometimes dominant, and we might indeed wish to remove it from the upper bound.

**Lemma C.1.** *Consider an MDP $\phi$ with $|S|$ states and a bounded vector $V \in \mathbb{R}^{|S|}$. For any $(s, a)$, we have $\mathrm{Var}_{sa}[V] \leq \mathrm{MD}_{sa}[V]^2$. If $\mathrm{MD}_{sa}[V] \leq 1$ then $\mathrm{Var}_{sa}[V] \leq \mathrm{MD}_{sa}[V]$.*

*Proof of Lemma C.1.* The result is obtained leveraging the Bhatia-Davis inequality [12]. Fix $(s, a)$, and consider a bounded vector $V$. Let $\mu(s, a) = \mathbb{E}_{s' \sim P(\cdot|s,a)}[V(s')]$, $M = \max_s V(s)$ and $m = \min_s V(s)$. Then, define

$$G(s, a) := \mathbb{E}_{s' \sim P(\cdot|s,a)}[(M - V(s'))(V(s') - m)].$$

We have $G(s, a) = -mM - \mathbb{E}_{s' \sim P(\cdot|s,a)}[V(s')^2] + (M + m)\mu(s, a)$. Since $0 \leq G(s, a)$,

$$-\mu(s, a)^2 \leq -mM - \mathbb{E}_{s' \sim P(\cdot|s,a)}[V(s')^2] + (M + m)\mu(s, a) - \mu(s, a)^2,$$
$$\mathrm{Var}_{P(s,a)}[V] \leq -mM + (M + m)\mu(s, a) - \mu(s, a)^2,$$
$$\mathrm{Var}_{P(s,a)}[V] \leq (M - \mu(s, a))(\mu(s, a) - m).$$

Since $\mathrm{MD}_{sa}[V] = \|V - \mu(s, a)\|_\infty = \max(M - \mu(s, a), \mu(s, a) - m)$, we conclude that

$$\mathrm{Var}_{P(s,a)}[V] \leq \max(M - \mu(s, a), \mu(s, a) - m)^2 = \mathrm{MD}_{P(s,a)}[V]^2.$$

This also implies that, if $\mathrm{MD}_{sa}[V] \leq 1$, then $\mathrm{Var}_{sa}[V] \leq \mathrm{MD}_{sa}[V]$. $\qquad\square$

More generally, we also note that

$$M_{sa}^k[V^\pi]^{2^{-k}} \leq \mathbb{E}_{s' \sim P(\cdot|s,a)} \left[ \left( \max_{s'} V^\pi(s') - \mathbb{E}_{\bar{s} \sim P(\cdot|s,a)}[V^\pi(\bar{s})] \right)^{2^k} \right]^{2^{-k}},$$

$$= \mathbb{E}_{s' \sim P(\cdot|s,a)} \left[ \left\| V^\pi - \mathbb{E}_{\bar{s} \sim P(\cdot|s,a)}[V^\pi(\bar{s})] \right\|_\infty^{2^k} \right]^{2^{-k}},$$

$$= \mathrm{MD}_{sa}[V^\pi].$$

### C.2 Alternative upper bounds

In this subsection, we establish the alternative upper bounds $\bar{U}_\varepsilon$ of the sample complexity lower bound proposed in Theorem 4.2. Our results extend those of [37] to MDPs where the optimal policy might not be unique.

#### C.2.1 Sample complexity lower bound

Assume for now that the way the learner interacts with the MDP corresponds to the generative model: in each round, she can pick any (state, action) pair and observe the corresponding next state and reward. Under this model, the following theorem provides a sample complexity lower bound satisfied by any $(\varepsilon, \delta)$-PAC algorithm.

**Theorem C.2** $((\delta, \varepsilon)$-PAC lower bound). *Consider $\varepsilon \geq 0$, and a communicating MDP $\phi$, not necessarily with a unique optimal policy. Then, the sample complexity $\tau$ of any $(\delta, \varepsilon)$-PAC algorithm under the generative model satisfies the following lower bound:*

$$\mathbb{E}_\phi[\tau] \geq T_\varepsilon \mathrm{kl}(\delta, 1 - \delta), \tag{15}$$

*where $T_\varepsilon = \sup_{\omega \in \Delta(S \times A)} T_\varepsilon(\omega)$ is the optimal characteristic time, and*

$$T_\varepsilon(\omega)^{-1} = \inf_{\psi \in \mathrm{Alt}_\varepsilon(\phi)} \mathbb{E}_{(s,a) \sim \omega}[\mathrm{KL}_{\phi|\psi}(s,a)]. \tag{16}$$

The proof follows the same lines as in [37]. A similar lower bound can be derived in the forward model where the learner has to follow the system trajectory [38]: it is obtained by replacing the supremum over $\omega \in \Delta(S \times A)$ by a supremum over $\omega \in \Omega(\phi)$, to account for the navigation constraints.

### C.2.2   Upper bound on $T_\varepsilon(\omega)$

As explained in [37], even for $\varepsilon = 0$, (16) is in general a non-convex problem. Therefore it may not always be possible to even approximately solve it. An alternative way, introduced in[37], consists in convexifying the problem. The solution of the new problem then gives an upper bound of $T_0$.

To this aim, we will start from the following result, providing a decomposition of the confusing set.

**Proposition C.3.** *We have $T_\varepsilon(\omega) \leq T(\omega)$ for all $\omega$, where $T(\omega)$ is defined as*

$$T(\omega)^{-1} = \min_{\pi \in \Pi_0^\star(\phi)} \min_{s, a \neq \pi(s)} \min_{\psi \in \mathrm{Alt}_{\pi,sa}(\phi)} \mathbb{E}_{(s,a) \sim \omega}[\mathrm{KL}_{\phi|\psi}(s,a)]. \tag{17}$$

*where $\mathrm{Alt}_{\pi,sa}(\phi) = \{\psi : \phi \ll \psi, Q_\psi^\pi(s,a) > V_\psi^\pi(s)\}$.*

*Proof.* A similar result was derived in [37]. Its proof follows directly from Lemma C.10 and Lemma C.11. From Lemma C.10 we have that the set $\mathrm{Alt}(\phi) = \{\psi : \psi \ll \phi, \Pi_0^\star(\phi) \cap \Pi_0^\star(\psi) = \emptyset\}$ contains $\mathrm{Alt}_\varepsilon(\phi)$. From Lemma C.11 we have that $\mathrm{Alt}(\phi) \subseteq \cup_{\pi \in \Pi_0^\star(\phi)} \cup_s \cup_{a \neq \pi(s)} \mathrm{Alt}_{\pi,sa}(\phi)$, where

$$\mathrm{Alt}_{\pi,sa}(\phi) = \{\psi : Q_\psi^\pi(s,a) > V_\psi^\pi(s)\}.$$

Therefore

$$T_\varepsilon(\omega)^{-1} \geq \min_{\pi \in \Pi_0^\star(\phi)} \min_{s, a \neq \pi(s)} \min_{\psi \in \mathrm{Alt}_{\pi,sa}(\phi)} \mathbb{E}_{(s,a) \sim \omega}[\mathrm{KL}_{\phi|\psi}(s,a)] = T(\omega)^{-1}.$$

$\square$

From the previous proposition, we are able to derive the upper bound of $T_\varepsilon$.

**Theorem C.4.** *Consider a communicating MDP $\phi$, not necessarily with a unique optimal policy. Then, for every $(s,a)$ there exists $\bar{k}(s,a) \in \mathbb{N}$ s.t. for all $\omega \in \Delta(S \times A)$ we have*

$$T_\varepsilon(\omega) \leq U(\omega), \tag{18}$$

*with*

$$U(\omega) = \max_{\pi \in \Pi_0^\star(\phi)} \max_{s, a \neq \pi(s)} \left( \frac{2 + 8\varphi^2 M_{sa}^{(\bar{k}(s,a))}[V_\phi^\star]^{2^{1-\bar{k}(s,a)}}}{\Delta_\pi(s,a)^2 \omega(s,a)} + \max_{s'} \frac{4C^\pi(s')(1+\gamma)^2}{\omega(s',\pi(s'))\Delta_\pi(s,a)^2(1-\gamma)^2} \right), \tag{19}$$

*where $\Delta_\pi(s,a) := V_\phi^\pi(s) - Q_\phi^\pi(s,a)$ and $C^\pi(s') = \max\left(1, 4\gamma^2\varphi^2 M_{s'\pi(s')}^{(\bar{k}(s',\pi(s')))}[V_\phi^\star]^{2^{1-\bar{k}(s',\pi(s'))}}\right)$.*

*Proof.* The proof is similar as that of Theorem 1 in [37]. We start from the result of Proposition C.3:

$$T_\varepsilon(\omega)^{-1} \geq \min_{\pi \in \Pi_0^\star(\phi)} \min_{s, a \neq \pi(s)} \inf_{\psi \in \mathrm{Alt}_{\pi,sa}(\phi)} \mathbb{E}_{(s,a) \sim \omega}[\mathrm{KL}_{\phi|\psi}(s,a)].$$

For a fixed $(\pi, s, a)$, the constraint $\inf_{\psi \in \mathrm{Alt}_{\pi,sa}(\phi)}$ does not involve the pairs $(\tilde{s}, \tilde{a}) \in S \times A \setminus \{(s,a), (s', \pi(s'))_{s' \in S}\}$. As argued in [37], by convexity, the solution must satisfy $\mathrm{KL}_{\phi|\psi}(\tilde{s}, \tilde{a}) = 0$ for those pairs. Hence

$$\inf_{\psi \in \mathrm{Alt}_{\pi,sa}(\phi)} \mathbb{E}_{(s,a) \sim \omega}[\mathrm{KL}_{\phi|\psi}(s,a)] =$$

$$\inf_{\psi \in \mathrm{Alt}_{\pi,sa}(\phi)} \omega(s,a)\mathrm{KL}_{\phi|\psi}(s,a) + \sum_{s'} \omega(s', \pi(s'))\mathrm{KL}_{\phi|\psi}(s', \pi(s')).$$

Let $\Delta_\pi(s,a) := V_\phi^\pi(s) - Q_\phi^\pi(s,a)$. Then, using the fact that $Q_\psi^\pi(s,a) > V_\psi^\pi(s)$, we obtain

$$\Delta_\pi(s,a) < V_\phi^\pi(s) - Q_\phi^\pi(s,a) + Q_\psi^\pi(s,a) - V_\psi^\pi(s).$$

This is similar to condition (5) in [37]. Next, let $\Delta r(s,a) = r_\psi(s,a) - r_\phi(s,a)$, $\Delta P(s,a) = P_\psi(s,a) - P_\phi(s,a)$, where the distribution $P(s,a)$ of the next state given $(s,a)$ is represented as a column vector of dimension $|S|$. Further define the vector difference between the value in $\psi$ and $\phi$ of $\pi$: $\Delta V^\pi = \begin{bmatrix} V_\psi^\pi(s_1) - V_\phi^\pi(s_1) & \cdots & V_\psi^\pi(s_{|S|}) - V_\phi^\pi(s_{|S|}) \end{bmatrix}^\top$. Then, letting $\mathbf{1}(s) = e_s$ be the unit vector with 1 in position $s$, we find

$$\Delta_\pi(s,a) < Q_\psi^\pi(s,a) - Q_\phi^\pi(s,a) - \mathbf{1}(s)^\top \Delta V^\pi,$$
$$< \Delta r(s,a) + \gamma(P_\psi(s,a)^\top V_\psi^\pi - P_\phi(s,a)^\top V_\phi^\pi) - \mathbf{1}(s)^\top \Delta V^\pi,$$
$$< \Delta r(s,a) + \gamma \Delta P(s,a)^\top V_\phi^\pi + (\gamma P_\psi(s,a) - \mathbf{1}(s))^\top \Delta V^\pi.$$

Now, observe that:

$$V_\psi^\pi(s) - V_\phi^\pi(s) = \Delta r(s,\pi(s)) + \gamma(P_\psi(s,\pi(s))^\top V_\psi^\pi - P_\phi(s,\pi(s))^\top V_\phi^\pi),$$
$$= \Delta r(s,\pi(s)) + \gamma(P_\psi(s,\pi(s))^\top \Delta V^\pi + \Delta P(s,\pi(s))^\top V_\phi^\pi),$$
$$\leq \left| \Delta r(s,\pi(s)) + \gamma(P_\psi(s,\pi(s))^\top \Delta V^\pi + \Delta P(s,\pi(s))^\top V_\phi^\pi) \right|,$$
$$\leq \max_{s'} |\Delta r(s',\pi(s')) + \gamma \Delta P(s',\pi(s'))^\top V_\phi^\pi| + \gamma \max_{\tilde{s}} |V_\psi^\pi(\tilde{s}) - V_\phi^\pi(\tilde{s})|.$$

We deduce that:

$$\|\Delta V^\pi\|_\infty \leq \frac{1}{1-\gamma} \left[ \max_{s'} |\Delta r(s',\pi(s'))| + \gamma |\Delta P(s',\pi(s'))^\top V_\phi^\pi| \right].$$

Using the fact that $\|\gamma P_\psi(s,a) - \mathbf{1}(s)\|_1 = |\gamma P(s|s,a) - 1| + \gamma(1 - P(s|s,a)) \leq 1 + \gamma$, we can bound $|(\gamma P_\psi(s,a) - \mathbf{1}(s))^\top \Delta V^\pi|$ as follows:

$$|(\gamma P_\psi(s,a) - \mathbf{1}(s))^\top \Delta V^\pi| \leq \|\gamma P_\psi(s,a) - \mathbf{1}(s)\|_1 \|\Delta V^\pi\|_\infty$$
$$\leq \frac{1+\gamma}{1-\gamma} \left[ \max_{s'} |\Delta r(s',\pi(s'))| + \gamma |\Delta P(s',\pi(s'))^\top V_\phi^\pi| \right].$$

Therefore,

$$\Delta_\pi(s,a) < |\Delta r(s,a)| + \gamma |\Delta P(s,a)^\top V_\phi^\pi| + \tfrac{1+\gamma}{1-\gamma} \left[ \max_{s'} |\Delta r(s',\pi(s'))| + \gamma |\Delta P(s',\pi(s'))^\top V_\phi^\pi| \right].$$

Write each of the terms as a fraction of $\Delta_\pi(s,a)$ using $\{\alpha_i\}_{i=1}^3$, which are non-negative terms satisfying $\sum_{i=1}^3 \alpha_i > 1$:

$$\begin{cases} \alpha_1 \Delta_\pi(s,a) = |\Delta r(s,a)|, \\ \alpha_2 \Delta_\pi(s,a) = \gamma |\Delta P(s,a)^\top V_\phi^\pi|, \\ \alpha_3 \Delta_\pi(s,a) = \dfrac{1+\gamma}{1-\gamma} \max_{s'} \left[ |\Delta r(s',\pi(s'))| + \gamma |\Delta P(s',\pi(s'))^\top V_\phi^\pi| \right]. \end{cases}$$

For the first term, using the Pinsker inequality, we immediately get: $(\alpha_1 \Delta_\pi(s,a))^2 \leq 2\mathrm{KL}_{q_\phi,q_\psi}(s,a)$.

For the second term, using Lemma C.9, we obtain:

$$(\alpha_2 \Delta_\pi(s,a))^2 \leq 8\gamma^2 \varphi^2 M_{sa}^{(\bar{k}(s,a))} [V_\phi^\star]^{2^{1-\bar{k}(s,a)}} \mathrm{KL}_{P_\phi,P_\psi}(s,a).$$

Finally, to bound the last term, using $(a+b)^2 \leq 2(a^2+b^2)$, Lemma C.9 and the Pinsker inequality, we have

$$\left( |\Delta r(s',\pi(s'))| + \gamma |\Delta P(s',\pi(s'))^\top V_\phi^\pi| \right)^2 \leq 2 \left( |\Delta r(s',\pi(s'))|^2 + \gamma^2 |\Delta P(s',\pi(s'))^\top V_\phi^\pi|^2 \right),$$
$$\leq 2 \left( 2\mathrm{KL}_{q_\phi,q_\psi}(s',\pi(s')) + 8\gamma^2 \varphi^2 M_{s'\pi(s')}^{(\bar{k}(s',\pi(s')))} [V_\phi^\star]^{2^{1-\bar{k}(s',\pi(s'))}} \mathrm{KL}_{P_\phi,P_\psi}(s',\pi(s')) \right),$$
$$\leq 4C^\pi(s')(\mathrm{KL}_{q_\phi,q_\psi}(s',\pi(s')) + \mathrm{KL}_{P_\phi,P_\psi}(s',\pi(s'))),$$

with $C^\pi(s') = \max\left(1, 4\gamma^2\varphi^2 M_{s'\pi(s')}^{(\bar{k}(s',\pi(s')))}[V_\phi^\star]^{2^{1-\bar{k}(s',\pi(s'))}}\right)$. Therefore

$$\alpha_3^2 \frac{(1-\gamma)^2}{(1+\gamma)^2}\Delta_\pi(s,a)^2 \leq 4\max_{s'} C^\pi(s')(\mathrm{KL}_{q_\phi,q_\psi}(s',\pi(s')) + \mathrm{KL}_{P_\phi,P_\psi}(s',\pi(s'))),$$

$$= 4\max_{s'} \frac{\omega(s',\pi(s'))}{\omega(s',\pi(s'))}C^\pi(s')(\mathrm{KL}_{q_\phi,q_\psi}(s',\pi(s')) + \mathrm{KL}_{P_\phi,P_\psi}(s',\pi(s'))),$$

$$\leq 4\max_{\tilde{s}} \frac{C^\pi(\tilde{s})}{\omega(\tilde{s},\pi(\tilde{s}))}\max_{s'}\omega(s',\pi(s'))(\mathrm{KL}_{q_\phi,q_\psi}(s',\pi(s')) + \mathrm{KL}_{P_\phi,P_\psi}(s',\pi(s'))).$$

In conclusion, we have the following set of inequalities:

$$\frac{\omega(s,a)(\alpha_1\Delta_\pi(s,a))^2}{2} \leq \omega(s,a)\mathrm{KL}_{q_\phi,q_\psi}(s,a),$$

$$\frac{\omega(s,a)(\alpha_2\Delta_\pi(s,a))^2}{8\gamma^2\varphi^2 M_{P_\phi(s,a)}^{(\bar{k}(s,a))}[V_\phi^\star]^{2^{1-\bar{k}(s,a)}}} \leq \omega(s,a)\mathrm{KL}_{P_\phi,P_\psi}(s,a),$$

$$\min_{s'}\frac{\omega(s',\pi(s'))(\alpha_3(1-\gamma)\Delta_\pi(s,a))^2}{4C^\pi(s')(1+\gamma)^2} \leq \max_{s'}\omega(s',\pi(s'))(\mathrm{KL}_{q_\phi,q_\psi}(s',\pi(s'))$$
$$+ \mathrm{KL}_{P_\phi,P_\psi}(s',\pi(s'))).$$

As in [37] we observe that we can replace $\alpha_i$ by $\alpha_i/\sum_j \alpha_j$ (since $\sum_i \alpha_i > 1$). Consequently, denoting by $\Delta_n$ the $n$-dimensional simplex, we have

$$T_\varepsilon(\omega)^{-1} \geq \min_{\pi\in\Pi_0^\star(\phi)} \min_{s,a\neq\pi(s)} \inf_{\psi\in\bar{\mathrm{Alt}}_{\pi,sa,\varepsilon}(\phi)} \omega(s,a)\mathrm{KL}_{\phi|\psi}(s,a) + \sum_{s'}\omega(s',\pi(s'))\mathrm{KL}_{\phi|\psi}(s',\pi(s')).$$

$$\geq \min_{\pi\in\Pi_0^\star(\phi)} \min_{s,a\neq\pi(s)} \inf_{\alpha\in\Delta_3} \sum_{i=1}^3 B_i(s,a)\alpha_i^2.$$

where

$$B_1(s,a) = \omega(s,a)\Delta_\pi(s,a)^2/2,$$

$$B_2(s,a) = \omega(s,a)\frac{\Delta_\pi(s,a)^2}{8\gamma^2\varphi^2 M_{sa}^{(\bar{k}(s,a))}[V_\phi^\star]^{2^{1-\bar{k}(s,a)}}},$$

$$B_3(s,a) = \min_{s'}\omega(s',\pi(s'))\frac{(\Delta_\pi(s,a)(1-\gamma))^2}{4C^\pi(s')(1+\gamma)^2}.$$

Therefore $T_\varepsilon(\omega)^{-1} \geq \min_{\pi\in\Pi_0^\star(\phi)} \min_{s,a\neq\pi(s)} \left(\sum_{i=1}^3 B_i(s,a)^{-1}\right)^{-1}$, from which we conclude that:

$$T_\varepsilon(\omega) \leq \max_{\pi\in\Pi_0^\star(\phi)} \max_{s,a\neq\pi(s)} \left(\frac{2+8\gamma^2\varphi^2 M_{sa}^{(\bar{k}(s,a))}[V_\phi^\star]^{2^{1-\bar{k}(s,a)}}}{\Delta_\pi(s,a)^2\omega(s,a)} + \max_{s'}\frac{4C^\pi(s')(1+\gamma)^2}{\omega(s',\pi(s'))\Delta_\pi(s,a)^2(1-\gamma)^2}\right).$$

(20)

$\square$

### C.2.3  Closed form solution under the generative model

Under the generative model, we are able to find a closed-form solution of the sample complexity upper bound by slightly relaxing our upper bound of $T_\varepsilon(\omega)$. The procedure is similar to that used in [37].

**Theorem C.5.** *Let $\varepsilon \geq 0$, and a communicating MDP $\phi$, with a unique optimal policy $\pi^star$. Then, for all $\omega \in \Delta(S \times A)$, we have:*

$$T_\varepsilon(\omega) \leq U(\omega) \leq \tilde{U}(\omega),$$ (21)

*where $U(\omega)$ is defined in the previous theorem, and*

$$\tilde{U}(\omega) = \max_{s,a\neq\pi^\star(s)} \frac{2+8\varphi^2 M_{sa}^{(\bar{k}(s,a))}[V_\phi^\star]^{2^{1-\bar{k}(s,a)}}}{\Delta(s,a)^2\omega(s,a)} + \frac{\max_{s'} 4C^{\pi^\star}(s')(1+\gamma)^2}{\min_{\tilde{s}}\omega(\tilde{s},\pi^\star(\tilde{s}))\Delta_{\min}^2(1-\gamma)^2}.$$ (22)

*where $\Delta(s,a) := V_\phi^{\pi^\star}(s) - Q_\phi^{\pi^\star}(s,a)$.*

*Proof.* The proof follows from the previous theorem. Since there is a unique optimal policy we have $\Delta_\pi(s,a) \geq \Delta_{\min}$, and thus

$$\tilde{U}(\omega) \leq \max_{s,a \neq \pi^\star(s)} \frac{2 + 8\varphi^2 M_{sa}^{(\bar{k}(s,a))}[V_\phi^\star]^{2^{1-\bar{k}(s,a)}}}{\Delta(s,a)^2 \omega(s,a)} + \max_{s'} \frac{4C^{\pi^\star}(s')(1+\gamma)^2}{\omega(s',\pi^\star(s'))\Delta_{\min}^2(1-\gamma)^2},$$

$$\leq \max_{s,a \neq \pi^\star(s)} \frac{2 + 8\varphi^2 M_{sa}^{(\bar{k}(s,a))}[V_\phi^\star]^{2^{1-\bar{k}(s,a)}}}{\Delta(s,a)^2 \omega(s,a)} + \frac{\max_{s'} 4C^{\pi^\star}(s')(1+\gamma)^2}{\min_{\tilde{s}} \omega(\tilde{s},\pi^\star(\tilde{s}))\Delta_{\min}^2(1-\gamma)^2}.$$

$\square$

For this particular bound, as in [37], we are able to find a closed form expression of the optimal generative allocation $\omega^\star \in \arg\min_{\omega \in \Delta(S \times A)} \tilde{U}(\omega)$ leading to an upper bound of the sample complexity lower bound. The following corollary is obtained by simply solving the optimization problem $\inf_{\omega \in \Delta(S \times A)} \tilde{U}(\omega)$.

**Corollary C.6.** *Consider a communicating MDP with unique optimal policy. Consider the bound defined in the previous theorem by $\tilde{U}(\omega)$. Then, the generative solution $\omega^\star = \arg\inf_{\omega \in \Delta(S \times A)} \tilde{U}(\omega)$ is given by*

$$\omega(s,a) = \begin{cases} H(s,a)/\Gamma & s,a \neq \pi^\star(s), \\ \sqrt{H \sum_{s,a \neq \pi^\star(s)} H(s,a)/|S|}/\Gamma & otherwise. \end{cases} \tag{23}$$

*where*

$$H(s,a) = \frac{2 + 8\varphi^2 M_{sa}^{(\bar{k}(s,a))}[V_\phi^\star]^{2^{1-\bar{k}(s,a)}}}{\Delta(s,a)^2 \omega(s,a)}, \quad H = \max_{s'} \frac{4C^{\pi^\star}(s')(1+\gamma)^2}{\Delta_{\min}^2(1-\gamma)^2}, \tag{24}$$

$$\Gamma = \sum_{s,a \neq \pi^\star(s)} H(s,a) + \sqrt{|S|H \sum_{s,a \neq \pi^\star(s)} H(s,a)}. \tag{25}$$

*Furthermore, the value of the problem is:*

$$\inf_{\omega \in \Delta(S \times A)} \tilde{U}(\omega) = \left( \sqrt{\sum_{s,a \neq \pi^\star(s)} H(s,a)} + \sqrt{|S|H} \right)^2 \leq 2 \left( \sum_{s,a \neq \pi^\star(s)} H(s,a) + |S|H \right). \tag{26}$$

### C.2.4 Technical lemmas

We finally state and prove the lemmas used in the derivation of our upper bounds of the sample complexity lower bound. These lemmas can be seen as an alternative to Lemma 4 used by the authors of [37] to derive their upper bounds.

In what follows, we consider a finite set $\Omega = \{\omega_1, \ldots, \omega_N\}$. For each $\omega \in \Omega$, let $f(\omega)$ be a real number, and we define the vector $\mathbf{f}(\Omega) = [f(\omega_1) \quad \ldots \quad f(\omega_N)]^\top$.

We start by a result, that can be deducted from the proof of Lemma 4 in [37].

**Lemma C.7.** *Let $P, Q$ be pmfs over some finite space $\Omega = \{\omega_1, \ldots, \omega_N\}$. Let $f : \Omega \to \mathbb{R}$ and $\mathbf{f}(\Omega) := [f(\omega_1) \quad \ldots \quad f(\omega_N)]^\top$.*
*Finally, we introduce the elementwise power[2] $\mathbf{f}^{\circ k}(\Omega) = \left[f(\omega_1)^k \quad \ldots \quad f(\omega_N)^k\right]^\top$. Then*

$$|(P-Q)^\top \mathbf{f}(\Omega)|^2 \leq 4d_H(P,Q)^2 \left( 2\mathbb{E}_{\omega \sim Q}[f(\omega)^2] + (P-Q)^\top (\mathbf{f}^{\circ 2}(\Omega)) \right), \tag{27}$$

*where $d_H$ is the Hellinger distance.*

---
[2]also known as as Hadamard power.

*Proof.* The proof can be easily deduced from Lemma 4 in [37]. We present the proof for completeness. Let $\sqrt{P}$ be the square root of the elements in $P$ (sim. $\sqrt{Q}$). We have:

$$(P - Q)^\top \mathbf{f}(\Omega) = \sum_\omega (P(\omega) - Q(\omega)) f(\omega),$$

$$= \sum_\omega (\sqrt{P(\omega)} - \sqrt{Q(\omega)})(\sqrt{P(\omega)} + \sqrt{Q(\omega)}) f(\omega),$$

$$= (\sqrt{P} - \sqrt{Q})^\top [(\sqrt{P} + \sqrt{Q}) \circ \mathbf{f}(\Omega)],$$

where $\circ$ is the Hadamard product. We apply the Cauchy-Schwartz inequality to the last term to get:

$$|(P - Q)^\top \mathbf{f}(\Omega)|^2 \leq \|\sqrt{P} - \sqrt{Q}\|_2^2 \|(\sqrt{P} + \sqrt{Q}) \circ \mathbf{f}(\Omega)\|_2^2.$$

Note that $\|\sqrt{P} - \sqrt{Q}\|_2 = \sqrt{2} d_H(P, Q)$. Regarding $\|(\sqrt{P} + \sqrt{Q}) \circ \mathbf{f}(\Omega)\|_2$, using the inequality $(a + b)^2 \leq 2(a^2 + b^2)$, we have:

$$\|(\sqrt{P} + \sqrt{Q}) \circ \mathbf{f}(\Omega)\|_2^2 \leq 2 \sum_\omega (P(\omega) + Q(\omega)) f(\omega)^2,$$

$$= 2 \sum_\omega (2Q(\omega) + P(\omega) - Q(\omega)) f(\omega)^2,$$

$$= 2 \left( 2\mathbb{E}_{\omega \sim Q}[f(\omega)^2] + (P - Q)^\top \mathbf{f}^{\circ 2}(\Omega) \right),$$

which concludes the proof. $\qquad\square$

Applying the above lemma recursively, we obtain the following result.

**Lemma C.8.** *Consider $f : \Omega \to \mathbb{R}$ as before. Assume that $\max_{\omega \in \Omega} |f(\omega)| \leq F < \infty$. Then,*

$$|(P - Q)^\top \mathbf{f}(\Omega)| \leq \sqrt{8} \varphi d_H(P, Q) \sup_{k \geq 1} \mathbb{E}_{\omega \sim Q}[f(\omega)^{2^k}]^{2^{-k}}, \tag{28}$$

*where $\varphi$ is the golden ratio.*

*Proof.* The idea is to observe that we can use Lemma C.7 to bound $(P - Q)^\top \mathbf{f}^{\circ 2}(\Omega)$ in Equation (27). Then

$$|(P - Q)^\top \mathbf{f}^{\circ k}(\Omega)|^2 \leq 4 d_H(P, Q)^2 \left( 2\mathbb{E}_{\omega \sim Q}[f(\omega)^{2k}] + (P - Q)^\top \mathbf{f}^{\circ 2k}(\Omega) \right).$$

For brevity, let $M_k = \mathbb{E}_{\omega \sim Q}[f(\omega)^k]$, then

$$|(P - Q)^\top \mathbf{f}(\Omega)| \leq 2 d_H(P, Q) \sqrt{2M_2 + (P - Q)^\top \mathbf{f}^{\circ 2}(\Omega)},$$

$$\leq 2 d_H(P, Q) \sqrt{2M_2 + 2 d_H(P, Q) \sqrt{2M_4 + (P - Q)^\top \mathbf{f}^{\circ 4}(\Omega)}},$$

$$\leq \alpha \sqrt{2M_2 + \alpha \sqrt{2M_4 + \alpha \sqrt{2M_8 + \cdots}}},$$

where $\alpha = 2 d_H(P, Q)$. A further rewriting yields

$$\alpha \sqrt{2M_2 + \alpha \sqrt{2M_4 + \alpha \sqrt{2M_8 + \cdots}}},$$

$$= \sqrt{2\alpha^2 M_2 + \alpha^3 \sqrt{2M_4 + \alpha \sqrt{2M_8 + \cdots}}},$$

$$= \sqrt{2\alpha^2 M_2 + \sqrt{2\alpha^6 M_4 + \alpha^7 \sqrt{2M_8 + \cdots}}},$$

$$= \sqrt{2\alpha^2 M_2 + \sqrt{2\alpha^6 M_4 + \sqrt{2\alpha^{14} M_8 + \cdots}}},$$

and note that the $k$-th term is given by $a_k = 2\alpha^{2(2^k-1)}M_{2^k}$. Consider now the sequence $b_k = (a_k)^{2^{-k}}$, and note that

$$\sup_{k \geq 1} b_k \leq \sup_{k \geq 1} \underbrace{\left(2\alpha^{2(2^k-1)}\right)^{2^{-k}}}_{(\bullet)} \cdot \sup_{k \geq 1} M_{2^k}^{2^{-k}}.$$

Observe that $(\bullet) = 2^{2^{-k}}\alpha^{2-2^{-k+1}}$ is a positive decreasing sequence, therefore we have that $\sup_{k\geq 1} b_k \leq \alpha\sqrt{2} \cdot \sup_{k\geq 1} M_{2^k}^{2^{-k}}$.

Now, we notice that $M_{2^k}^{2^{-k}}$ is bounded for all $k \geq 1$ from the boundedness of $f$ over $\Omega$

$$M_{2^k}^{2^{-k}} = \mathbb{E}_\omega[f(\omega)^{2^k}]^{2^{-k}} \leq F < \infty.$$

Hence, by letting $M = \alpha\sqrt{2} \cdot \sup_{k\geq 1} M_{2^k}^{2^{-k}}$, and using Herschfeld's convergence theorem [26], we find the desired result:

$$\sqrt{2\alpha^2 M_2 + \sqrt{2\alpha^6 M_4 + \sqrt{2\alpha^{14} M_8 + \cdots}}}$$

$$\leq \sqrt{M^2 + \sqrt{M^{2^2} + \sqrt{M^{2^3} + \cdots}}},$$

$$= M\sqrt{1 + \sqrt{1 + \sqrt{1 + \cdots}}} = M\varphi.$$

$\square$

We are now ready to state the result that serves as an alternative to Lemma 4 in [37]. Let $(\Delta P(s,a))_{s'} = P_\psi(s'|s,a) - P_\phi(s'|s,a)$.

**Lemma C.9.** *Consider a fixed state-action pair $(s,a)$ and define $\bar{V}_\phi^\pi(s,a) := \mathbb{E}_{s'\sim P_\phi(\cdot|s,a)}[V_\phi^\pi(s')]$. Let $f_\phi^\pi(s,a,s') = V_\phi^\pi(s') - \bar{V}_\phi^\pi(s,a)$ and $M_k(s,a) = \mathbb{E}_{s'\sim P_\phi(\cdot|s,a)}[f_\phi^\pi(s,a,s')^{2^k}]$. Then, there exists $\bar{k} \in \mathbb{N}$ such that*

$$|\Delta P(s,a)^\top \mathbf{f}_\phi^\pi(s,a)|^2 \leq 8\varphi^2 \mathrm{KL}_{P_\phi,P_\psi}(s,a)M_{\bar{k}}(s,a)^{2^{1-\bar{k}}}, \tag{29}$$

*where $\mathbf{f}_\phi^\pi(s,a) = \begin{bmatrix} f_\phi^\pi(s,a,s_1) & f_\phi^\pi(s,a,s_2) & \cdots & f_\phi^\pi(s,a,s_{|S|}) \end{bmatrix}^\top$ and $\mathrm{KL}_{P_\phi,P_\psi}(s,a) = \mathrm{KL}(P_\phi(s,a), P_\psi(s,a))$.*

*Proof.* Consider a fixed $(s,a)$. For any $s' \in S$ we have that $|f_\phi^\pi(s,a,s')| \leq \mathrm{MD}_{sa}[V_\phi^\pi]$, therefore $\|\mathbf{f}_\phi^\pi(s,a)\|_\infty < \infty$. Using Lemma C.8 with $\mathbf{f}_\phi^\pi(s,a)$ we find the result by taking the square on both sides, and using that $d_H^2(P,Q) \leq \mathrm{KL}(P,Q)$.

$\square$

### C.2.5 Decomposition of the set of confusing MDPs

Decomposing the set $\mathrm{Alt}_\varepsilon(\phi)$ directly presents several challenges. It does even seem possible to obtain a decomposition easy to work with. Instead, we relax the problem and work on $\mathrm{Alt}(\phi) = \{\psi : \psi \ll \phi, \Pi_0^\star(\phi) \cap \Pi_0^\star(\psi) = \emptyset\}$, a set containing $\mathrm{Alt}_\varepsilon(\phi)$.

**Lemma C.10.** *Let $\varepsilon \geq 0$. Then, in general $\mathrm{Alt}_\varepsilon(\phi) \subseteq \mathrm{Alt}(\phi)$.*

*Proof.* The statement can be derived by contradiction: assume that $\psi \in \mathrm{Alt}_\varepsilon(\phi)$ does not belong to $\mathrm{Alt}(\phi)$. However, that implies that there is $\pi \in \Pi_0^\star(\phi)$ s.t. $\pi \in \Pi_0^\star(\psi)$, which is not true since by assumption $\Pi_\varepsilon^\star(\phi) \cap \Pi_\varepsilon^\star(\psi) = \emptyset$. $\square$

**Lemma C.11.** *Let $\mathrm{Alt}(\phi) = \{\psi : \psi \ll \phi, \Pi_0^\star(\phi) \cap \Pi_0^\star(\psi) = \emptyset\}$. Then $\mathrm{Alt}(\phi) \subseteq \cup_{\pi\in\Pi_0^\star(\phi)} \cup_s \cup_{a\neq\pi(s)} \mathrm{Alt}_{\pi,sa}(\phi)$, where*

$$\mathrm{Alt}_{\pi,sa}(\phi) = \{\psi : \psi \ll \phi, Q_\psi^\pi(s,a) > V_\psi^\pi(s)\}.$$

*Proof.* The proof follows the same steps as that of the decomposition lemma in [37], and we give it for completeness.

By contradiction, consider $\psi \in \mathrm{Alt}(\phi)$ s.t. for all $\pi \in \Pi_0^\star(\phi)$ and $s, a \neq \pi(s)$ we have $Q_\psi^\pi(s, a) \leq V_\psi^\pi(s)$. Since $Q_\psi^\pi(s, \pi(s)) = V_\psi^\pi(s)$, the following inequality holds for all $\pi \in \Pi_0^\star(\phi)$ and for all $(s, a)$

$$Q_\psi^\pi(s, a) \leq V_\psi^\pi(s).$$

Define the Bellman operator for a generic policy $\pi'$ under $\psi$ as $(T_\psi^{\pi'} V)(s) = r_\psi(r, \pi'(s)) + \mathbb{E}_{s' \sim P(s, \pi'(s))}[V(s')]$. Then, from the above inequality that holds for all $(s, a)$ we get the following result

$$T_\psi^{\pi_\psi^\star} V_\psi^\pi \leq V_\psi^\pi.$$

By monotonicity of the Bellman operator, we get $T_\psi^{\pi_\psi^\star} T_\psi^{\pi_\psi^\star} V \leq T_\psi^{\pi_\psi^\star} V_\psi^\pi \leq V_\psi^\pi$. Iterating, we find

$$V_\psi^{\pi_\psi^\star} = \lim_{n \to \infty} \left( T_\psi^{\pi_\psi^\star} \right)^n V \leq V_\psi^\pi,$$

which is a contradiction since $\pi$ is not optimal under $\psi$.

$\square$