# OpenReview forum: "Model-Free Active Exploration in Reinforcement Learning"
_NeurIPS.cc/2023/Conference — NeurIPS 2023 poster_

### Official Review · Reviewer_1tWW · 2023-07-03

**Soundness:** 4 excellent
**Presentation:** 3 good
**Contribution:** 4 excellent
**Rating:** 7
**Confidence:** 4

**Summary:**

In this paper, the authors propose a way of obtaining tighter PAC bounds for model-free reinforcement learning. The new theoretical results allow the authors to propose new practical methods for exploration in both discrete and continuous state spaces. The proposed algorithms use ensembles of Q-values, and the results are very competitive when compared with model-based approaches.

**Strengths:**

* The new methods adapt to specific problems in an automated way.

* The paper is well written, and the discussion is supported by relevant citations. The contributions are backed up by both theoretical arguments and empirical results on a range RL domains.

* The contributions seem relevant and important. The results indicate that the proposed methods have merit.

* The paper comes with a comprehensive and long appendix (which would be a pain to read for a busy reviewer), but the future readers will most certainly appreciate it. The appendix provides many details that contribute to the quality of the work.


**Weaknesses:**

* The authors should make it clear from the start that the new algorithms are ensembles. After reading a few pages of the paper, I was anticipating a clever algorithm that would be purely based on the proposed theory. It was a bit disappointing that ensembles were used at the end. I don't say that this is bad, well there is no free lunch, but it would be fair to expect that, when the new algorithms are mentioned for the first time, ensembles are mentioned too.

* The paper argues that model-base approaches are expensive to run. The methods presented in this paper are ensembles. What is their runtime? Are that they really faster than model-based methods?

* Since the new methods proposed in this paper are ensembles, I would expect some discussion on ensembles in RL. I would expect that there must exist ensemble based RL methods that could be competitive with the methods presented here. Literature review on ensembles in RL would be useful. A classic paper on this is: Wiering, M.A. and Van Hasselt, H., 2008. Ensemble algorithms in reinforcement learning. IEEE Transactions on Systems, Man, and Cybernetics, Part B (Cybernetics), 38(4), pp.930-936.

**Questions:**

* How exactly is the alearotic uncertainty in the value function accounted for by the methods proposed in this paper?

* There is probability p in Alg. 1. It would be useful if its role and rationale were explained.

* The use of the variance of the value function is slick, and the authors are careful saying that this addresses the aleatoric uncertainty only, but it would be useful if the authors explained why epistemic uncertainty is not addressed.

**Limitations:**

I tried to discuss this in the weaknesses and questions sections above. Even though some of my comments may appear critical, I am very positive about this paper. This is solid work.

---

> ### Author Rebuttal · Authors · 2023-08-08
>
> We really appreciate your  review and  positive endorsement of our paper's merits. We're particularly pleased that the reviewer recognizes the innovative approach to tighter PAC bounds, the quality of the writing, and the comprehensive appendix.
>
>
> > How exactly is the alearotic uncertainty in the value function accounted for by the methods proposed in this paper? [...] it would be useful if the authors explained why epistemic uncertainty is not addressed by the variance of the value function.
>
>  Your questions on aleatoric uncertainty, and why epistemic uncertainty is not addressed by the bound, are both very important.
>
> 1. Regarding the former question, the variance of the value function in the next state  represents a sort of "difficulty" in learning the optimal policy by looking at future trajectories (due to the stochasticity of the MDP). It quantifies the dispersion of  values in the next state. In contrast, while the sub-optimality gap provides a measure of how far our current policy is from the optimal one, the variance measures the uncertainty in the value estimate due to the aleatoric nature of the MDP. Finally, our bound generalizes to even-th moments of the value function, but the rationale remains the same.
> 2. Regarding the latter question, to our understanding, the reason why parametric/epistemic uncertainty is not addressed by the bound is that the result that we along with [1] propose can only be achieved asymptotically, i.e., when the parametric uncertainty has vanished (when all state-action pairs have been visited infinitely often). Our take on this is that the derivation of the bound does not account for the current uncertainty in the estimates of the $Q$-values and $M$-values. Using the certainty equivalence principle, we simply use the plug-in estimator of these values, without considering this uncertainty. This opens up an exciting research direction on how to incorporate this uncertainty knowledge in the derivation of the bound (perhaps using Bayesian methods).
>
> To overcome this issue, we leverage a bootstrapping approach to characterize this type of parametric uncertainty and use the ensemble to sample the $(Q,M)$-functions to derive the allocation strategy through Corollary 5.1.
>
>
>
>
>
> We hope that this explanation is clear and addresses the reviewer's questions.
>
>
> > There is probability $p$ in Alg. 1. It would be useful if its role and rationale were explained.
>
>  Regarding the mask probability computation $p$, it is a user-chosen hyper-parameter that draws similarity to classical Bootstrapping and can speed up the learning process. The higher the value of $p$, the less accurate the characterization will be of the parametric uncertainty of the $Q,M$-values, while a smaller value of $p$ may compromise exploration efficiency. While we also discuss more in detail the algorithm in the appendix, we will make sure to include a more detailed explanation of the masking probability in the revised version of the manuscript.
>
>
> > Clarification on Ensemble Approach.
>
> We understand and share the concern regarding the late introduction of the ensemble approach in our algorithms. To be more transparent, we will introduce the ensemble aspect earlier in the paper.
> We will also include a comprehensive review on ensembles in RL, incorporating the cited classic paper to contextualize our work.
> Regarding the use of ensembles, we found that randomized prior value functions provide an excellent way to reduce the computational complexity of model-based methods. We typically do not need to increase the number of ensemble members but simply tune the scale parameter of the random prior function.
> Lastly, we believe that model-based approaches are useful depending on the problem and can be used efficiently in conjunction with our method.
>
> \
> &nbsp;
>
> Thank you once again for the detailed feedback and your overall favorable view of our work.

---

> > ### Comment · Reviewer_1tWW · 2023-08-20
> > **Thank you for your response**
> >
> > Thank you for answering reviewers' questions. I don't have any other questions.

---

### Official Review · Reviewer_qwpS · 2023-07-05

**Soundness:** 3 good
**Presentation:** 2 fair
**Contribution:** 2 fair
**Rating:** 5
**Confidence:** 2

**Summary:**

The paper introduces a model-free exploration approach developed on an information theoretical basis. Firstly, the lower bound on the number of samples for a near-optimal policy is estimated, and based on this lower bound, the paper develops an exploration strategy for both tabular and deep RL approaches. The exploration strategy is further validated via experiments, where it is found to be superior to other competing approaches.

**Strengths:**

The paper is generally well written and addresses an important problem. The developed exploration strategy having an information theoretic basis sets it apart from some of the previous approaches which have mainly been based on heuristics.


**Weaknesses:**

The paper does not include experiments in more complex environments such as Montezuma’s revenge, where exploration is known to be a key challenge. A more detailed summary of the main intuitions and theoretical results from [1] could have made the paper more readable.


**Questions:**

1. Can more challenging environments possibly be added? If not, are there any fundamental limitations that make this infeasible?

2. Wouldn’t the approximation errors in the deep RL version of the algorithm affect the suboptimality gap? Is this somehow accounted for?

3. Apart from the current results, perhaps for some of the environments, the actual exploration trajectories could be traced/charted out to explicitly show how the exploration is modified. I noticed something along these lines included in the appendix, but it would be good to bring similar results forward into the main text.

4. It would benefit readers to include more details about the main results in [1].

5. Although it may be obvious, in the cartpole swingup environment, it would be good to include a brief paragraph regarding why/how increasing k makes the task more difficult.

6. In line 297, what is N?

7. I am not sure why Fig 3 shows the performance in Riverswim vs |S| and ForkedRiverswim vs N

8. How is $\Delta_{min}$ initialized in the Deep RL version?


**Limitations:**

Currently, there is no explicit section outlining the limitations. I suggest the authors include a short paragraph about this covering at least the basic assumptions and requirements for the proposed to be applicable. Perhaps some comments regarding the scalability of the approach to more complex environments could also be added if that is indeed an issue.

---

> ### Author Rebuttal · Authors · 2023-08-08
>
> Thank you for your  insights and constructive feedback. We appreciate the effort you invested in reviewing our paper and have carefully considered your suggestions. Below, we respond to your queries and address the highlighted concerns.
>
> > Currently, there is no explicit section outlining the limitations.
>
> We actually discuss limitations, including basic assumptions and requirements, in the appendix (right after the table of content, page 17) due to lack of space.
>
>
> > What is $N$?  Why Fig 3 shows the performance in Riverswim vs $|S|$ and ForkedRiverswim vs $N$? How is $\Delta_{min}$ initialized?
>
> We understand the confusion. The parameter $N$ is more properly explained in the appendix, Section A.2 (page 19). The Forked Riverswim environment consists of two rivers, each of length $N-1$, plus the starting state (that's why $|S|=2N-1$). To improve clarity, we can change the $x$-label of Figure 3 to maintain consistency.
>
> Regarding $\Delta_{min}$, we suggest to initialize it a small value. In the attached code, we can see that we initialized it to $10^{-6}$ for the Slipping DeepSea problem, and to $10^{-2}$ for the Cartpole Swingup problem.
>
> > Wouldn’t the approximation errors in the deep RL version of the algorithm affect the suboptimality gap?
>
> We understand the reviewer's concern. In this case, the approximation error that is introduced is, somehow, modeled by the parametric uncertainty, and thus  covered by the technique that we propose (as long as the neural networks are expressive enough to model the true $Q$-value function).
>
>
> > [...]the actual exploration trajectories could be traced/charted out to explicitly show how the exploration is modified.
>
> We appreciate your suggestion to bring forward some of the exploration trajectories into the main text of the paper. This could indeed help readers to better understand the practical impact of our exploration strategy. We will consider this for the final version of our paper, by adding perhaps an image in the introduction.
>
> > Can more challenging environments possibly be added? If not, are there any fundamental limitations that make this infeasible?
>
> As for conducting more complicated experiments, we understand the concern of the reviewer. There are not fundamental limitations that make this infeasible. We simply used hard exploration environments that are used for example by BSP [39] and the DeepMind BSuite library (where they propose the DeepSea problem and the Cartpole Swingup problem to assess exploration properties). Furthermore, we used a more difficult version of the classical DeepSea environment (because of the slipping probability), and evaluated the environments for various difficulty levels for all environments. However, we will try to include more environments and add plots to improve clarity.
>
>
> > It would benefit readers to include more details about the main results in [1].
>
>  Your suggestion to include more details about the main results from [1] is noted. We initially aimed for conciseness, but understand that more context could make the paper more readable. We will strive to strike a better balance in the revised version, by adding a more detailed explanation.
>
> > In the Cartpole Swingup environment, it would be good to include a brief paragraph regarding why/how increasing $k$ makes the task more difficult.
>
> For a detailed explanation of the environment, please refer to Section A.6 of the appendix (page 27). While we are constrained by the page limit for the main paper, we understand the reviewer's request for more details.We will make sure to clarify where to find more information in the main text of the manuscript.
>
> In the Cartpole Swingup problem the agent incurs in a negative reward unless the cart's position and the pole's angle satisfy, respectively, the conditions $|x|<1-k/20$ and $cos(\theta)>k/20$. Therefore we see that as $k$ increases, the agent needs to learn a more stabilizing controller to collect positive rewards.
>
>
> \
> &nbsp;
>
> Thank you again for your review and for being overall positive about the novelty and soundness of our approach.

---

> > ### Comment · Reviewer_qwpS · 2023-08-13
> > **Thanks for your respones.**
> >
> > I thank the authors for their responses. Overall, the paper could be improved in terms of readability. It seems most of the comments I raised are already addressed in the Appendix. I urge the authors to either bring these into the main paper as much as possible, or at least include sentences in the main paper pointing to the appropriate Appendix locations where these are addressed. It would also benefit readers if the high level messages behind the mathematical results are better emphasized to bring clarity to the overall contribution.

---

> > > ### Author Response · Authors · 2023-08-13
> > >
> > > Thanks a lot for your comments and feedback. In the revised version of the manuscript, we will address the points you raised and, to do so, move some parts from the appendices to the main document. We will also discuss in more detail the intuition behind our main mathematical results. Thank you again!

---

### Official Review · Reviewer_M4pn · 2023-07-09

**Soundness:** 2 fair
**Presentation:** 2 fair
**Contribution:** 3 good
**Rating:** 6
**Confidence:** 2

**Summary:**


The authors propose a model free approach to exploration in RL, that is based around best policy identification. This technique, unlike prior work, uses stochastic approximation to learn a lower bound on the policy performance based on collected samples.


**Strengths:**



- The paper presents an model-free approach that's heavily guided by theory, and results show how this technique can be effective in improving exploration in terms of improved performance as well as reduced performance variance.


**Weaknesses:**



- The paper is a bit tough to get through; being so theoretically heavy, some intuitive high-level explanation of their implications would be really helpful for the reader to follow along the thought process of the authors.

- It is unclear where the derivation of the many theorems is in the text. For example, looking for a derivation on how the authors reached at corollary 5.1, the closest reference I found in the appendix was section B.3, but that's just an explanation on how to use that corollary.

**Questions:**


- Could you please clarify where the proof and derivations of the different theorems are? For instance, for theorem 4.2 there is a nice explanation of the implications of the theorem, but it is not stated how that upper bound is derived. The same is true about corollary 5.1.
Please clarify this, as the evaluation of soundness of the paper heavily relies on being able to find how these theorems where proved.

- Algorithm 1 only samples actions according to the allocation omega, does this mean that over time omega converges to pi^*?

- Corollary 5.1, the denominator in  H_epsilon contains the term delta_min is defined in terms of the optimal policy for the MDP. How are you sampling according to corollary 5.1 (algorithm 1), when the optimal policy pi^* is needed to compute that value?
This is a bit unclear.

---

> ### Author Rebuttal · Authors · 2023-08-08
>
> Thank you for your insightful review and positive feedback. We appreciate the time you took to review our paper and have taken into consideration your suggestions. We address below your questions and discuss the perceived weaknesses.
>
>
> > Could you please clarify where the proof and derivations of the different theorems are?[...] Please clarify this, as the evaluation of soundness of the paper heavily relies on being able to find how these theorems where proved.
>
>  We apologize for any confusion regarding the location and details of the derivations.  The derivation of all the proofs can be found in the appendix, Section C. In Section C.1, we derive some general results, and later in Section C.2, we derive the alternative bound. The main lemma used to prove Theorem 4.2 is Lemma C.10 (line 1156 in the appendix). Corollary 5.1 is proved in Section C.2.3 (see Corollary C.8). Please, note that the bound that we derive in the appendix is \emph{more general} that the one we present in the main paper. In fact, the lower bound we find in the appendix holds also for MDPs with multiple $\varepsilon$-optimal policies. We also want to thank the reviewer for asking this question, we will include additional references in the revised manuscript to clarify where to find the proofs for Corollary 5.1 and Theorem 4.2.
>
>
>
> > Algorithm 1 only samples actions according to the allocation $\omega_t$, does this mean that over time $\omega_t$ converges to $\pi^\star$?
>
>  Note that the allocation $\omega_t$ is an exploration policy, and it _should not_  converge to the greedy policy $\pi^\star$. However, the samples that we obtain when exploring should be used to learn the greedy policy $\pi^\star$ in an off-policy way. Therefore, this method is inherently off-policy.
>
> As a final remark: the exploration policy $\omega_t$  is derived from Corollary 5.1. When applied in a model-free fashion, as we do, it should converge to the optimal allocation given by Corollary 5.1 projected on the set defined by the navigation constraints. For the interested reader that wants to try with a toy example, in the code you can find in `BoundsAnalysis/utils/utils.py` the function to compute the projection given an allocation and the transition function (`project_omega` or `compute_stationary_distribution`).
>
>
> > How are you sampling according to corollary 5.1 (algorithm 1), when the optimal policy $\pi^\star$ is needed to compute that value?
>
> The confusion surrounding the use of the optimal policy in Corollary 5.1's sampling is understandable. As in [1], and other papers that use information-theoretical arguments, we use a certainty-equivalence principle, where we use the current plug-in estimator of the quantity of interest. In our case, we use the $Q$-values of the greedy policy, learnt through off-policy learning, to derive $\pi_t^\star$, the current estimator of the greedy policy at time $t$.
>
> > ...some intuitive high-level explanation of their implications would be really helpful for the reader to follow along the thought process of the authors.
>
> We understand that the theoretical density of the paper can be challenging to navigate. We will include in the revised version a more intuitive high-level explanation of the concepts and their implications to assist the readers in grasping our thought process.
>
> For completeness, we briefly summarize the general idea is as follows: for an MDP $M$ the quantity $T_\varepsilon(\omega)$ represents the characteristic time, i.e., characterizes the sample complexity of estimating the optimal policy using an exploration policy $\omega$. Therefore, the minimum value $\min_\omega T_\varepsilon(\omega)$ yields the characteristic time of the lowest sample complexity that one can possibly achieve. Unfortunately, computing the best optimal exploration policy  $\arg\min_\omega T_\varepsilon(\omega)$ amounts to solving a non-convex problem (see [1] for an example).
>
> To address this challenge, the main idea is to find a convex upper bound of $T_\varepsilon(\omega)$, that we call $\bar U(\omega)$, and then compute the minimizer $\bar \omega^\star=\arg\min_\omega\bar U(\omega)$ (for example, using corollary 5.1 to obtain a closed form solution). Therefore, we can use $\bar \omega^\star$ to explore an environment.
>
>
> \
> &nbsp;
>
> We would like to thank you again for your review, and  for acknowledging the novelty and soundness of our model-free approach to exploration in RL.

---

> > ### Comment · Reviewer_M4pn · 2023-08-18
> > **Response to Authors**
> >
> > I thank the authors for their response. You comments and pointing to the right section in appendix for proofs helped clarify some questions, but keep in mind that readers will not be able to ask where to find the derivation of different theorems.
> > The paper with an appendix is over 50 pages long, so it is unreasonable to expect a reader to dig through it and find them.
> > I remain positive about the paper, but I think you can improve readability.
> > I suggest editing the paper so that for every theorem/derivation, you point to the exact appendix section where the proof can be found.
> > Some of those might even make sense to bring into the main text.

---

> > > ### Author Response · Authors · 2023-08-18
> > >
> > > Thank you for being positive about the paper and for your constructive feedback. Given the depth of content, we understand your concerns about navigating the manuscript.
> > >
> > > In our revised version, we will ensure that each theorem or significant point in the main text directly references the corresponding section in the appendix, making the manuscript more accessible. Moreover, based on the feedback, we'll also evaluate if certain crucial derivations or explanations should be shifted from the appendix to the main body to enhance clarity (especially intuitive high-level explanation of the theoretical results, and  their implications, as suggested also by some other reviewer).
> > >
> > > Once again, thank you for your feedback.

---

### Official Review · Reviewer_KZND · 2023-07-12

**Soundness:** 4 excellent
**Presentation:** 3 good
**Contribution:** 3 good
**Rating:** 7
**Confidence:** 3

**Summary:**

This paper introduces an approximation of a lower bound on the number of samples needed to identify a nearly optimal policy directly applicable to model free RL.
They further propose a model free exploration strategy that can be applied to the tabular and continuous MDPs.


**Strengths:**

The paper is clear and well written. With experiments showing the proposed approach seems competitive with other exploration approaches.

**Weaknesses:**

The paper lacks a comparison of the current approach with more common exploration strategies in DeepRL such as 34, 37.
The results remain very toy. It would be great to have an understanding of the kind of exploration the proposed approach performs well, with increasingly sparse rewards? on which circumstances is this preferable.



**Questions:**

Why have you used the 2k-th moments in the bound?
Can the authors clarify which eq. Compute Allocation, training and estimate minimum gap refer to?
How is the mask probability being computed? are the results sensitive to this parameter?




**Limitations:**

The authors could provide a more thorough limitation description of the approach under which circumstances does the proposed approach fail, how critical and reasonable are the assumptions made, for instance the communicating MDPs. How does the approach perform in more challenging tasks such as Montezuma's revenge, and more structured exploration tasks. Under which types of exploration difficulty does the proposed approach excel or fail.

---

> ### Author Rebuttal · Authors · 2023-08-08
>
> Thank you for your thorough review and positive feedback on our paper. We appreciate the constructive suggestions and have taken them into consideration. We address below your questions  and discuss the perceived weaknesses.
>
> > Why have you used the 2k-th moments in the bound?
>
>  There are multiple reasons for the use of 2k-th moments in the bound: (1) first, we wanted to find an alternative bound to [1] that did not involve the span of an MDP, since this quantity is difficult to estimate online (since it involves an $\infty$-norm), and may not characterize very well the difficulty of learning the MDP; (2) secondly, we found that the span of an MDP can be lower bounded by the square root of the  variance  of the value function (Lemma C.1 in the appendix, line 969). This prompted us to find an alternative bound, through an alternative proof technique (Lemma C.10 is the main tool, line 1156 in the appendix) that led us to the refinement of the bound in [1].
> This refinement allows us to characterize the bound in terms of the $2k$-th moment of the value function in each state-action space, which can be estimated online through the use of stochastic approximation.
>
> > Can the authors clarify which eq. Compute Allocation, training and estimate minimum gap refer to? How is the mask probability being computed? are the results sensitive to this parameter?
>
>  For your question about which equations Compute Allocation, training, and estimate minimum gap refer to, please refer to Algorithm 6 in the appendix, page 38.
>
> Regarding the mask probability computation $p$, it is a user chosen hyper-parameter. It draws similarity from classical Bootstrapping, and it can speed-up the learning process. The higher the value of $p$, the worse the characterization will be of the parametric uncertainty of the $Q,M$-values, while a smaller value of $p$ may compromise exploration efficiency. While we also discuss more in detail the algorithm in the appendix, we will make sure to include a more detailed explanation of the masking probability in the revised version of the manuscript.
>
>
>
> > The paper lacks a comparison of the current approach with more common exploration strategies in DeepRL such as 34, 37
>
>  Unless we misunderstood the reviewer comment, we compare with the bootstrapped technique from [34,37,38]. In the numerical results it is called BSP (Bootstrapping with additive prior [38], which is an advancement on classical Bootstrapped DQN). We mainly focused on comparing with other information-theoretical strategy for a fair comparison.
>
>
> > It would be great to have an understanding of the kind of exploration the proposed approach performs well, with increasingly sparse rewards? on which circumstances is this preferable.
>
>  From the numerical results, we see that the strategy performs well in environments with sparse reward. In general, we expect good performance in sparse reward environments due to characterization of the parametric uncertainty through Bootstrapping with random prior functions. This technique, combined with our proposed bound, allows us to derive effective exploration strategies that only explore where it is needed to learn the optimal value function. However, the exploration strategy, since it is guided from theory, relies on some assumptions such as having a communicating MDP. Therefore, it is possible that one may need to tailor it depending on the type of problem. For example, one may not need to explore at every step, but may try to pair the exploration policy with another policy depending on the problem. Finally, we remark that we discuss limitations, including basic assumptions and requirements, in the appendix (right after the table of content, page 17).
>
> \
> &nbsp;
>
> We thank the reviewer once again for your valuable feedback and for your positive endorsement of our paper's contributions.

---

### Official Review · Reviewer_VjsS · 2023-07-17

**Soundness:** 3 good
**Presentation:** 3 good
**Contribution:** 3 good
**Rating:** 6
**Confidence:** 2

**Summary:**

This work focuses on the exploration of reinforcement learning and introduces a novel model-free algorithm. The authors derive a new bound for the lower bound of the number of samples needed to identify a near-optimal policy. Based on that, they develop a model-free exploration strategy that is applicable to both tabular and continuous MDPs. Experimental results demonstrate that their strategy is competitive to existing approaches in efficiently identifying optimal policies.

**Strengths:**

- Theoretically, the new proposed bounds look novel and sound. They transform the original lower bound into a more manageable form, making it easier to handle and apply in practical scenarios.

- Empirically, the experiments confirm the superiority of the proposed algorithm over existing methods, aligning with their claims.

**Weaknesses:**

- There seems to be a big gap between the theory and the algorithm. The theoretical results involve certain quantities (e.g., $\Delta(s,a)$) that are unknown to the algorithm. The authors addressed this issue by approximating these quantities through Q-value (and M-value) learning, subsequently treating the approximated Q and M as ground truth for all computations. This introduces a substantial gap between the theory and the algorithm.

- It would improve if the authors could provide an intuitive explanation of the terms in the complicated expressions presented in Theorems 4.1 and 4.2. Understanding and evaluating these results as a whole seems challenging without a better explanation.

- Although I admit that the main contributions lie in the theoretical aspects, it would be better to conduct more complicated experiments.

**Questions:**

- Considering the gap between the theory and the algorithm mentioned above, is there any convergence guarantee for the algorithm?

- When the authors select $\bar k=1$ (line 211) and arrive at $\bar{U}^1_{\epsilon}$, does it remain an upper bound for $T_\epsilon$?

- I was a bit confused about the main idea. The authors first proposed an upper bound $\tilde{U}$ for the well-established lower bound $T_0$, and then they proceeded to carry out all computations based on this new bound. Although I understand that it is their point to derive this approximation that makes the bound easier to handle, it remains unclear to me why this approach is rational. Typically, what we usually do is to derive a more manageable upper bound on another harder upper bound, or derive a more manageable lower bound on another harder lower bound. Then we just need to minimize the new bound, because in doing so, the original bound is indirectly minimized (or maximized). However, this paper proposed an upper bound for the lower bound, making the aforementioned relationship inapplicable. Therefore, I am curious what the underlying principle here is.

**Limitations:**

No negative societal impact was identified.

---

> ### Author Rebuttal · Authors · 2023-08-08
>
> Thank you for your insightful comments and constructive feedback. We appreciate the time you took to review our paper and will address each of your comments individually.
>
>
> > When the authors select $\bar k=1$ (line 211) and arrive at $\bar U_\epsilon^1$, does it remain an upper bound for $T_\epsilon$ ?
>
>  Concerning your question about $\bar k=1$ and whether $\bar U_\epsilon^1$ remains an upper bound for $T_\epsilon$, the answer is no. Note that for each state-action pair $(s,a)$ the optimal value of $\bar k$ may differ. In principle, it is possible to extend our algorithm to account for this, at the cost of an increased computational complexity that does not seem worth given the numerical results that we obtained. Lastly, the choice of $\bar k=1$ follows simply from the scaling argument that we outline in the paper.
>
> > Considering the gap between the theory and the algorithm mentioned above, is there any convergence guarantee for the algorithm?
>
> Regarding the convergence guarantee for our algorithm, we understand your concern. In this context, asymptotic almost sure convergence of the $Q,M$-values is guaranteed if we mix the allocation with an $\epsilon$-soft policy (see, for example, Algorithm 6, line 6 in the `EstimateAllocation` function, page 38 of the appendix; similarly, also for the tabular case).
>
>
> However, compared to [1,28], it is hard to derive a sample complexity upper bound due to the various approximations. It is not  within the scope of this paper, but we believe that, in absence of approximations, we could apply similar ideas from [1] and [28] to find a sample complexity upper bound.
> Finally, since the $Q,M$-values are all you need to compute the optimal allocation of Corollary 5.1, then asymptotically, we can still converge a.s. to the minimizer of $\bar U_\varepsilon$.
>
> Lastly, regarding your concern about the gap, it is common in information-theoretical algorithms of this type to use the current estimate according to a certainty-equivalence principle (see for example [1,28] and other papers on best arm identification [17,22]).
>
>
>
> > I was a bit confused about the main idea. [...] Therefore, I am curious what the underlying principle here is.
>
> To clarify our approach of deriving an upper bound for the lower bound, we realize that the derivation is not straightforward.
>
> The idea is as follows: for an MDP $M$ the quantity $T_\varepsilon(\omega)$ represents the characteristic time, i.e., characterizes the sample complexity of estimating the optimal policy using an exploration policy $\omega$. Therefore, the minimum value $\min_\omega T_\varepsilon(\omega)$ yields the characteristic time of the lowest sample complexity that one can possibly achieve. Unfortunately, this minimum cannot be easily computed since it involves solving a non-convex problem (see [1] for an example). It is then not possible to find a non-trivial smaller bound that is convex, unless it works only for a certain type of MDP.
>
> However, it is possible to find a convex upper bound of $T_\varepsilon(\omega)$, that we call $\bar U(\omega)$, and then compute the minimizer $\bar \omega^\star=\arg\min_\omega\bar U(\omega)$. How far is this new minimum from the minimum of $T_\varepsilon$ is still an open question (and quite difficult to answer).
>
>
>
>
> > It would improve if the authors could provide an intuitive explanation of the terms in the complicated expressions presented in Theorems 4.1 and 4.2.
>
> Your suggestion to provide a more intuitive explanation for the complicated expressions in Theorems 4.1 and 4.2 is well taken. Using the explanation of the main idea in the paragraph above, we can see now how $\bar U(\omega)$ is a convex upper bound of $T(\omega)$ that we try to minimize. This upper bound is characterized by some quantities, e.g. $\Delta(s,a)$ and $M_{sa}^{\bar k}[V^\star]$. The former represents the sub-optimality gap (the hardness of learning the optimal action in a certain state). While the latter term represents a sort of variance of the value function in the next state (to be precise, the $2k$-th moment), and thus, it represents a sort of "difficulty" in learning the optimal policy by looking at future trajectories (it measures the uncertainty in the value estimate due to the aleatoric nature of the MDP).
>
> In comparison, in [1] they found a characterization based also on the span of the MDP, which is however not necessary in our formulation. To summarize, if we think of $\bar U$ as the characteristic time of the MDP, then we see that the sample complexity is completely characterized by the suboptimality gaps and the $2k$-th moment of the value function in each state.
>
>
> > Although I admit that the main contributions lie in the theoretical aspects, it would be better to conduct more complicated experiments.
>
>
> As for conducting more complicated experiments, we understand the concern of the reviewer. However, note that we used hard exploration environments that are also used by other practitioners when evaluating exploration algorithms ( see for example [39], or the Bsuite library, where they propose the DeepSea problem and the Cartpole swingup problem to assess exploration properties). Furthermore, we used a more difficult version of the classical DeepSea environment (because of the slipping probability), and evaluated the environments for various difficulty levels for all environments.
>
> \
> &nbsp;
>
> Thank you again for your review that will greatly help us to improve our paper. We are pleased that you acknowledged the novelty and soundness of our proposed bounds and how they transform the lower bound into a more manageable form.

---

> > ### Comment · Reviewer_VjsS · 2023-08-17
> >
> > Thanks for the detailed feedback. It addressed most of my concerns. However, I am not certain about the "certainty-equivalence principle" mentioned by the author, so I have reservations about the gap between the theory and the algorithm (as proposed in the first weakness). In addition, I am still worried about the third question (the reason for deriving an upper bound for the lower bound). I can see that the distance between the new minimum and the true minimum is usually hard to estimate. However, if we are deriving the lower bound for the lower bound, then it is fine because we usually don't need to care about the distance between them if we can improve the new lower bound. Nevertheless, since the authors are deriving an upper bound for a lower bound, if we can't estimate the distance between them, there is not any guarantee for the original problem. Hence, I am not sure how the proposed results can be applied.

---

> > > ### Author Response · Authors · 2023-08-18
> > >
> > > Thank you for your feedback on our paper. We understand your concerns and will attempt to elucidate further.
> > >
> > > > However, I am not certain about the "certainty-equivalence principle" mentioned by the author
> > >
> > > Using the certainty-equivalence principle means that we are using the current estimates (of the sub-optimality gap and the variance of the Q-function) as if they were exact when computing the exploration policy. This does not create a gap between theory and the algorithm, because in our algorithm, we mix the computed exploration policy with an $\epsilon$-soft policy, where the parameter $\epsilon$ is decreased over time (for example, see Lemma B.1 in Appendix to see what is the rate at which $\epsilon$ is decreased).
> > >
> > > In the code that we provided, you can see how we mix the allocation $\omega$ with this $\epsilon$-soft policy (e.g., in the tabular case, for MF-BPI, check the `forward` function in `RiverSwim/agents/mfbpi.py`, line 68 and 69). We first bootstrap the $Q,M$-values using the ensembles (to account for the parametric uncertainty), compute the exploration policy according to Corollary 5.1, and then mix with an $\epsilon$-soft policy (see also line 266-267 in the main paper).
> > >
> > > By doing this, we ensure enough exploration so that the estimation error vanishes with time (the estimates converge asymptotically). Mixing with an $\epsilon$-soft policy is often referred to as “forced exploration” in the literature.  We will make sure to clarify this point in the paper.
> > >
> > > Certainty-equivalence principles are common in the RL literature, see e.g. [1, 17, 22, 28, 49, 51, 52, 53]. We haven’t explain all this in detail, but we will add a more exhaustive discussion in the revised version of the manuscript.
> > >
> > >
> > > >   I am still worried about the third question (the reason for deriving an upper bound for the lower bound). I can see that the distance between the new minimum and the true minimum is usually hard to estimate. However, if we are deriving the lower bound for the lower  bound, then it is fine because we usually don't need to care about the distance between them if we can improve the new lower bound.  Nevertheless, since the authors are deriving an upper bound for a lower bound, if we can't estimate the distance between them, there is not  any guarantee for the original problem. Hence, I am not sure how the proposed results can be applied.
> > >
> > > The true lower bound (see e.g (1)) specifies the *minimum* amount of exploration needed to identify an approximately optimal policy with some level of certainty. The lower bound is information-theoretical, and cannot be beaten by any PAC algorithm. Hence, one cannot explore less, because this would imply that we could fail at identifying an approximately optimal policy. In other words, an algorithm starting from a lower bound of the lower bound would not enjoy any performance guarantee, because you are exploring less than needed.
> > >
> > > This is why we use the upper bound of the lower bound. In this way, we ensure that we identify an approximately optimal policy, but at a cost of  "over-exploring" a bit, at a rate corresponding  to the gap between the upper bound and the true lower bound.
> > >
> > > This approach is not unique to our study and has been adopted in several works like [1,28,53]. Guarantee-wise, in the paper, we show that our bound obtains a scaling that is comparable, if not better, to the minimax lower bound, as explained after Theorem 4.2 (please see also the related work section where we discuss the minimax lower bound). We believe this underpins the validity of our approach in the context of the problem that we study.
> > >
> > > We're always open to continued dialogue to improve and refine our work further. Thank you for your time and dedication to reviewing our paper.

---

> > > > ### Comment · Reviewer_VjsS · 2023-08-19
> > > >
> > > > Thanks for your additional response, which addressed my concerns. Although there is still some limitation (eg, unclear how to estimate the gap), I believe the contribution is already significant enough. I have raised my rating to 6 to vote for acceptance.

---

> > > > > ### Author Response · Authors · 2023-08-20
> > > > >
> > > > > Thank you for taking the time to engage in this discussion and for recognizing the significance of our work. We genuinely appreciate your feedback and we will keep your suggestions into account to further improve our manuscript, especially in terms of adding more intuitive explanations of the results.

---

### Decision · Program_Chairs · 2023-09-21

**Decision:**

Accept (poster)

**Comment:**

After the discussion, the reviewers in general agree the algorithm is novel and proposed method and analysis provide a good contribution to the problem of exploration in RL.